# Functional connectomics reveals general wiring rule in mouse visual cortex

Zhuokun Ding[1,2,3,4,26], Paul G. Fahey[1,2,3,4,26], Stelios Papadopoulos[1,2,3,4,26], Eric Y. Wang[1], Brendan Celii[1,5], Christos Papadopoulos[1], Andersen Chang[1], Alexander B. Kunin[1,6], Dat Tran[1], Jiakun Fu[1,7], Zhiwei Ding[1], Saumil Patel[1,2,3,4], Lydia Ntanavara[1,2,3,4], Rachel Froebe[1,2,3,4], Kayla Ponder[1], Taliah Muhammad[1], J. Alexander Bae[8,9], Agnes L. Bodor[10], Derrick Brittain[10], JoAnn Buchanan[10], Daniel J. Bumbarger[10], Manuel A. Castro[8], Erick Cobos[1], Sven Dorkenwald[8,11], Leila Elabbady[10], Akhilesh Halageri[8], Zhen Jia[8,11], Chris Jordan[8], Dan Kapner[10], Nico Kemnitz[8], Sam Kinn[10], Kisuk Lee[8,12], Kai Li[11], Ran Lu[8], Thomas Macrina[8,11], Gayathri Mahalingam[10], Eric Mitchell[8], Shanka Subhra Mondal[8,9], Shang Mu[8], Barak Nehoran[8,11], Sergiy Popovych[8,11], Casey M. Schneider-Mizell[10], William Silversmith[8], Marc Takeno[10], Russel Torres[10], Nicholas L. Turner[8,11], William Wong[8], Jingpeng Wu[8], Wenjing Yin[10], Szi-chieh Yu[8], Dimitri Yatsenko[1,13], Emmanouil Froudarakis[1,14,15], Fabian Sinz[1,16,17], Krešimir Josić[18], Robert Rosenbaum[19], H. Sebastian Seung[8,11], Forrest Collman[10], Nuno Maçarico da Costa[10], R. Clay Reid[10], Edgar Y. Walker[20,21], Xaq Pitkow[1,5,22,23,24], Jacob Reimer[1✉] & Andreas S. Tolias[1,2,3,4,5,25✉]

Understanding the relationship between circuit connectivity and function is crucial for uncovering how the brain computes. In mouse primary visual cortex, excitatory neurons with similar response properties are more likely to be synaptically connected[1–8]; however, broader connectivity rules remain unknown. Here we leverage the millimetre-scale MICrONS dataset to analyse synaptic connectivity and functional properties of neurons across cortical layers and areas. Our results reveal that neurons with similar response properties are preferentially connected within and across layers and areas—including feedback connections—supporting the universality of 'like-to-like' connectivity across the visual hierarchy. Using a validated digital twin model, we separated neuronal tuning into feature (what neurons respond to) and spatial (receptive field location) components. We found that only the feature component predicts fine-scale synaptic connections beyond what could be explained by the proximity of axons and dendrites. We also discovered a higher-order rule whereby postsynaptic neuron cohorts downstream of presynaptic cells show greater functional similarity than predicted by a pairwise like-to-like rule. Recurrent neural networks trained on a simple classification task develop connectivity patterns that mirror both pairwise and higher-order rules, with magnitudes similar to those in MICrONS data. Ablation studies in these recurrent neural networks reveal that disrupting like-to-like connections impairs performance more than disrupting random connections. These findings suggest that these connectivity principles may have a functional role in sensory processing and learning, highlighting shared principles between biological and artificial systems.

In the late 1800s, Santiago Ramón y Cajal—while poring over the structure of Golgi-stained neurons using only light microscopy—imagined the 'neuron doctrine', the idea that individual neurons are the fundamental units of the nervous system[9]. Implicit in the neuron doctrine is the idea that the functions of individual neurons—their role in what we would now call neural computation—are inextricably linked to their connectivity in neural circuits. A variety of influential proposals about the relationship between connectivity and function have been advanced in the past century. For example, Donald Hebb's cell assembly hypothesis[10]—colloquially stated as "neurons that fire together, wire together"—predicted that interconnected neuronal subnetworks 'reverberate' to stabilize functionally relevant activity patterns. In the cortical visual system, Hubel and Wiesel proposed that the hierarchical organization of connected neurons might build more complex feature preferences from simpler ones; for example, the position invariance of orientation-selective complex cells might be derived from convergent inputs of like-oriented simple cells with spatially scattered receptive fields (RFs)[11,12].

Testing these predictions has been difficult because of the challenges of measuring neural activity and synaptic-scale connectivity in the same population of neurons. In the mammalian visual cortex, evidence for several varieties of like-to-like connectivity (that is, increased connectivity for cells with similar response preferences) has been found via spine imaging[6], combined in vivo imaging and in vitro multipatching[1–3,13], combined in vivo imaging and rabies monosynaptic retrograde tracing[4,7], and combined in vivo imaging with electron microscopy (EM) reconstruction[5,8]. However, a caveat of these important early studies is that they have mostly been limited to small volumes, usually single lamina of primary visual cortex (except refs. 4,7), mostly owing to the challenge of identifying synaptic connections between functionally characterized neurons across distances larger than a few hundred micrometres. Thus, many questions remain unanswered about how these rules generalize across areas and layers.

The MICrONS dataset is the largest functionally imaged EM dataset so far[14], with mesoscopic calcium imaging[15] performed in vivo and subsequent EM imaging[16,17] and dense reconstruction[18–24] for an approximately 1 mm³ volume spanning visual cortical areas primary visual cortex (V1), anterolateral (AL), lateromedial (LM) and rostrolateral (RL) higher visual areas in a single mouse. In contrast to previous studies that have selectively reconstructed presynaptic or postsynaptic partners of a small set of functionally characterized target cells[5,25], the MICrONS volume is densely reconstructed, offering access to segmentation of all neurons in the volume and enabling analyses that are not possible in targeted sparse reconstructions. Here we take advantage of the dense reconstruction to compare the functional similarity of connected pairs with unconnected 'bystanders'—pairs of neurons with closely apposed axons and dendrites that had the opportunity to form synaptic connections, yet did not.

Our analysis of functional similarity builds on recent advances in using machine learning to characterize the response properties of neurons in visual cortex. By training a neural network to replicate the responses of recorded neurons across a rich stimulus set of natural and parametric videos[26], we produce a 'digital twin' of the cortical population that can accurately predict the response of a neuron to naturalistic visual stimulus. The digital twin makes it possible to explore a much larger stimulus space with in silico experiments than would be possible (owing to time constraints) with in vivo measurements[26]. We have extensively validated this approach by looping back in vivo and validating model predictions of the most-exciting natural images and synthetic stimuli for a neuron[27]. As part of the current study, we have validated the correspondence between model predictions and empirically observed visual response properties, including signal correlations, orientation tuning and spatial RF location. These validation results are described below. Finally, the digital twin model enabled us to separate each neuron's tuning into two components: a feature component (what the neuron responded to) and a spatial component (where the neuron's RF is located), enabling us to dissociate these two aspects of function and their relationship to connectivity.

## MICrONS functional connectomic dataset

Data were collected and processed as described in the accompanying MICrONS data release publication[14] (Fig. 1). In brief, a single mouse expressing GCaMP6s in excitatory neurons underwent 14 two-photon (2P) scans (awake and head-fixed on treadmill) of a 1,200 × 1,100 × 500 µm³ volume (anteroposterior × mediolateral × radial depth) spanning layers 2 to 6 at the conjunction of lateral V1 and AL, LM and RL (Fig. 1a). Mice rapidly acclimatized to head fixation and were able to walk, groom and adjust their posture during imaging. We monitored treadmill velocity and collected video of the pupil to track behavioural state. Neuronal responses from 115,372 functional units representing an estimated 75,909 unique excitatory neurons were collected in

response to visual stimuli composed of natural and rendered videos and parametric dynamic stimuli (Fig. 1b). A state-of-the-art deep recurrent neural network (RNN) was trained to predict neural responses to arbitrary stimuli[26] and used to characterize the in silico functional properties of imaged neurons (Fig. 1c).

After functional imaging, the tissue was processed for electron microscopy and imaged[16] at 4 × 4 × 40 nm³ resolution (Fig. 1a). The EM images were aligned[20] and automatically segmented using 3D convolutional networks into 'atomic' supervoxels, which were agglomerated to create objects (such as neurons) with corresponding 3D meshes[19,21–24], and synapses were automatically detected and assigned to presynaptic and postsynaptic partners[14,18,19,22]. The analysis presented here is restricted to the overlap of subvolume 65[14] and the 2P functional volume (Fig. 1a), an approximately 560 × 1,100 × 500 µm³ volume (in vivo dimensions) that has been both densely functionally and structurally characterized. Of 82,247 automatically extracted neuronal nuclei in this subvolume, 43,679 were both classified as excitatory and located at the intersection of the EM reconstructed volume and functional volume.

The 2P and EM volumes were approximately aligned (Fig. 1a) and 13,952 excitatory neurons were manually matched between the two volumes[14] (Fig. 1a). Retinotopically matched regions in V1 and higher visual areas AL and RL (collectively referred to as HVA) were chosen to increase the likelihood of inter-area connections, and visually responsive neurons within these regions were chosen for manual proofreading to increase the accuracy of the connectivity graph. Proofreading focused on extending axonal branches—with an emphasis on enriching projections across the V1/HVA boundary—and on removing false merges[14] (instances where other somas, glia, axons or dendrites were incorrectly merged into a neuron's reconstruction) (ref. 14 and Supplementary Table 1). Postsynaptic partners of the proofread neurons were automatically cleaned of false merges with NEURD[28]. In total, this resulted in a connectivity graph consisting of 148 functionally characterized presynaptic neurons and 4,811 functionally characterized postsynaptic partners (Fig. 1d and Extended Data Tables 1 and 2), with the presynaptic–postsynaptic numeric asymmetry resulting primarily from the labour-intensive nature of manual extension of presynaptic fine axonal projections.

## Multi-scale anatomical controls

Connectivity between neurons may be affected by numerous mechanisms, ranging from developmental processes that broadly organize neural circuits, to fine-scale plasticity mechanisms that modulate the strength of individual synaptic connections. The MICrONS volume offers the opportunity to examine function–structure relationships at both of these scales. Because it is densely reconstructed, we know not only the distance between every pair of cell bodies in the volume, but also the relative geometry of their axons and dendrites. With this information, we can determine whether two neurons experience any fine-scale axon–dendrite proximities (ADP), with axon and dendrite coming within 5 µm of each other. Furthermore, for neuron pairs with one or more ADP, we can compute the axon–dendrite co-travel distance[5] $L_d$, a pairwise measurement that captures the total extent of postsynaptic dendritic skeleton within 5 µm from any point on the presynaptic axonal skeleton.

With this metric in hand, we can define three cohorts of other neurons for functional comparisons with each presynaptic neuron (Fig. 2a–c and Extended Data Fig. 1). The first cohort are the connected postsynaptic targets of the presynaptic cell—these are neurons in the cortical region of interest that receive at least one synaptic input from the presynaptic neuron. The second group are 'ADP controls'—these are neurons with dendrites that come within 'striking range' (5 µm) of the presynaptic axon, but do not form a synaptic connection. Finally, there are 'same-region controls'—non-ADP neurons in the same cortical region

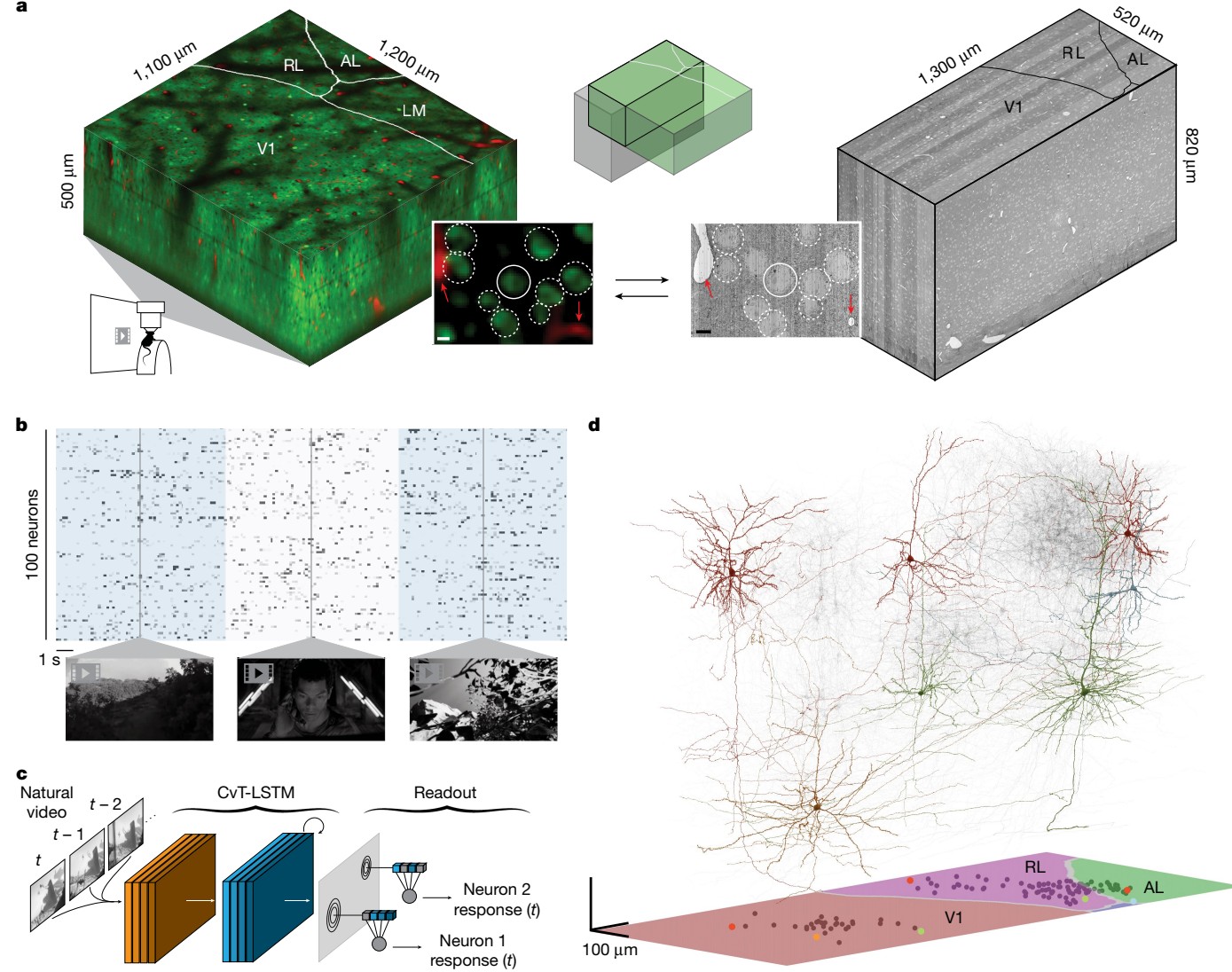

**Fig. 1 | Overview of MICrONS dataset. a**, Depiction of functionally characterized volumes (left; GCaMP6s in green, vascular label in red) and EM data (right). Middle top, the overlap of the functional 2P (green) and structural EM (grey) volumes from which somas were recruited. Middle bottom, an example of matching structural features in the 2P and EM volumes, including a soma constellation (dashed white circles) and unique local vasculature (red arrows), used to build confidence in the manually assigned 2P–EM cell match (central white circle). All MICrONS data are from a single mouse. Scale bars, 5 μm. **b**, Deconvolved calcium traces from 100 imaged neurons. Alternating blue and white column overlay represents the duration of serial video trials, with sample frames of natural videos depicted below. Parametric stimuli (not pictured) were also shown for a shorter duration than natural videos. **c**, Schematic of the digital twin deep recurrent architecture. During training, video frames (left) are input into a shared convolutional deep recurrent core (orange and blue layers), resulting in a learned representation of local spatiotemporal stimulus features. Each neuron is associated with a location (spatial component) in the visual field (grey layer) to read out feature activations (shaded blue vectors), and the dot product with the neuron-specific learned feature weights (shaded lines, feature component) results in the predicted mean neural activation for that time point. CvT-LSTM, convolutional vision transformer, long short-term memory **d**, Top, depiction of 148 manually proofread mesh reconstructions (grey), including representative samples from layer 2/3 (red), layer 4 (blue), layer 5 (green) and layer 6 (gold). Bottom, presynaptic soma locations relative to visual area boundaries.

(V1 or HVA). All connected neurons, ADP controls and same-region controls are restricted to visually responsive neurons with high-quality predictions from the digital twin (Methods).

At the axonal scale, we can explore how selective axon trajectories are within the volume, and whether neurons with axons and dendrites that meet and co-travel together have more similar tuning than nearby neurons that do not have any examples of ADPs. Selectivity at this scale could occur, for example, if a target cortical area has topographically organized functional properties such as RF location (retinotopy)[29,30] or preferred orientation[31,32], and if axons preferentially target subregions with similar functional properties. In this case, we would expect the functional properties of a presynaptic neuron and its ADP cohort to be more similar than those of random neurons selected from anywhere within the target region (same-region control).

At the synaptic scale, we can test whether there is a relationship between functional properties and connectivity beyond the axonal scale—that is, beyond what can be explained by the axonal trajectory and the spatial organization of functional properties within the volume. For this analysis, we compare the functional similarity between synaptically connected neurons, and that between unconnected ADP controls, quantifying how frequently a certain amount of axon–dendrite co-travel distance is converted to a synapse. One hypothesis is that converting proximities to synapses is independent of the functional similarity between pre- and postsynaptic neurons. In this case,

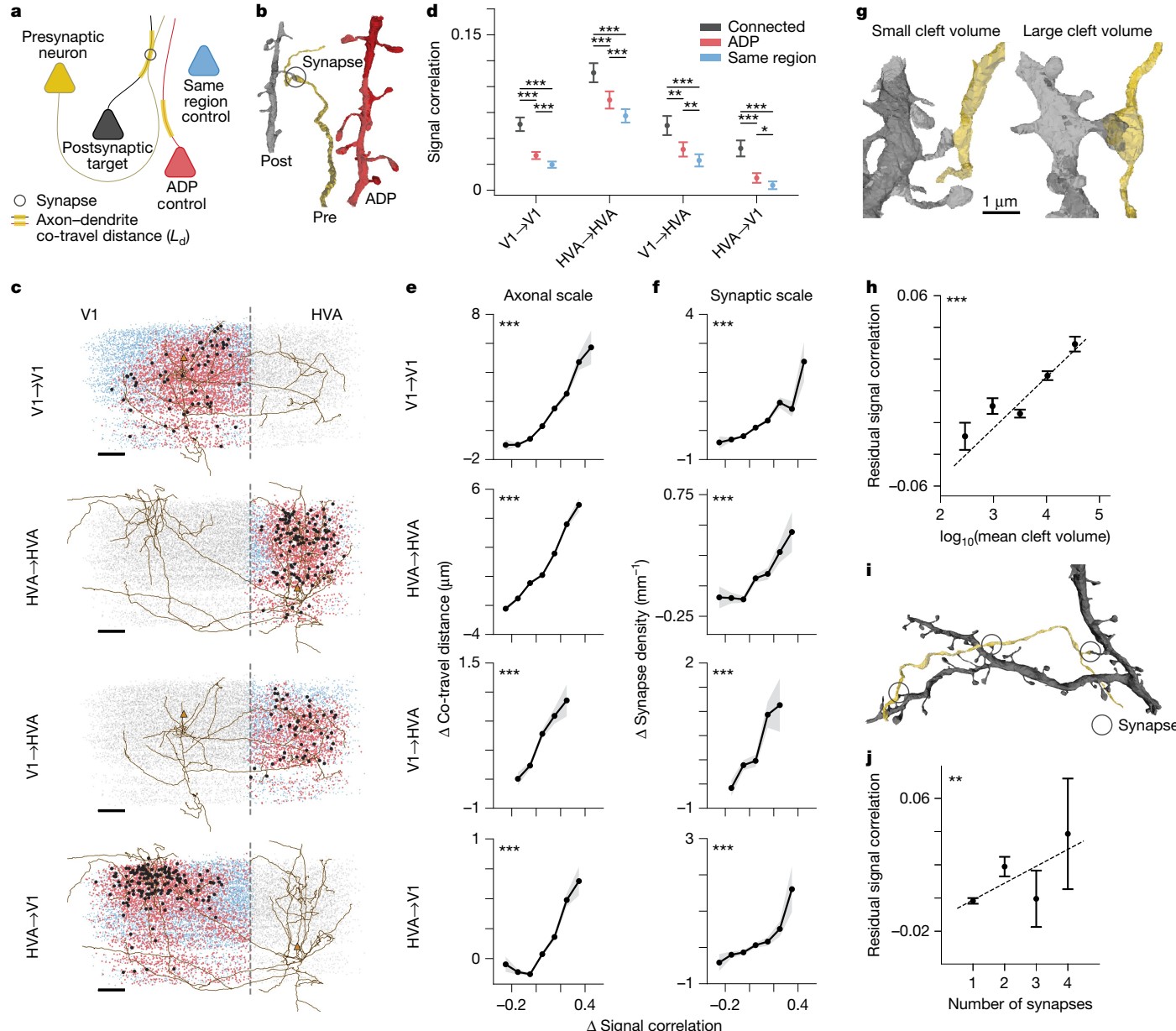

**Fig. 2 | Neurons with higher signal correlation are more likely to form synapses. a**, Anatomical control selection schematic. For each presynaptic neuron (yellow), true postsynaptic partners (black) have controls drawn from unconnected neurons with non-zero axon–dendrite co-travel distance (ADP, red) or zero co-travel distance in the same cortical region (blue). **b**, Meshes showing presynaptic (yellow axon), postsynaptic (black dendrite) and ADP control (red dendrite) neurons. **c**, Presynaptic axons in EM space for all projection types (V1→V1, HVA→HVA, V1→HVA and HVA→V1), with soma centroids of connected partners (black), ADP controls (red), same-area controls (blue) and all other functionally matched neurons (grey). Orange triangles represent presynaptic soma. Dashed line is the V1–HVA boundary. Scale bars, 100 μm. Nucleus IDs: V1, 327859; HVA, 560530. **d**, Mean signal correlation differs between synaptic partners, ADP controls and same-region controls across projection types. Data are mean ± s.e.m.; two-sided paired *t*-test. Sample size in Supplementary Table 2. **e**, $L_d$ increases with signal correlation (Δ co-travel distance and Δ signal correlation represent deviations from mean per presynaptic neuron).

V1→V1: mean $L_d$ = 9.03 μm; HVA→HVA: mean $L_d$ = 9.83 μm; V1→HVA: mean $L_d$ = 4.17 μm; HVA→V1: mean $L_d$ = 1.53 μm. Bands represent bootstrapped s.e.m. Sample sizes for GLMM statistics are shown in Supplementary Tables 3 and 4. **f**, Synapse density ($N_{syn}/L_d$) increases with signal correlation across projections. V1→V1: mean $N_{syn}/L_d$ = 1.12 mm⁻¹; HVA→HVA: mean $N_{syn}/L_d$ = 0.83 mm⁻¹; V1→HVA: mean $N_{syn}/L_d$ = 1.55 mm⁻¹; HVA→V1: mean $N_{syn}/L_d$ = 1.26 mm⁻¹. Bands represent bootstrapped s.e.m. Sample sizes for GLMM statistics are shown in Supplementary Tables 5 and 6. **g**, Meshes with small (896 voxels) and large (41,716 voxels) synapse cleft volumes. **h**, Synapse size (log₁₀ cleft volume in voxels) correlates with signal correlation (6,608 pairs, $P = 3.997 \times 10^{-21}$, linear regression on unbinned data). Residual signal correlation adjusted for $L_d$. Bars show bin-wise s.e.m. **i**, Multisynaptic meshes (yellow, presynaptic; black, postsynaptic). **j**, Signal correlations increase with synapse count (6,608 pairs, $P = 0.009$, linear regression on unbinned data). Residual signal correlation adjusted for $L_d$. Bars show bin-wise s.e.m. *$P < 0.05$, **$P < 0.01$, ***$P < 0.001$ for all figures; corrected for multiple comparisons by Benjamini–Hochberg procedure.

axon trajectories and ADPs would be sufficient to explain all of the observed connectivity between neurons (Peter's rule)[33–35]. A competing hypothesis is that synapse formation and/or stabilization depends on the functional similarity between pre- and postsynaptic neurons.

In this case, we might expect to find an additional boost in synaptic connections in similarly tuned neurons above and beyond any selectivity that already exists owing to axonal trajectories and functional inhomogeneities in the volume. The densely reconstructed MICrONS

volume offers the first opportunity to distinguish between these two hypotheses at a scale spanning layers and areas.

## Similarity across spatial scales

We tested whether neurons whose activity patterns are more correlated during visual stimulation are more likely to form synaptic connections. We quantified this using signal correlations—that is, the Pearson correlation coefficient between two neurons' responses to visual stimuli (Methods). Signal correlations provide a more general measure of functional similarity and have been shown to predict connectivity in V1 L2/3 better than orientation or direction tuning[3]. The digital twin was used to calculate the in silico signal correlation across a large battery of novel natural videos (250× 10-s clips). This approach was validated in a set of control experiments in a separate cohort of mice to ensure that the in silico signal correlation faithfully reproduced in vivo signal correlation measurements. In these control experiments, in silico signal correlations from the digital twin closely resembled the benchmark in vivo signal correlation matrix computed across a set of 30 video clips each presented 10 times, and in fact were more accurate than the in vivo signal correlation matrix computed with only 6 movie clips each presented 10 times (the number of clips available in the MICrONS data; Extended Data Fig. 2). This excellent correspondence between in vivo and in silico signal correlation estimates was achieved even though none of the in vivo clips was used during training or testing of the digital twin.

For each proofread presynaptic neuron, we computed the mean signal correlation with postsynaptic neurons, ADP controls and same-region controls (Fig. 2d). We found that mean signal correlations were higher for connected neurons than both ADP and same-region control groups, indicating that functional properties and connectivity are indeed related at the scale of individual synapses. Furthermore, signal correlations across pairs of neurons that form at least one ADP were significantly higher than across same-region controls, indicating that there is also functional specificity at the axonal scale, with axons being more likely to travel near dendrites of similarly tuned neurons. These effects were independently observed when subsets of neuron pairs were considered within V1 (V1→V1), within HVAs (HVA→HVA; including within a single HVA and between two HVAs), feedforward (V1→HVA) and feedback (HVA→V1) projection types (Fig. 2d and Supplementary Table 2).

In summary, we observed a functional like-to-like rule at the level of axonal trajectories and for connectivity at the synaptic scale.

We explored this finding further by testing for the presence of a graded relationship between the axon–dendrite co-travel distance and the corresponding boost in signal correlations (Fig. 2e).

For this analysis, to avoid confounding variability due to the size of each presynaptic neuron's axonal arbor and their varying mean signal correlations, we first computed the mean $L_d$ and mean signal correlations across all ADP targets and same-region control neurons for each presynaptic neuron. Then for each of the pairwise comparisons, we subtracted the pre-computed mean and kept only the difference from the mean for each metric. This approach has the effect of centring both the $x$ and $y$ axes in Fig. 2e,f, in order to focus on the relative effect within each presynaptic neuron and its downstream partners, removing neuron-to-neuron variability in both metrics.

Binning these differences revealed that larger-than-average $L_d$ between a presynaptic neuron and a downstream target was associated with higher-than-average signal correlation between the two neurons. This result was significant when repeated across all projection types, and indicates that the axons and dendrites of neurons with more similar functional properties are likely to meet more frequently and/or travel further together in the volume, and there is a graded relationship in this effect that is observed both within and across cortical areas.

We next performed a similar analysis for synapses, looking at connected neuron pairs. For each presynaptic neuron we first computed the mean number of synapses ($N_{syn}$) per millimetre of $L_d$, along with mean signal correlations across all pairs of synaptic and ADP targets. Then, for each of these pairwise comparisons for a single presynaptic neuron, we subtracted the mean and kept only the difference from the mean. After centring on the means for each presynaptic cell in this way, the binned differences again revealed a strong graded relationship between synaptic connectivity and functional similarity (Fig. 2f). Specifically, higher-than-average rates of synaptic density (synapses per unit $L_d$) were associated with higher-than-average functional similarity, again in a graded fashion.

Given this relationship between synapse frequency and functional similarity, we next tested whether there might be a relationship between functional similarity and either synapse size (a proxy for synaptic strength[36]) and/or the multiplicity of synaptic connections between two neurons. Indeed, previous studies have found that functionally similar presynaptic–postsynaptic pairs have stronger synaptic connections[3] and larger postsynaptic densities (PSDs)[5]. In the MICrONS dataset, segmented synapses were automatically annotated with the cleft volume, which is positively correlated to spine head volume, postsynaptic density area and synaptic strength[19,28,36] (Fig. 2g). We found that signal correlation positively correlates with cleft volume (Fig. 2h; $r = 0.032$, $P < 0.001$).

Considering the multiplicity of connections between neurons (the number of individual synapses connecting two cells), we also found that presynaptic–postsynaptic pairs with multiple synapses also had higher signal correlations (Fig. 2i,j) when compared with monosynaptic pairs. In Fig. 2h,j, the synaptic-scale effect is isolated by regressing out the contribution of $L_d$ to signal correlation. In summary, both the strength (synaptic volume) and multiplicity of connections are higher when neurons are more functionally similar, consistent with an underlying Hebbian plasticity mechanism that might act to strengthen and stabilize connections between jointly active neurons.

Finally, to ensure the robustness of these findings, we ran the same analyses described above with signal correlations measured directly from in vivo responses (rather than from the digital twin) and found that they replicated the like-to-like results achieved using the in silico signal correlations—including the graded relationships at the axonal and synaptic scale, and the relationships with synaptic cleft volume and synapse multiplicity (Extended Data Fig. 3a–e).

## Factorized in silico representation

A key advantage of the digital twin[26] (Fig. 1c) is the factorization of each modelled neuron's predicted response into two factors: readout location in visual space—a pair of azimuth–altitude coordinates; and readout feature weights—the relative contribution of the core's learned features in predicting the target neuron's activity. Intuitively, these learned features can be thought of as the basis set of stimulus features that the network then weighs to predict the neural responses. For each neuron, the combination of feature weights (what) and RF location (where) together encode everything the model has learned about that neuron's functional properties, and enable the model's predictive capacity for that neuron. This factorized representation enabled us to examine the extent to which these two aspects of neural selectivity independently contribute to the relationship between signal correlation and connectivity that we observed in Fig. 2. Feature weight similarity was measured as the cosine similarity between the vectors of presynaptic and postsynaptic feature weights. RF location similarity was measured as the visual angle difference between the centre of the model readout locations, with smaller distance between the centres (centre distance) corresponding to greater location similarity. We conducted a separate series of experiments to validate the model's readout location as an estimate of RF centre. These experiments demonstrated that

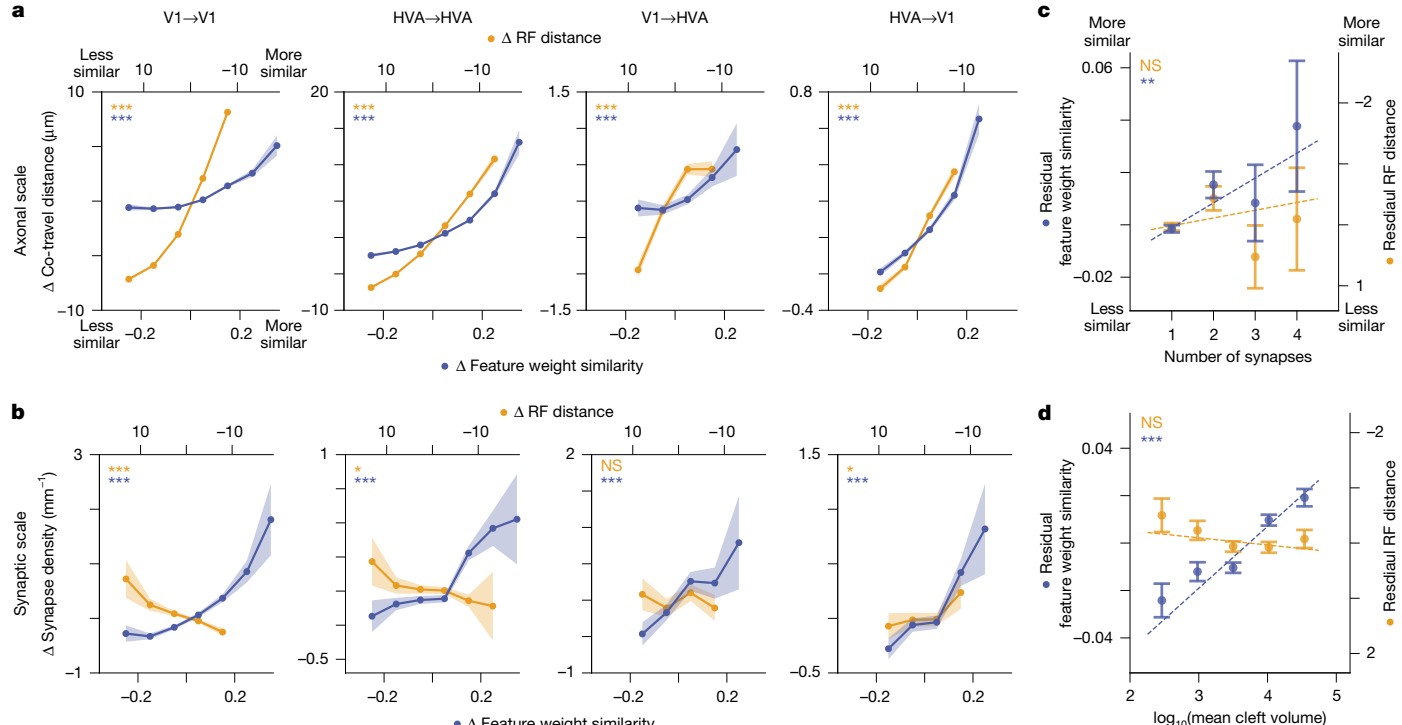

**Fig. 3 | Feature weight similarity predicts synaptic selectivity more accurately than RF centre distance. a**, Axon–dendrite co-travel distance increases with feature weight similarity and decreasing RF centre distance for within-area (V1→V1 and HVA→HVA), feedforward (V1→HVA) and feedback (HVA→V1) connectivity. Bands represent bootstrapped s.e.m. Sample sizes for GLMM statistics are shown in Supplementary Tables 7–10. **b**, Synapse density increases with feature weight similarity, but not with RF distance, except for HVA→V1 projections. Bands represent bootstrapped s.e.m. Sample sizes for GLMM statistics are shown in Supplementary Tables 11–14. **c**, Multiple synapses are associated with increasing feature similarity, but not RF centre distance for 6,608 pairs of connected neurons, after regressing out $L_d$. Error bars represent s.e.m. Feature weight similarity: $P = 0.003$; RF distance: $P = 0.358$; by linear regression. **d**, Only feature similarity (and not RF centre distance) is associated with an increase in cleft volume for 6,608 pairs of connected neurons after regressing out $L_d$. Error bars represent s.e.m. Feature weight similarity: $P = 2.391 \times 10^{-21}$; RF distance: $P = 0.451$; Benjamini–Hochberg corrected.

that the readout location correlates strongly with RF centres measured using classical sparse noise (dot-mapping) stimuli (Extended Data Fig. 4a–c). Moreover, our approach outperformed classical linear in vivo measurements of the spatial RF for the significant fraction of neurons that are not responsive to the dot-mapping stimuli, even with 1 h of dot-mapping data (Extended Data Fig. 4d–i).

## Functional similarity increases with $L_d$

Among pairs of neurons with at least one ADP, axon–dendrite co-travelling for longer-than-average distances was associated with higher-than-average feature similarity (Fig. 3a). Similarly, neurons with higher-than-average RF similarity (that is, RFs closer to each other), also co-travelled for longer-than-average distances. Thus, both feature tuning and RF location are positively correlated with the extent of ADP between pairs of neurons, and these relationships held within and across cortical areas. This result is consistent with a scenario in which axonal projections are enriched in downstream regions with similar tuning properties, either via axon guidance cues during development or via selective stabilization of axons in areas with similar functional properties, or both.

## Different rules at synaptic scale

In contrast to the functional similarity in features and RF locations associated with $L_d$, synaptic connectivity between neurons was positively correlated only with similarity in feature preferences (Fig. 3b). RF location similarity was either not correlated with synapse density or, in the case of V1, was anti-correlated. Thus, at the synaptic scale, only

like-to-like feature preference (not smaller spatial RF centre distance) is associated with increased synaptic connectivity. This is a prominent difference between axonal-scale and synaptic-scale relationships with function, and suggests that Hebbian plasticity mechanisms operating at the level of individual synapses are driven by feature similarity rather than RF centre distance. Consistent with this view, both synapse multiplicity (Fig. 3c) and synaptic cleft volume (Fig. 3d) strongly increase with feature similarity rather than RF location similarity (after regressing out $L_d$, as for Fig. 2h,j).

## Like-to-like effects across area and layers

To achieve a more detailed understanding of the organization of connections across layers and areas, for each functional similarity metric (signal correlation, feature weight similarity and RF centre distance), we also tested the relationship with connectivity across two areas (V1 and HVA) and three layers (layer 2/3 (L2/3), layer 4 (L4) and layer 5 (L5)) (Fig. 4 and Extended Data Fig. 5). For signal correlation (Fig. 4a,b and Supplementary Tables 15 and 16; in vivo analysis in Extended Data Fig. 3f,g) and feature weight similarity (Fig. 4c,d and Supplementary Tables 17 and 18), like-to-like effects (red squares) were widespread across many area and layer combinations, at both the axonal and the synaptic scale.

In the case of RF centre distance, whereas like-to-like effects (red squares) were widespread at the axonal scale, these effects disappeared when considering synaptic-scale specificity. This finding is consistent with the view that there is no further selectivity for retinotopic overlap gained at the synapse level, compared with that obtained at the axon trajectory level (Fig. 4e,f and Supplementary Tables 19 and 20). In this

analysis, individual presynaptic baselines (such as variable $L_d$, synapse rate or signal correlation) were accounted for with a generalized linear mixed model (GLMM) (Methods). Distributions of all pairwise functional measurements, including in vivo signal correlation, in silico signal correlation, feature weight similarity and RF distance, are provided in Extended Data Fig. 6. Varying the inclusion thresholds of the above analyses across varying levels of digital twin model performance (quartiles of neurons ranked by prediction accuracy) did not substantially change the main results (Extended Data Fig. 7).

## Like-to-like orientation tuning in V1

Many neurons in mouse V1 and higher visual areas are strongly tuned for orientation, and a number of previous functional connectivity studies have used differences in preferred orientation as a metric for visual similarity within V1. To compare our findings more directly with this previous work, we repeated the central analysis in Figs. 3 and 4, but using only the difference in preferred orientation—rather than signal correlations—to determine functional similarity.

We used the digital twin to estimate orientation tuning, and we validated this approach with in vivo validation experiments (Extended Data Fig. 8a,b), where we compared the in silico orientation tuning curve with the tuning curve estimated from the in vivo data. Orientation-selective responses were driven by lowpass filtered noise with coherent orientation and motion, a stimulus that we have previously used to drive strong visual responses in orientation-tuned cells[26,31]. For orientation-tuned neurons (global orientation selectivity index (gOSI) >0.25, corresponding to more that 50% of co-registered neurons; Methods), the in silico orientation tuning curves align closely with in vivo orientation tuning curves (Extended Data Fig. 8c–f).

We found that connected neurons in V1 have more similar orientation tuning than unconnected controls (Extended Data Fig. 9), as reported by previous studies[1,5,7]. However, in contrast to previous studies, we did not observe a similar significant like-to-like effect when restricting the analysis specifically to projections within V1 L2/3 excitatory neurons. To understand this deviation from previous studies, we first determined that connected neuron pairs within V1 L2/3 projections in the MICrONS dataset did indeed have similar orientation preferences (Extended Data Fig. 10), as expected. However, unconnected pairs showed the same level of similarity in orientation preference. We believe that this is the result of a local orientation bias where the MICrONS volume is located in V1[31].

Overall, we found that the model feature weight similarity is a better predictor of connectivity than classical orientation preference, even for neurons that are tuned to oriented stimuli (Extended Data Fig. 11). Recent work has emphasized that optimal stimuli for neurons in mouse V1 can exhibit complex spatial features that deviate markedly from Gabor-like stimuli[27,37]. These results highlight the advantages of studying more complete tuning functions, such as the model feature weights that we focus on here, rather than single tuning parameters such as orientation preference.

Finally, we repeated the analyses for similarity in orientation tuning, readout location and feature weight with respect to 3D cortical distance between presynaptic and postsynaptic soma centres in EM coordinates (Extended Data Fig. 12) and with respect to summed synapse cleft volume per millimetre of co-travel distance (synapse size density, Extended Data Fig. 13), and in both cases observed similar overall trends.

## Neurons with common input are similar

If the pairwise like-to-like rule were the sole organizing principle of the visual cortex—implying that all postsynaptic neurons closely resemble their presynaptic partners—we would expect postsynaptic neurons to exhibit a certain degree of similarity to one another.

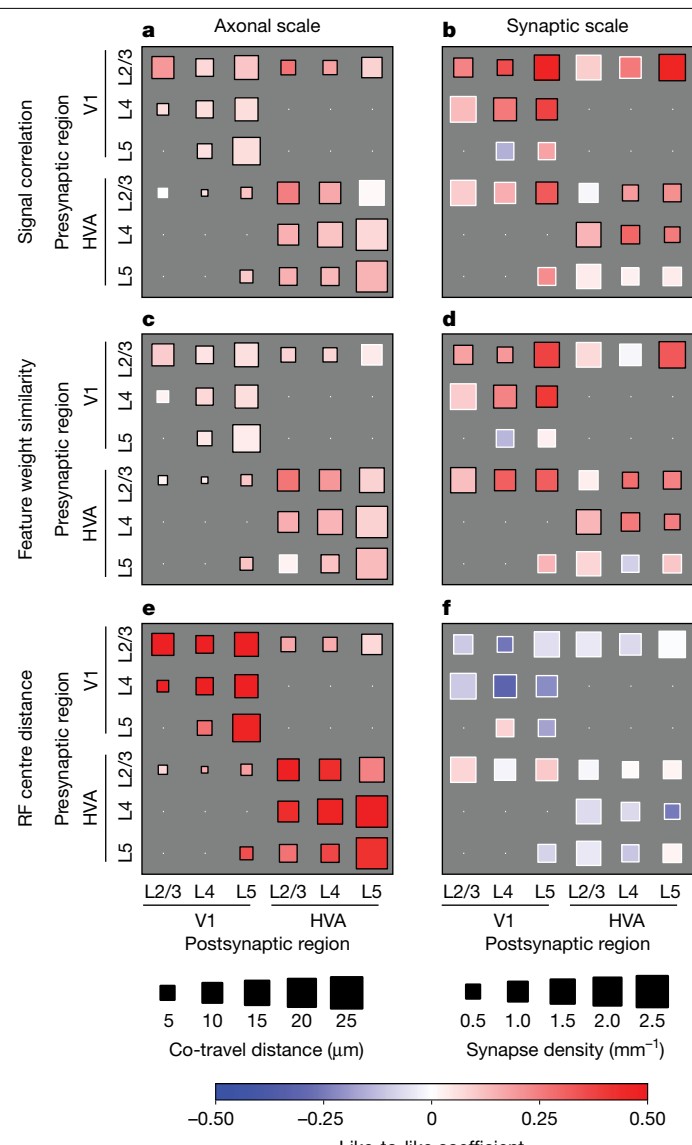

**Fig. 4 | Like-to-like effects are widespread but vary across brain areas, cortical layers and tuning similarity metrics. a–f**, Degree of like-to-like broken down by area and layer membership measured at axonal (**a,c,e**) and synaptic (**b,d,f**) scales. On the colour bar, red indicates more like-to-like and blue indicates less. Like-to-like coefficients are the coefficients of GLMMs fitted to predict axon–dendrite co-travel distance or synapse density with the corresponding functional similarity. Black borders indicate significance ($P < 0.05$); white border indicates $P > 0.05$; by GLMM fit after Benjamini–Hochberg correction for multiple comparisons. Details of statistical tests are presented in Supplementary Tables 15–20.

However, neural feature selectivity is likely to arise from more complex connectivity rules, so a cohort of neurons downstream of a single presynaptic neuron might, on average, be less (Fig. 5a, left) or more (Fig. 5a, right) functionally similar to each other. To evaluate whether the similarity among postsynaptic neurons differs from the prediction of the like-to-like rule, we built a simple model network, and introduced the empirical relationships between presynaptic–postsynaptic functional similarity and connectivity that we observed in our data. Specifically, we replicated the empirical distribution of signal correlations, feature weight similarities and RF location distances over all model neuron pairs, and then predicted the expected number of synapses between neuron pairs—on the basis of their functional similarity—with a Poisson linear mixed-effects model (Fig. 5b). We confirmed

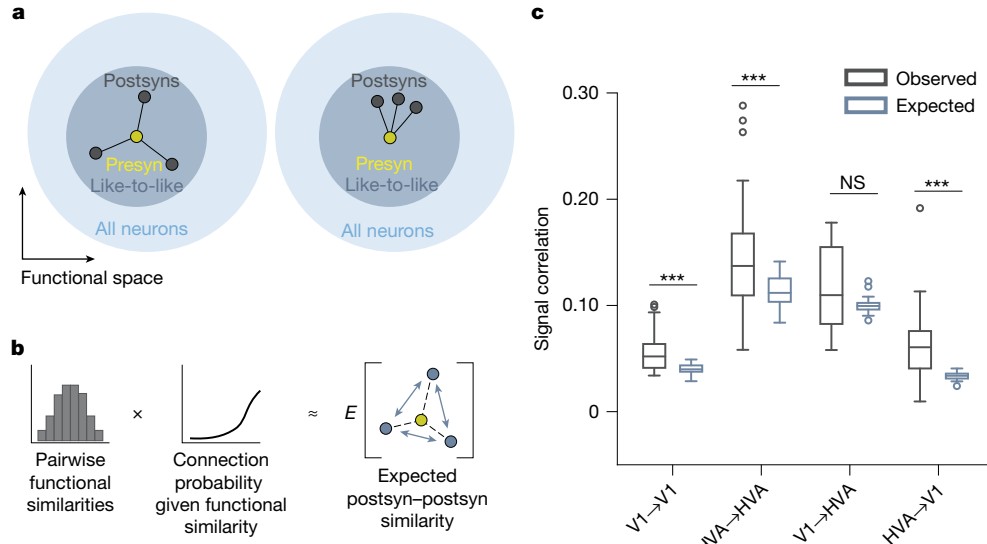

**Fig. 5 | Postsynaptic neurons with a common input are more functionally similar to each other than expected from a pairwise like-to-like rule.**
**a**, Left, schematic illustrating the null hypothesis that postsynaptic neurons (postsyns, dark grey circles) of a common presynaptic neuron (presyn, yellow circle) have no additional feature similarity with each other beyond their like-to-like similarity with their common presynaptic neuron. Neurons are shown embedded in functional space, where units with similar functional properties are closer together. In this scenario, postsynaptic neurons are distributed uniformly around the presyn in the like-to-like region of functional space (large grey circle). Right, schematic illustrating the alternative hypothesis that the postsynaptic neurons are closer in functional space than predicted from a pairwise like-to-like rule, equivalent to being clustered non-uniformly within the like-to-like region. **b**, Schematic illustrating the functional connectivity model used to simulate the null hypothesis in **a**. Left, pairwise functional measurements—including signal correlations, feature weight similarity and RF location distance—were passed through a function relating functional similarity to connection probability. Right, then, within this modelled network, we computed the pairwise similarity of all postsynaptic neurons downstream of a common presynaptic neuron. **c**, Observed mean pairwise signal correlations between postsynaptic neurons compared with those expected from model predictions across projection types ($n = 52$ presynaptic neurons for HVA→HVA, $n = 38$ for HVA→V1, $n = 17$ for V1→HVA, $n = 35$ for V1→V1). Box plots show median, interquartile range (box) and 1.5× interquartile range (whiskers); points indicate outliers. Three of four projection types showed significantly higher similarities than predicted. Two-sided Wilcoxon signed-rank test; exact $P$ values are presented in Supplementary Table 21.

that this model replicated the expected functional similarity between connected neurons, indicating that it accurately captured the same pairwise like-to-like rule that we observed in the data (Extended Data Fig. 14). Then, we measured the similarity among all postsynaptic neurons downstream of a single presynaptic neuron by calculating the mean pairwise signal correlations. We found that postsynaptic neurons that received common synaptic inputs in the MICrONS dataset were even more similar than the like-to-like model predicted (Fig. 5c; in vivo analysis in Extended Data Fig. 3h). These relationships held when tested at both axonal and synaptic scales for three out of the four projection types (Extended Data Fig. 14). This suggests the existence of higher-order functional organization beyond the simple pairwise relationships that we focused on up to this point.

## Like-to-like connectivity in RNNs

A possible functional role for the like-to-like connectivity that we observed in our data is suggested by theoretical work on RNN models, starting with early work on attractor-based models such as Hopfield networks[38,39]. In classical Hopfield networks, connectivity after learning is proportional to functional covariance, so like-to-like connectivity emerges by design. To test the generality of this phenomenon, we considered a model that does not by definition exhibit like-to-like connectivity after learning. Specifically, we trained a vanilla RNN model on a simple image classification task (Fig. 6a and Methods).

The trained RNN showed increased like-to-like connectivity compared with the same model before training (Fig. 6b,c), and a small shift in the distributions of signal correlations, similar to those in our data (Extended Data Figs. 6 and 15). Ablating like-to-like connections in the trained model decreased performance more than ablating random connections with the same connection strengths (Fig. 6d), suggesting that like-to-like connectivity has a functional role. Finally, the trained model exhibits an increase in signal correlations within cohorts of postsynaptic cells defined by a shared presynaptic neuron, similar to the higher-order connectivity rule observed in our data (Fig. 6e).

## Discussion

Understanding neural computations at the level of circuit-level mechanisms requires that we identify the principles that relate structure to function in the brain. Here we use the MICrONS multi-area dataset to study the relationship between the connections and functional responses of excitatory neurons in mouse visual cortex across cortical layers and visual areas. Our findings reveal that neurons with highly correlated responses to natural videos (that is, high signal correlations) tend to be connected with each other, not only within the same cortical areas but also across multiple layers and visual areas, including feedforward and feedback connections. Although the overall principle of like-to-like connectivity that we describe here is consistent with a number of previous studies[1,2,4,5,7], our work leverages three unique strengths of the MICrONS dataset to extend and refine these previous findings.

First, the scale of the volume enables us to explore connection principles across layers 2–5 of cortex, not just within V1, but also between V1 and higher visual areas. In agreement with previous findings from V1 L2/3, we find that pairs of cells with higher signal correlations are more likely to be connected[1–3]. This general principle holds not only in V1 L2/3, but also in higher visual areas and for inter-area feedforward and feedback projections.

Second, we are able to take advantage of the dense reconstruction to ask questions about functional specificity at the axonal scale that

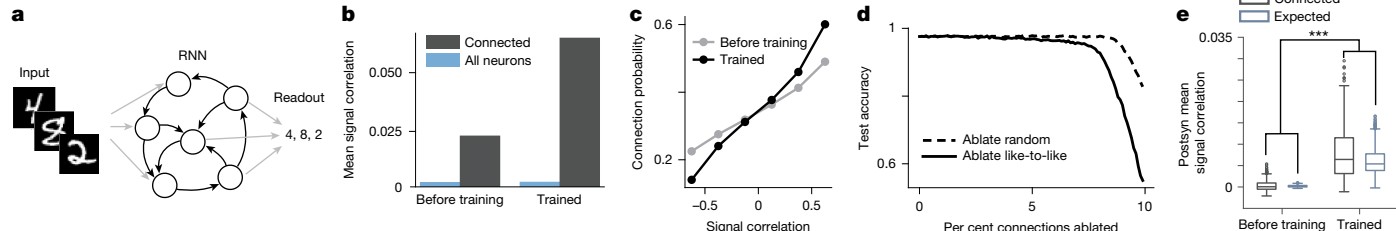

**Fig. 6 | Like-to-like connectivity in an RNN. a**, A vanilla RNN with 1,000 hidden units was provided images as inputs and weights were trained so that a readout of the final state identified the input's label. **b**, Mean signal correlations among all and connected neuron pairs for the same RNN before and after training. Neurons were classified as connected when their weights exceeded a fixed threshold. **c**, Connection probability as a function of signal correlation for the same network before and after training. **d**, Test accuracy of the network as a function of the number of connections that were ablated when ablating random or like-to-like connections. Connections were classified as like-to-like whenever the weight and signal correlation both exceeded a fixed threshold. **e**, Mean postsynaptic–postsynaptic signal correlations and the expected postsynaptic–postsynaptic signal correlation given a pairwise model similar to Fig. 5c before and after training (two-sided sign test: $P = 9.85 \times 10^{-5}$). The postsynaptic–postsynaptic signal correlations are computed for all 1,000 hidden units in the RNN. Box plots show median, interquartile range (box) and 1.5× interquartile range (whiskers); points indicate outliers.

would be difficult to address with other data. We find that axons are more likely to co-travel with dendrites of similarly tuned neurons, even for long-range axons that span areas. The dense reconstruction also enables us to compute a set of null distributions for the expected synaptic connectivity between neurons based on ADPs. These controls enable us to distinguish whether the relationships that we observe between connectivity and function are due to the overall geometry of axonal and dendritic arbors in the volume, or whether they reflect a more precise connectivity rule at the level of individual synapses. For example, it is only with the inclusion of both same-region and ADP controls that we are able to observe the diverging findings of axon trajectory level selectivity for RF centre distance and synaptic level selectivity for feature weight similarity (Fig. 3a,b). These different controls can be mapped onto potential developmental or adult plasticity mechanisms that may shape the coarse axon trajectory and fine-scale synaptic connectivity across the brain.

Finally, our deep learning neural predictive modelling approach enables us to comprehensively characterize the tuning function of a neuron, factorize it into spatial and feature tuning components, and facilitates in silico exploration with neural responses to novel visual stimuli. The digital twin model enables us to extract signal correlations over a much larger set of naturalistic videos, resulting in better connectivity predictions compared with in vivo measurements from a smaller stimulus set (Extended Data Fig. 11). Moreover, the model's factorized architecture provides a unique opportunity to discover distinct synaptic organizing principles for two interpretable components of neuronal tuning: what the neuron is tuned to and where its RFs are located. Notably, the digital twin model demonstrates excellent out-of-training-set performance (Extended Data Fig. 2) even for novel stimulus domains (Extended Data Fig. 4). This generalization ability presents exciting possibilities for future in silico visual experiments, although independent validation experiments remain essential when studying the digital twin model with new stimulus domains. Currently, we treat this model as a black box, but future models could constrain the architecture to make internal model parameters more interpretable. Additionally, recent studies have emphasized the explanatory power of behavioural states and task variables[40,41]. Future digital twins could incorporate additional behavioural measurements that make it possible to study more general relationships between structure and function, beyond visual processing.

It is important to acknowledge limitations of the dataset that should be considered when interpreting our results. While the morphological reconstructions are the largest of their kind, it should not be assumed that they fully capture the axonal and dendritic arbor, due both to proofreading constraints or because of truncation at volume boundaries. Where there are incomplete reconstructions, we may not observe

synapses between pairs of cells that are actually connected. As a result, pairs that are actually connected would be included in the set of controls, making our measured like-to-like effect a conservative estimate.

In addition to proofreading, our results depend on the accuracy of automated synapse detection. In general, it is more difficult to detect smaller synapses, so there exists a possibility that our results are biased to larger synapses. However, manual validation of synapse detection has been performed previously and indicates that the algorithm performs well, with a precision of 96%, recall of 89%, and a partner assignment accuracy of 98%[14].

Although the nonlinearity of the digital twin allows for excellent predictive performance and generalizability, in some cases the properties fit by the model can deviate from properties obtained with classical methods. For example, model readout locations tend to be shifted towards the centre of the monitor compared with traditional in vivo RF centres estimated via spike-triggered average (STA) (Extended Data Fig. 4b). This difference could be attributed either to a potential inductive bias in the model's architecture or to a more realistic representation of RF centres, unconstrained by the linear assumption that underlies STA estimation. In any case, in silico STA centres align closely with in vivo STA centres (Extended Data Fig. 4c), and our key findings remain consistent when using model readout locations or in silico STA centres (Extended Data Fig. 4j,k). Importantly, our extensive validation experiments demonstrate the model's overall effectiveness: it generates more coherent retinotopic maps than classical methods (Extended Data Fig. 4d,f,h) even for neurons where reliable in vivo estimates are unavailable (Extended Data Fig. 4e,g,i), shows high correspondence with in vivo measurements of signal correlations and orientation tuning (Extended Data Figs. 2 and 8), and can even outperform direct in vivo measurements when data are limited (Extended Data Fig. 2). Additionally, the in silico functional similarities outperform classical in vivo measurements predicting anatomical connectivity (Extended Data Fig. 11). These results highlight the strengths as well as the caveats of using deep learning models to characterize neural responses: whereas they can be extremely accurate at capturing subtle aspects of neural function, care should be taken in interpreting their internal representations.

In theoretical models of neural circuit function, like-to-like connectivity is a recurring theme, including Hebb's theory of neural assemblies[10], Hubel and Wiesel's theory of RF formation[11], and later work by Hopfield[38] and others[42] on attractor-based models. Like-to-like connectivity is often assumed a priori or emerges owing to Hebbian plasticity in these models, but our analysis of a vanilla RNN trained by gradient descent shows that like-to-like connectivity not only arises naturally from optimizing a recurrent system for a simple visual task, but also has important functional consequences for task performance (Fig. 6). As in

Hopfield networks, like-to-like connectivity in our vanilla RNN pushes neural activity towards similar states in response to similar stimuli. Unlike Hopfield networks, this was achieved with a weaker like-to-like effect—similar to the magnitude that we observed in the MICrONS data (Extended Data Figs. 6 and 15). More generally, we expect like-to-like connectivity to emerge in networks for which increased connectivity promotes stronger functional covariance and for which similar responses to similar stimuli are promoted by learning.

In addition, there is still a question of whether there exist higher-order functional motifs beyond simple, pairwise relationships. We explore one such higher-order pattern in our analysis of functional similarity among postsynaptic neurons that share at least one common input in both the biological data (Fig. 5) and the RNN (Fig. 6). We observe functionally similar postsynaptic cohorts, suggesting the presence of more complex organizational principles than a pairwise rule. Previous studies of presynaptic cells that converge on a single common postsynaptic neuron in V1[4,7,25] also suggest that like-to-like connectivity may only partially capture more complicated principles relating structure and function. For example, Wertz et al.[4] found that the similarity of inputs differed depending on layer origin. Several studies also point to an interplay between the geometric relationship of RF positions and feature preferences[6,7,43,44]. For example, Rossi et al.[7] found that the spatial offset between the RFs of excitatory and inhibitory inputs matched the direction selectivity of the postsynaptic cell[7].

As proofreading and annotation in the MICrONS volume continues to yield a more complete and richly coloured graph, it will become possible to relate connectivity motifs of higher orders, with more complex functional properties, and take into account additional features such as morphological or ultrastructural details. Although the incredible accuracy of machine learning-based reconstruction methods has rightly increased optimism about the potential discoveries that can be made from large EM volumes—especially when combined with functional characterization—the magnitude of the challenge contained in even a 1 mm$^3$ volume of cortex in a single mouse should not be forgotten. The analyses in this Article are focused on only a small number of manually proofread neurons, but even this small subset of neurons contains more than 1.5 m of axonal and dendritic reconstruction. Ongoing investments in proofreading, matching and extension efforts within this volume will have exponential returns for future analyses as they yield a more complete functional connectomic graph and reduce or eliminate potential biases in the connections. As more large-scale datasets like MICrONS are publicly released, there will be much more to discover about the organizing principles that relate structure and function in other brain areas[45] and even other model organisms[46]. Our hope is that this dataset, including both the structural anatomy and the immortalized digital twin for future in silico experiments, will be a community resource that will yield concrete insights as well as inspiration about the scale of investigation that is now possible in neuroscience.

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

[1]Center for Neuroscience and Artificial Intelligence and Department of Neuroscience, Baylor College of Medicine, Houston, TX, USA. [2]Department of Ophthalmology and Byers Eye Institute, Stanford University School of Medicine, Stanford, CA, USA. [3]Stanford Bio-X, Stanford University, Stanford, CA, USA. [4]Wu Tsai Neurosciences Institute, Stanford University, Stanford, CA, USA. [5]Department of Electrical and Computer Engineering, Rice University, Houston, TX, USA. [6]Department of Mathematics, Creighton University, Omaha, NE, USA. [7]Salk Institute for Biological Studies, La Jolla, CA, United States. [8]Princeton Neuroscience Institute, Princeton University, Princeton, NJ, USA. [9]Electrical and Computer Engineering Department, Princeton University, Princeton, NJ, USA. [10]Allen Institute for Brain Science, Seattle, WA, USA. [11]Computer Science Department, Princeton University, Princeton, NJ, USA. [12]Brain and Cognitive Sciences Department, Massachusetts Institute of Technology, Cambridge, MA, USA. [13]DataJoint, Houston, TX, USA. [14]Department of Basic Sciences, Faculty of Medicine, University of Crete, Heraklion, Greece. [15]Institute of Molecular Biology and Biotechnology, Foundation for Research and Technology Hellas, Heraklion, Greece. [16]Institute for Bioinformatics and Medical Informatics, University Tübingen, Tübingen, Germany. [17]Institute for Computer Science and Campus Institute Data Science, University Göttingen, Göttingen, Germany. [18]Departments of Mathematics, Biology and Biochemistry, University of Houston, Houston, TX, USA. [19]Departments of Applied and Computational Mathematics and Statistics and Biological Sciences, University of Notre Dame, Notre Dame, IN, USA. [20]Department of Neurobiology and Biophysics, University of Washington, Seattle, WA, USA. [21]Computational Neuroscience Center, University of Washington, Seattle, WA, USA. [22]Department of Computer Science, Rice University, Houston, TX, USA. [23]Neuroscience Institute, Carnegie Mellon University, Pittsburgh, PA, USA. [23]Department of Machine Learning, Carnegie Mellon University, Pittsburgh, PA, USA. [25]Department of Electrical Engineering, Stanford University, Stanford, CA, USA. [26]These authors contributed equally: Zhuokun Ding, Paul G. Fahey, Stelios Papadopoulos. ✉e-mail: reimer@bcm.edu; tolias@stanford.edu

## Methods

### The MICrONS dataset

The MICrONS dataset was collected in a single mouse as described in ref. 14, including neurophysiological data collection, visual stimulation, stimulus composition, EM data collection, automatic EM segmentation and reconstruction, EM synapse detection, manual EM proofreading, volume coregistration and manual soma–soma matching between the functional and EM volumes. The sections 'Neurophysiological experiments', 'Visual stimulation' and 'Validation of the digital twin model' below are specific to additional experiments described in Extended Data Figs. 2, 4 and 8.

### Neurophysiological experiments

All procedures were approved by the Institutional Animal Care and Use Committee of Baylor College of Medicine. Animals were housed in a room with 20–22 °C, 30–70% humidity, 12 h light:12 h dark cycle room, with experiments performed during the subjective night. Ten mice (*Mus musculus*, 3 female, 7 males, 78–190 days old at first experimental scan) expressing GCaMP6s in excitatory neurons via *Slc17a7*-Cre and Ai162 transgenic lines (provided by H. Zeng; JAX stock 023527 and 031562, respectively) were anaesthetized and a 4-mm craniotomy was made over the visual cortex of the right hemisphere as described previously[47,48].

Mice were head-mounted above a cylindrical treadmill and calcium imaging was performed with a mesoscope[15] as described[14], with surface power not exceeding 20 mW, depth constant of 220 μm, and greatest laser power of ~86 mW used at approximately 400 μm from the surface.

The cranial window was levelled with regard to the objective with six degrees of freedom. Pixel-wise responses from a region of interest spanning the cortical window (3,600 × 4,000 μm, 0.2 pixels per μm, 200 μm from surface, 2.5 Hz) to drifting bar stimuli were used to generate a sign map for delineating visual areas[30].

For the validation data in Extended Data Figs. 2, 4 and 8, our target imaging site was a 1,200 × 1,100 μm² area spanning L2–L5 at the conjunction of lateral V1 and three lateral higher visual areas: AL, LM and RL. This resulted in an imaging volume that was roughly 50% V1 and 50% HVA. This target was chosen in order to mimic the area membership and functional property distribution in the MICrONS animal. Each scan was performed at 6.3 Hz, collecting eight 620 × 1,100 μm² fields per frame at 0.4 pixels per μm *xy* resolution to tile a 1,190–1,200 × 1,100 μm² field of view at 4 depths (2 planes per depth, 40–50 μm overlap between coplanar fields). The 4 imaging planes were distributed across layers with at least 50 μm spacing, with two planes in L2/3 (depths: 180 μm, 230 μm), one in L4 (325 μm), and one in L5 (400 μm).

Video of the animal's eye and face was captured throughout the experiment. A hot mirror (Thorlabs FM02) positioned between the animal's left eye and the stimulus monitor was used to reflect an IR image onto a camera (Genie Nano C1920M, Teledyne Dalsa) without obscuring the visual stimulus. The position of the mirror and camera were manually calibrated per session and focused on the pupil. Field of view was manually cropped for each session. The field of view contained the left eye in its entirety, 212–330 pixels height × 262–424 pixels width at 20 Hz. Frame times were time-stamped in the behavioural clock for alignment to the stimulus and scan frame times. Video was compressed using Labview's MJPEG codec with quality constant of 600 and stored the frames in AVI file.

Light diffusing from the laser during scanning through the pupil was used to capture pupil diameter and eye movements. A DeepLabCut model[49] was trained on 17 manually labelled samples from 11 animals to label each frame of the compressed eye video (intraframe only H.264 compression, CRF:17) with 8 eyelid points and 8 pupil points at cardinal and intercardinal positions. Pupil points with likelihood >0.9 (all 8 in 69.8–99.2% of frames per scan) were fit with the smallest enclosing circle, and the radius and centre of this circle was extracted. Frames with <3 pupil points with likelihood >0.9 (<1.1% frames per scan), or producing a circle fit with outlier >5.5 × s.d. from the mean in any of the three parameters (centre *x*, centre *y*, radius, <0.1% frames per scan) were discarded (total <1.2% frames per scan). Gaps were filled with linear interpolation.

The mouse was head-restrained during imaging but could walk on a treadmill. Rostro-caudal treadmill movement was measured using a rotary optical encoder (Accu-Coder 15T-01SF-2000NV1ROC-F03-S1) with a resolution of 8,000 pulses per revolution, and was recorded at ~100 Hz in order to extract locomotion velocity.

### Visual stimulation

For the validation data in Extended Data Figs. 2, 4 and 8, monitor size and positioning relative to the mouse were as described[14], with the exception of replacing the dot stimulus for monitor positioning with 10 × 10 grid tiling a central square (approximately 90° width and height) with 10 repetitions of 200 ms presentation at each location.

A photodiode (TAOS TSL253) was sealed to the top left corner of the monitor, and the voltage was recorded at 10 kHz and time-stamped with a 10 MHz behaviour clock. Simultaneous measurement with a luminance meter (LS-100 Konica Minolta) perpendicular to and targeting the centre of the monitor was used to generate a lookup table for linear interpolation between photodiode voltage and monitor luminance in cd m⁻² for 16 equidistant values from 0–255, and 1 baseline value with the monitor unpowered.

At the beginning of each experimental session, we collected photodiode voltage for 52 full-screen pixel values from 0 to 255 for 1-s trials. The mean photodiode voltage for each trial was collected with an 800-ms boxcar window with 200-ms offset. The voltage was converted to luminance using previously measured relationship between photodiode voltage and luminance and the resulting luminance versus voltage curve was fit with the function $L = B + A \times P^{\gamma}$, where $L$ is the measured luminance for pixel value $P$, and the $\gamma$ of the monitor was fit as 1.73. All stimuli were shown without linearizing the monitor (that is, with monitor in normal gamma mode).

During the stimulus presentation, display frame sequence information was encoded in a three-level signal, derived from the photodiode, according to the binary encoding of the display frame (flip) number assigned in order. This signal underwent a sine convolution, allowing for local peak detection to recover the binary signal together with its behavioural time stamps. The encoded binary signal was reconstructed for >93% of the flips. Each flip was time-stamped by a stimulus clock (MasterClock PCIe-OSC-HSO-2 card). A linear fit was applied to the flip time stamps in the behavioural and stimulus clocks, and the parameters of that fit were used to align stimulus display frames with scanner and camera frames. The mean photodiode voltage of the sequence encoding signal at pixel values 0 and 255 was used to estimate the luminance range of the monitor during the stimulus, with minimum values of approximately 0.003–0.60 cd m⁻² and maximum values of approximately 8.68–10.28 cd m⁻².

### Preprocessing of neural responses and behavioural data

Fluorescence traces from the MICrONS dataset and the additional data for Extended Data Figs. 2, 4 and 8 were detrended, deconvolved and aligned to stimulus and behaviour as described[26], and all traces were resampled at 29.967 Hz. Possible redundant traces, where a single neuron produced segmented masks in multiple imaging fields, were all kept for downstream model training. We elected to remove one of the 14 released scans from the analysis (session 7, scan_idx 4) due to compromised optics (water ran out from under the objective for ~20 min), leaving 13 scans.

### Model architecture and training of the digital twin model

The model architecture and training for the digital twin model used for assessing in silico signal correlation, feature weight similarity, and RF centre distance is the same as the CvT-LSTM model described in ref. 26.

In brief, the core network of the CvT-LSTM models was trained on eight scans collected from eight mice with natural video stimuli to capture cortical representations of visual stimuli shared across mice. The parameters of the core network are then frozen, and the rest of the network parameters are trained for each scan with trials where natural videos are shown in the MICrONS dataset. Trials were excluded from model training if more than 25% of their pupil frames were untrackable. This issue most commonly arose when the animal closed its eye, rendering the functional relationship between neural activity and the visible stimulus ambiguous. The number of excluded trials varied across scans, ranging from 2 to 123 per scan, representing 0.6–38.0% of total trials.

To assess orientation tuning similarity (Extended Data Figs. 8, 9, 10 and 13), the parameters of the core of the CvT-LSTM model above were frozen, and the rest of the network parameters are fine-tuned with both natural videos and oriented noise stimuli available from the MICrONS dataset to improve alignment between in vivo and in silico orientation tuning.

### Functional unit inclusion criteria

In order to focus our analyses on neurons that are visually responsive and well modelled by the digital twin, we applied a dual functional threshold over two metrics (in vivo reliability and model prediction performance) prior to all analyses related to signal correlation, RF centre distance and feature weight similarity.

**In vivo reliability threshold.** In order to estimate the reliability of neuronal responses to visual stimuli, we computed the upper bound of correlation coefficients for each neuron ($CC_{max}$; Schoppe et al.[50]) across 60 s of natural video stimuli repeated 10 times across the stimulus period (10 min total). $CC_{max}$ was computed as:

$$CC_{max} = \sqrt{\frac{N\mathrm{Var}(\overline{y}) - \overline{\mathrm{Var}(y)}}{(N-1)\mathrm{Var}(\overline{y})}},$$

where $y$ is the in vivo responses, and $N$ is the number of trials. A threshold of $CC_{max} > 0.4$ was applied. Where more than one 2P functional unit was matched to a given EM unit, the functional trace with the higher oracle score was used for analysis.

**Model prediction performance threshold.** In order to focus our analyses on neurons for which adequate model performance indicated sufficiently accurate representation of the neuronal tuning features, we computed the test correlation coefficient on the withheld oracle test dataset, which was not part of the training set. Test correlation coefficients ($CC_{abs}$) were computed as:

$$CC_{abs} = \frac{\mathrm{Cov}(\overline{x}, \overline{y})}{\sqrt{\mathrm{Var}(\overline{x})\mathrm{Var}(\overline{y})}},$$

where $x$ is the in silico response and $y$ is the in vivo response. A threshold of $CC_{abs} > 0.2$ was applied.

144 out of 148 presynaptic neurons and 3,920 out of 4,811 postsynaptic neurons passed the dual functional unit inclusion criteria.

**Oracle score.** The oracle score was computed for all units as described in the accompanying Article[14]. Oracle score is later used to select presynaptic neurons for morphological proofreading.

### 2P–EM matching

The matching between 2P functional units and EM cells aligns closely with table coregistration_manual_v4[14] with some additional restrictions applied. First, the matches to the excluded scan described in 'Preprocessing of neural responses and behavioural data' were removed. Then, two thresholds were applied directly to the table (residual <20 and score >−10).

### Morphological proofreading

Whereas automation of the EM segmentation has progressed to where dense reconstruction is possible at the millimetre scale, even state-of-the-art methods still leave imperfections in the graph relative to human expert performance. The two categories of reconstruction error are false merges (the incorrect grouping of segmented objects, such as including an axon or dendrite that does not belong to a specific soma) and false splits (the incorrect separation of objects, such as excluding an axon or dendrite that does belong to a specific soma). These errors lead to incorrect associations between pre- and post-synaptic partners and ultimately an incorrect connectivity graph. Proofreading corrects false merges by 'cleaning' the reconstruction (removing incorrectly associated segments), and corrects false splits by 'extending' the reconstruction (adding back missing segments). We used two proofreading approaches in this study: manual and automatic. 'Manual proofreaders' were trained to both clean and extend reconstructions to a high degree of accuracy, as validated by expert neuroanatomists. All of the presynaptic cells in this study were manually proofread. The manual proofreading protocol can be found in the primary dataset Article[14]. For the rest of the cells (postsynaptic and control neurons), we used the NEURD package[28] to perform automated proofreading. Automated proofreading cleans reconstructions to a high degree of accuracy relative to manual proofreaders, but it does not extend reconstructions.

**Dendritic proofreading.** At baseline, reconstructed dendrites were generally complete and required little extension[51]. However, they often contained false merges that required cleaning[51]. The dendrites of all of the presynaptic neurons were manually cleaned and extended. The dendrites of other neurons were cleaned with NEURD[28].

**Axonal proofreading.** At baseline, reconstructed axons require both cleaning and extension[51]. Only the axons of presynaptic neurons were manually cleaned and extended. In order to balance morphological completeness (per neuron) and coverage (across projection types), we extended axons to varying degrees of completion. Specifically, we performed full manual proofreading on a subset of neurons ($n = 84$), which involved thoroughly cleaning and extending all axonal branches throughout the dataset. For the remaining neurons ($n = 64$), we applied partial proofreading, focusing exclusively on extending axonal branches that were pre-screened to feedback from HVA to V1. The full list of proofread presynaptic neurons, their area and layer membership, and whether they were fully or partially proofread is included in Supplementary Table 1, and a subset of proofread axons are shown in Extended Data Fig. 1.

**Manual proofreading completion.** As part of manual proofreading protocol[14], proofreaders were instructed to leave annotations at the termination of every neurite indicating its status, whether natural or incomplete. From these annotations, we estimate the frequency of untraceable ends, a rough indicator of incompleteness. For dendrites, the median neuron had a percentage of untraceable ends of 1% ($n = 148$ fully proofread dendritic arbors). For axons, the median neuron had a percentage of untraceable ends (not including those at dataset boundaries) of 43% ($n = 84$ fully proofread axonal arbors).

### Presynaptic neuron selection

Our approach for selecting presynaptic neurons for manual proofreading was designed to enrich for higher-order connectivity motifs within and (especially) across visual areas. Because connection probability drops off with distance[52], we elected to initially focus proofreading efforts on spatially clustered cells in two cylindrical columns spanning cortical layers 2–5, with the first column located in V1 and the second located in RL. Column centres were chosen according to

retinotopic maps, as it has been shown that inter-areal projections are retinotopically matched[29,43]. During the proofreading process we added an additional column in V1 and another spanning the RL and AL border, to increase coverage of the volume. Finally, a few HVA cells that were postsynaptic to proofread V1 cells were chosen to enrich for higher-order motifs ($n = 9$).

All neurons selected for proofreading had an oracle score greater than 0.25 and model test correlation (model predictive performance from an intermediate version of the digital twin) greater than 0.15. The first 40 neurons were selected by experienced neuroscientists unblinded to functional properties for an emphasis on functional diversity. All remaining neurons were chosen blind to functional properties.

## Anatomical controls

In order to control for anatomy at the axonal scale, we recruited all visually responsive, well predicted, functionally matched excitatory neurons ($CC_{max} > 0.4$, $CC_{abs} > 0.2$) that are located in the same region as the postsynaptic target, but are not observed to form a synapse with the presynaptic neuron (same-region control). Area membership labels per neuron were used from the MICrONS release[14]. Additionally, control candidates that meet criteria for both the same-region control and the ADP control (described below) will only be included in ADP control.

In order to control for anatomy at the synaptic scale, we recruited all visually responsive, well predicted, functionally matched excitatory neurons ($CC_{max} > 0.4$, $CC_{abs} > 0.2$) with a dendritic skeleton passing within 5 μm of the presynaptic neuron axonal synapse in the presynaptic axonal arbor (3D euclidean distance), but which are not observed to form a synapse with the presynaptic neuron (ADP control). Presynaptic axonal skeletons were computed using the pcg_skel package developed by collaborators at the Allen Institute for Brain Science[53,54]. For postsynaptic dendritic skeletons, we used the automatically proofread and skeletonized dendritic arbors as described[28].

To compute the axon–dendrite co-travel distance ($L_d$) between a pair of neurons, we first discretized both the axonal skeleton of one neuron and the dendritic skeleton of the other neuron so that no edge exceeded a length of 1 μm. Next, we identified all pairs of vertices from the two skeletons that were within 5 μm of each other by performing spatial queries using the KDTree query_ball_tree method from the scipy.spatial module in SciPy[55]. From these proximal vertices (proximity), we identified the associated dendritic edges. The lengths of these dendritic edges were summed to obtain $L_d$.

Synapses were obtained from Table synapses_pni_2[14] and were assigned to an ADP if they were within 3 μm of any vertex in the proximity.

In the case of the joint area and layer analysis (Fig. 4), candidates in both the same-region and ADP controls must additionally match the same layer classification as the postsynaptic target in order to be included. Layer assignment was performed as described[56].

## Measuring functional similarities

**In silico response correlations.** To characterize the pairwise tuning similarity between two modelled neurons, we computed the Pearson correlation of their responses to 2,500 s of natural videos. The natural videos were fed in to the model as trials of 10 s. Model responses were generated at 29.967 Hz and Pearson correlations were computed after binning the responses into 500 ms non-overlapping bins and concatenating across trials.

**In silico feature weight similarity and RF centre distance.** The digital twin model architecture includes a shared core which is trained to represent spatiotemporal features in the stimulus input, and a final layer where the spatiotemporal features at a specific readout location are linearly weighted in order to produce the predicted activity of a specific neuron at the current time point[26]. The readout location and linear feature weight are independently learned for each neuron.

In order to measure the feature weight similarity between two units, we extract the linear feature weights from this final step as vector of length 512, and take the cosine similarity between the two vectors. In order to measure the RF centre distance between two units, we extract the readout location as 2D coordinates on the monitor, and take the angle between them with respect to the mouse's eye, assuming the monitor is centred on, 15 cm away from, and normal to the surface of the mouse's eye at the closest point.

**In silico difference in preferred orientation.** Two hundred and forty blocks of parametric directional visual stimuli (Monet) are shown to the model, with each 15-s block consisting of 16 trials of equally distributed and randomly ordered unique directions of motion between 0 and 360°. A modelled neuron's direction tuning curve is computed as its mean responses to 16 directions averaged across blocks. We calculated the gOSI and the orientation selectivity index (OSI) from the modelled neuron's tuning curve as follows:

$$\text{gOSI} = \frac{\Sigma R_\theta e^{2i\theta}}{\Sigma R_\theta}, \text{OSI} = \frac{R_{po} - R_{ortho}}{R_{po} + R_{ortho}} \tag{1}$$

where $\theta$ is the direction of the stimulus, $R_\theta$ is the mean modelled response to the stimulus at direction $\theta$, and $R_{po}$ and $R_{ortho}$ are the mean modelled responses at the preferred and orthogonal orientation, respectively. The gOSI metric is based on the $1 - \text{CircVar}$ metric in ref. 57, which is a vector-based method designed to reduce the uncertainty in quantifying orientation selectivity of responses, especially in cases where high throughput, unbiased recording methods return many cells with low orientation selectivity, as is the case with calcium imaging. Only neurons with gOSI > 0.25 were included in the analyses in this Article. For neurons selected with our gOSI threshold > 0.25, the computed OSI ranges from 0.43 to 0.99, with mean of 0.56. For both thresholds, the fraction of cells considered orientation tuned (57.4% of co-registered V1 neurons has gOSI > 0.25, 62.7% of co-registered V1 neurons has OSI > 0.4) is similar to those reported in other studies (72% in V1 layer 2/3 (ref. 1), 62.9% in V1 layer 2/3 and 58.0% in V1 layer 4 (ref. 58)). Unit-wise direction tuning curves are then parameterized by a mixture of two von Mises functions with an offset as described in[26]. In brief, the model has the following form:

$$f(\theta|\mu, \kappa, \alpha, \beta, \gamma) = \alpha e^{\kappa \cos(\theta - \mu)} + \beta e^{\kappa \cos(\theta - \mu + \pi)} + \gamma,$$

where $\alpha$ and $\beta$ are the amplitudes of the two von Mises function, $\mu$ is the preferred direction, $\kappa$ is the dispersion, and $\gamma$ is the offset. The parameters are estimated through least squares optimization, minimizing $\sum_\theta (f(\theta|\mu, \kappa\alpha, \beta, \gamma) - R_\theta)^2$. The preferred orientation of a neuron is taken as the modulus of $\mu$ to 180°.

## Validation of the digital twin model

**Validation of in silico signal correlations.** To validate the in silico signal correlations generated by our digital twin model, we first established a benchmark for in vivo signal correlations. We began by determining the optimal number of stimulus repetitions for measuring in vivo signal correlations. Two mice were presented with 6 unique 10-s natural video clips, each repeated 60 times over a 60-min period. Based on the results shown in Extended Data Fig. 2a, we determined that ten repetitions per clip provided a reliable estimate of in vivo signal correlation while maintaining a reasonable experimental time for presenting a large number of clips in subsequent experiments.

With this optimal repetition count established, we conducted experiments with three mice using an expanded set of visual stimuli. These stimuli contained those presented in the MICrONS dataset as described[14], including natural videos, global directional parametric stimuli (Monet), and local directional parametric stimuli (Trippy). Additionally, we presented 36 unique 10-s natural video clips, each

repeated 10 times, totaling 60 min of stimulation. To facilitate comparison with the MICrONS dataset and establish a robust ground truth, we divided these 36 clips into two sets: a benchmark set of 30 clips repeated 10 times, serving as our 'ground truth' for signal correlation, and a MICrONS-equivalent set of 6 clips repeated 10 times, mimicking the amount of repeated natural clip data available in the MICrONS dataset.

For each mouse, we trained a digital twin model using the same architecture and training data as the MICrONS digital twin. This allowed us to generate three signal correlation matrices for comparison: an in vivo matrix computed from the MICrONS-equivalent set, an in silico matrix generated by the digital twin model using 250 novel natural video clips, and a benchmark matrix computed from the 30-clip set. To compare these matrices, we randomly sampled submatrices of signal correlations between 1,000 neurons. We then performed hierarchical clustering using Ward's method on the benchmark matrix and used the resulting dendrogram to sort neurons. This sorting was applied to the MICrONS-equivalent and in silico matrices for visual comparison, as shown in Extended Data Fig. 2b. Following this initial comparison, we calculated the Pearson correlation coefficient between the corresponding entries in the lower triangles of the three matrices. To assess statistical significance, we employed a resampling approach, performing 1,000 random splits of the benchmark and MICrONS-equivalent sets, from which we estimated the standard deviation and resampling-based P value of the Pearson correlations. This comprehensive approach enabled us to evaluate how well our digital twin model's in silico signal correlations matched the ground truth compared to in vivo measurements with limited data, thus validating the model's performance in replicating neural response correlations.

**Validation of RF centre.** To validate the RF estimates of our digital twin model, we conducted additional experiments and analyses comparing in vivo and in silico RF measurements. We collected three additional functional scans using an expanded set of visual stimuli. These stimuli contained those presented in the MICrONS dataset[14], including natural videos, global directional parametric stimuli (Monet), and local directional parametric stimuli (Trippy). Additionally, we presented 57.6 min of sparse noise stimuli. The sparse noise stimuli consisted of bright (pixel value 255) and dark (pixel value 0) square dots, each approximately 6° in visual angle, presented on a grey background (pixel value 127) in a randomized order. These dots were presented at 12 positions covering 70° of visual angle along both the horizontal and vertical axes of the screen. Each presentation lasted 200 ms, and each condition was repeated 60 times.

We computed the in vivo STA RFs by cross-correlating the visual stimuli with deconvolved calcium traces. STAs for bright dots (on-STAs) and dark dots (off-STAs) were estimated independently and then combined by taking the pixel-wise maximum of the on- and off-STAs. We then presented the same sparse noise stimuli to the digital twin model and computed in silico STA RFs using the model responses. To assess STA quality, we generated response predictions by multiplying each neuron's STA with the stimulus frames and compared these predictions to either the in vivo trial-averaged responses or model responses using Pearson correlation coefficients. Neurons with correlations greater than 0.2 were considered well-characterized. We then extracted the STA RF centres by fitting a 2D Gaussian to the STAs, with fits yielding an *r*-squared value over 0.5 considered well fit. Our analysis revealed that 44% of all imaged neurons had well-characterized, well-fit in vivo STAs.

To visualize the retinotopic maps measured with either in vivo STA or in silico STA, we converted the STA RF centres to azimuth and elevation angles, assuming the mouse was looking at the centre of the monitor. To exclude partially measured STAs, we included only neurons with well-characterized and well-fitted STA centres located in the central 8 × 8 square of the entire 12 × 12 stimulus grid (27% of all imaged neurons) for the analysis presented in Extended Data Fig. 4a–d,f,h. For the analysis in Extended Data Fig. 4e,g,i, we included neurons with the bottom 25% of response correlations that have STA centres located in the central 8 × 8 square.

Finally, we quantified the coherence of the retinotopic maps as Spearman's correlation between cortical distances and retinotopic distance of 10,000 randomly sampled neuron pairs in a cortical region that does not contain a retinotopic reversal (shown as the dotted circled region in the retinotopic maps visualized in Extended Data Fig. 4).

**Validation of orientation tuning.** To validate in silico orientation tuning with in vivo orientation tuning, we collected three additional functional scans with an expanded set of stimuli. These stimuli contained those presented in the MICrONS dataset[14], including natural videos, global directional parametric stimuli (Monet), and local directional parametric stimuli (Trippy). In addition, each stimulus contained an additional 40 min of trials, randomly intermixed, as follows:
- Unique global directional parametric stimulus (Monet): 120 seeds, 15 s each, 1 repeat per scan, 30 min total. Seeds conserved across all scans.
- Oracle global directional parametric stimulus (Monet): 4 seeds, 15 s each, 10 repeats, 10 min total. Seeds conserved across all scans.

We characterized both the in vivo orientation tuning in response to 30 min of global directional parametric stimulus (Monet; Extended Data Fig. 8a), as well as the in silico orientation tuning as described above for digital twin models with shared cores and readouts trained on neurons from the same scans, in response to stimuli matching the composition and duration of the MICrONS release scans (Extended Data Fig. 8b). When we applied a threshold of gOSI >0.25, we found that 95% of cells had an absolute difference between their in silico and in vivo preferred orientations less than 19.7°.

## Statistics and reproducibility

All functional and anatomical data used for functional connectomics analysis in this Article were collected from a single mouse. No sample size calculation was performed a priori for this mouse, and sample sizes (number of connections tested) were determined by using all available data for each experimental group, matching or exceeding previous studies of similar design. Within the context of this single mouse, we treated pairs of neurons as independent samples for statistical analysis. For the validation of in silico properties in Extended Data Figs. 2, 4 and 8, we reproduced the validation results in three mice, with this number of validation mice chosen empirically without formal sample size calculations.

## Statistical analysis of mean signal correlations

We employed paired *t*-tests to compare signal correlations between presynaptic neurons and three groups of potential target neurons: connected postsynaptic neurons, ADP neurons, and same-region control neurons. Our analysis focused on presynaptic neurons with more than ten postsynaptic targets for each projection type to ensure robust comparisons. For each presynaptic neuron, we computed mean signal correlations with its synaptically connected postsynaptic targets, ADP neurons (neurons with dendrites in proximity to the presynaptic axon but not synaptically connected), and same-region control neurons (neurons in the same brain region but without proximal axon–dendrite contacts). We then performed paired *t*-tests to compare these mean correlations. For example, to compare connected and ADP neuron pairs, we conducted a paired *t*-test between each presynaptic neuron's mean signal correlation with its postsynaptic targets versus its mean signal correlation with ADP neurons. This approach allowed us to control for variability across presynaptic neurons while directly comparing their correlations with different target groups. All statistical analyses were performed using the scipy package in Python. We set the significance level (α) at 0.05 for all tests. To account for multiple comparisons, we adjusted P values using the Benjamini–Hochberg procedure as implemented in the statsmodels package.

## Visualization of the relationship between $L_d$, $N_{syn}/L_d$ and the functional similarities

**Visualization of $L_d$.** To quantify the changes in $L_d$ as a function of functional similarities, we restrict our analysis to neuron pairs with no synaptic connections observed between them. We then follow these steps:

(1) We compute the mean $L_d$ and mean functional similarities for each presynaptic neuron across all other neurons that no synaptic connections with the presynaptic neuron were observed.

(2) We subtract the presynaptic mean from each of the pairwise $L_d$ and functional similarities between every neuron pair to compute $\Delta L_d$ and $\Delta$similarity.

(3) The neurons pairs are then binned by $\Delta$similarity and the average $\Delta L_d$ is computed for each bin.

(4) The standard deviation of average $\Delta L_d$ is estimated by bootstrapping. Specifically, we resampled the neuron pairs 1,000 times with replacement and repeated steps 1–3.

Only bins with more than ten connected neuron pairs and more than ten presynaptic neurons are included in the visualization.

**Visualization of $N_{syn}/L_d$.** To quantify the changes in synapse density, $N_{syn}/L_d$, as a function of functional similarities, we restrict our analysis to neuron pairs with positive $L_d$ observed between them. We then follow these steps:

(1) We compute the mean $N_{syn}/L_d$ and mean functional similarities for each presynaptic neuron across all other neurons that no synaptic connections with the presynaptic neuron were observed.

(2) We subtract the presynaptic mean from each of the pairwise $N_{syn}/L_d$ and functional similarities between every neuron pair to compute $\Delta N_{syn}/L_d$ and $\Delta$similarity.

(3) The neurons pairs are then binned by $\Delta$similarity and the average $\Delta N_{syn}/L_d$ is computed for each bin.

(4) The standard deviation of average $\Delta L_d$ is estimated through bootstrapping. Specifically, we resampled the neuron pairs 1,000 times with replacement and repeated steps 1–3.

Only bins with more than ten connected neuron pairs and more than ten presynaptic neurons are included in the visualization.

## Statistical modelling of like-to-like rules for different anatomical measurements

**Axon–dendrite co-travel distance.** $L_d$ measures the distance dendrites of one neuron travel within 5 μm from another neuron's axon. Most pairs of neurons' axons and dendrites never come into close proximity with each other, and their $L_d$ is zero. Thus, the $L_d$ distribution is a non-negative continuous distribution with a substantial non-zero probability measure at zero $L_d$. Thus, we modelled $L_d$ as a random variable following the Tweedie exponential dispersion family (with Tweedie index parameter $\xi \in (1, 2)$). Tweedie distributions with such index parameters are Poisson mixtures of gamma distributions, commonly used to model continuous data with exact zeros. We assume two neurons' axons and dendrites travel within 5 μm at $N$ proximity points, where $N \sim \text{Pois}(\lambda^*)$, $\lambda^*$ is the mean number of axonal dendritic proximal contacts of the Poisson distribution. When $N > 0$, we assume the distance dendrites travel within 5 μm at each proximal point $z_i(i = 1, ..., N)$ follows a Gamma distribution $\text{Gam}(\mu, \phi)$. Under these assumptions, the total potential synapsing distance

$$L_d = \Sigma_{i=1}^N z_i,$$

where $L_d = 0$ when $N = 0$, follows a Tweedie distribution with $1 < \xi < 2$. We then model the relationship between $L_d$, functional similarities Sim (for example, signal correlation, feature weight similarity and RF location distance between two neurons), and projection types Proj using

a Tweedie-distributed GLMM with a log link function. For analysis at the brain area level, Proj is a nominal variable with four categories: V1 intra-area, HVA intra-area projections, feedforward projections and feedback projections. Similarly for analysis at the brain area and layer level, Proj is a nominal variable that contains categories of all area and layer projection types. We apply GLMMs for modelling as they have been recommended for accounting for multi-level data dependencies in datasets[59], such as the projection types and presynaptic neuron proofreading progress in our study. We specify the model as follows:

$$\log(L_{d_{ij}}) = \beta_0 + \beta_1 \text{Sim}_{ij} + \beta_2 \text{Proj}_{k(i,j)}$$
$$+ \beta_3 \text{Sim}_{ij} \times \text{Proj}_{k(i,j)} + u_{k(i,j),i} + \epsilon_{ij}$$

where:

- $L_{d_{ij}}$ is the axon–dendrite co-travel distance between presynaptic neuron $i$ and postsynaptic neuron $j$.
- $\text{Sim}_{ij}$ is the functional similarity between the neuron pair.
- $\text{Proj}_{k(i,j)}$ is the projection type of the neuron pair $(i, j)$.
- $\beta_1, \beta_2$ and $\beta_3$ are the fixed effect coefficients of the functional similarity, projection type and their interaction term, respectively.
- $\beta_0$ is the intercept.
- $u_{k(i,j),i}$ is the random effect accounting for the projection type $k$ and the proofread status associated with presynaptic neuron $i$.
- $\epsilon_{ij}$ is the error term, following a Tweedie distribution.

The coefficients $\beta_1, \beta_2$ and $\beta_3$ represent how functional similarities and projection types affect connectivity at the axonal scale. We fit the models for each functional similarity independently using the glmmTMB R package. The goodness-of-fit of the estimated models is reported as Nakagawa's $R$ squared, computed with the performance R package. We define the axonal-scale like-to-like coefficients for each functional similarity and projection type as the estimated linear association between each category of functional similarity conditioned on the projection type. The coefficient estimates and the corresponding significance tests are computed for the fitted GLMM using the emtrends function from the emmeans R package.

**Number of synapses.** $N_{syn}$ measures the number of synapses between two neurons. We model it as a Poisson-distributed random variable and its relationship to functional similarities as a GLMM model with the following specifications:

$$\log(N_{syn_{ij}}) = \beta_0 + \beta_1 \text{Sim}_{ij} + \beta_2 \text{Proj}_{k(i,j)}$$
$$+ \beta_3 \text{Sim}_{ij} \times \text{Proj}_{k(i,j)} + u_{k(i,j),i} + \epsilon_{ij}$$

where:

- $N_{syn_{ij}}$ is the number of synapses between presynaptic neuron $i$ and postsynaptic neuron $j$.
- $\text{Sim}_{ij}$ is the functional similarity between the neuron pair.
- $\text{Proj}_{k(i,j)}$ is the projection type of the neuron pair $(i, j)$.
- $\beta_1, \beta_2$, and $\beta_3$ are the fixed effect coefficients of the functional similarity, projection type, and their interaction term, respectively.
- $\beta_0$ is the intercept.
- $u_{k(i,j),i}$ is the random effect accounting for the projection type $k$ and the proofread status associated with presynaptic neuron $i$.
- $\epsilon_{ij}$ is the error term, following a Poisson distribution.

The coefficients $\beta_1, \beta_2$, and $\beta_3$ estimate how the functional similarities and projection types affect connectivity regardless of the spatial scales (axonal or synaptic). We fit the models for each functional similarity independently using the glmmTMB R package. The goodness-of-fit of the estimated models is reported as Nakagawa's $R$ squared, computed with the performance R package. We define the axonal-scale like-to-like

coefficients for each functional similarity and projection type as the estimated linear association between each category of functional similarity conditioned on the projection type. The coefficient estimates and the corresponding significance tests are computed for the fitted GLMM using the emtrends function from the emmeans R package.

**Synapse conversion rate.** $N_{\mathrm{syn}}/L_{\mathrm{d}}$ measures the number of synapses per millimetre axon–dendrite co-travel distance for each neuron pair. To quantify its relationship to functional similarities, we adopted the following GLMM model:

$$\log(N_{\mathrm{syn}_{ij}}) = \beta_0 + \beta_1 \mathrm{Sim}_{ij} + \beta_2 \mathrm{Proj}_{m(i,j)}$$
$$+ \beta_3 \mathrm{Sim}_{ij} \times \mathrm{Proj}_{k(i,j)} + u_{k(i,j),i} + \epsilon_{ij} + \log(L_{\mathrm{d}_{ij}})$$

where:
- $N_{\mathrm{syn}_{ij}}$ is the number of synapses between presynaptic neuron $i$ and postsynaptic neuron $j$.
- $L_{\mathrm{d}_{ij}}$ is the axon–dendrite co-travel distance between the neuron pair.
- $\mathrm{Sim}_{ij}$ is the functional similarity between the neuron pair.
- $\mathrm{Proj}_{m(i,j)}$ is the projection type of the neuron pair $(i,j)$.
- $\beta_1, \beta_2$ and $\beta_3$ are the fixed effect coefficients of the functional similarity, projection type and their interaction term, respectively.
- $\beta_0$ is the intercept.
- $u_{k(i,j),i}$ is the random effect accounting for the projection type $k$ and the proofread status associated with presynaptic neuron $i$.
- $\epsilon_{ij}$ is the error term, following a Poisson distribution.
  The above equation can be re-arranged to:

$$\log\!\left(\frac{N_{\mathrm{syn}_{ij}}}{L_{\mathrm{d}_{ij}}}\right) = \beta_0 + \beta_1 \mathrm{Sim}_{ij} + \beta_2 \mathrm{Proj}_{k(i,j)}$$
$$+ \beta_3 \mathrm{Sim}_{ij} \times \mathrm{Proj}_{k(i,j)} + u_{k(i,j),i} + \epsilon_{ij}$$

Thus, $\beta_1, \beta_2$ and $\beta_3$ model how the functional similarities affect synapse conversion rate ($N_{\mathrm{syn}}/L_{\mathrm{d}}$) at the synaptic scale. We fit the models for each functional similarity independently using the glmmTMB R package. The goodness-of-fit of the estimated models is reported as Nakagawa's $R$ squared, computed with the performance R package. We define the like-to-like coefficients of each functional similarity for each projection type as the estimated linear association between each category of functional similarity conditioned on the projection type. The coefficient estimates and the corresponding significance tests are computed for the fitted GLMM using the emtrends function from the emmeans R package. To avoid fitting models to projection types with little data or dominated by few presynaptic neurons, for all the models described above, we only include and report like-to-like coefficients to projection types with more than 30 synapses observed, more than 5 presynaptic neurons, and with none of the presynaptic neurons contributing more than half of all synapses observed.

## Statistical analysis of functional similarities and synaptic anatomy
We investigated the relationship between functional similarities of neurons and the anatomical features of their synaptic connections. Our analysis accounted for the confounding effect of axon–dendrite co-travel distance ($L_{\mathrm{d}}$), which correlates with both functional similarities and synaptic measurements. To isolate the effect of synaptic anatomy on functional similarity, we employed a two-step regression approach:

First, we condition our analysis on the effect of $L_{\mathrm{d}}$ from the functional similarity measure. This process involves:
(1) Fitting a linear regression model with functional similarity as the dependent variable and $L_{\mathrm{d}}$ as the independent variable.
(2) Calculating the residuals from this model, which represent the variation in functional similarity that cannot be explained by $L_{\mathrm{d}}$ alone.

These residuals become our new measure of functional similarity, adjusted for the influence of $L_{\mathrm{d}}$. Next, we constructed a linear regression model using these residuals as the dependent variable. The independent variables in this model included anatomical measurements of synaptic connections, the total number of synapses between neuron pairs and the mean synaptic cleft volume.

This approach allows us to test whether synaptic measurements significantly predict functional similarities between neurons, beyond what can be explained by their physical proximity (as measured by $L_{\mathrm{d}}$).

## Relationship between like-to-like connectivity and somatic distance
To quantify how like-to-like connectivity changes with the physical distance between neuron somas within V1, we measured the 3D EM somatic distances between connected neuron pairs compared to all other unconnected functionally co-registered neuron pairs that shared the same presynaptic neuron population. We then assessed how several measures of functional similarity varied with somatic distance. These measures included in vivo signal correlation, in silico signal correlation, feature weight similarity, and RF centre distance. Somatic distances were binned into 100-μm intervals for this analysis.

## RNN model
The RNN model used to produce the results in Fig. 6 consisted of a vanilla RNN layer with 1,000 hidden units and a hyperbolic tangent activation function simulated over 20 time steps. Static inputs were obtained by passing MNIST images through a linear layer. Outputs were obtained by passing the hidden activations at the last time step through another linear layer. All three layers were trained for 10 epochs, a batch size of 512, the categorical cross entropy loss function, and the Adam optimizer in PyTorch. A pre- and postsynaptic neuron pair was classified as connected if the associated weight was in the top 35th percentile of all weights, specifically if the weight larger than 0.01. In Fig. 6d, weights were chosen as candidates for ablation if the weight was above 0.01 and the neurons' signal correlation was above 0.2. About 10.5% of the weights met these criteria, and ablated weights were selected randomly from this set. Changing the thresholds for weights and signal correlations did not change our conclusions. For several different hyperparameters (number of hidden units, number of time steps, and batch size), we tested sensitivity by varying the hyperparameter across a fourfold range of values. Our overall findings did not change. We also verified that the results are similar when the model is trained on FashionMNIST in place of MNIST.

## Common input analysis
**Functional similarity among all postsynaptic neurons sharing one common input.** For a connectivity graph $G$, we define

$$\rho_G(i) = \frac{\Sigma_{j \neq i} \Sigma_{k \notin (i,j)} \mathrm{Sim}_{jk} N_{\mathrm{syn}_{ij}} N_{\mathrm{syn}_{ik}}}{\Sigma_{j \neq i} \Sigma_{k \notin (i,j)} N_{\mathrm{syn}_{ij}} N_{\mathrm{syn}_{ik}}},$$

where $i$ is a presynaptic neuron and $j$, $k$ are any two neurons in the volume. $\rho$ measures the average similarity of all postsynaptic neurons of the presynaptic neuron $i$.

**Estimation of $\rho$ expected by pairwise like-to-like connectivity rules.** With the observed connectivity graph $G$, we estimated the relationship between $N_{\mathrm{syn}}$ and the functional similarities (in silico signal correlation, feature weight similarity and RF centre distance) with GLMM similar to the specifications for modelling the number of synapses described above. Instead of modelling each functional similarity independently, we included all functional similarities and their interaction with projection types in a single model to account for as much pairwise connectivity rule as possible. We then estimated the expected functional similarity among all postsynaptic neurons sharing one common input $i$ as:

$$\rho_G'(i) = \frac{\Sigma_{j \neq i} \Sigma_{k \notin (i,j)} \mathrm{Sim}_{jk} N'_{\mathrm{syn}_{ij}} N'_{\mathrm{syn}_{ik}}}{\Sigma_{j \neq i} \Sigma_{k \notin (i,j)} N'_{\mathrm{syn}_{ij}} N'_{\mathrm{syn}_{ik}}},$$

where $N'_{\mathrm{syn}_{ij}}$ is the predicted number of synapses between neurons $i$ and $j$ given their functional similarities by the GLMM.

**Common input analysis in the RNN model.** To replicate the common input analysis in the RNN model, we applied the same methodology described above. We first binarized connections in the RNN model with the same threshold described in the RNN section above. The relationship between connectivity and signal correlations is then modelled as a logistic regression model. The average postsynaptic neuron similarity and the expected similarity is estimated as:

$$\rho_G(i) = \frac{\Sigma_{j \neq i} \Sigma_{k \notin (i,j)} \mathrm{Sim}_{jk} W_{ij} W_{ik}}{\Sigma_{j \neq i} \Sigma_{k \notin (i,j)} W_{ij} W_{ik}},$$

where $W_{ij}$ is the binarized weight between artificial neurons $i$ and $j$.

$$\rho_G'(i) = \frac{\Sigma_{j \neq i} \Sigma_{k \notin (i,j)} \mathrm{Sim}_{jk} P_{ij} P_{ik}}{\Sigma_{j \neq i} \Sigma_{k \notin (i,j)} P_{ij} P_{ik}},$$

where $P_{ij}$ is the connection probability between artificial neurons $i$ and $j$ predicted by the logistic regression model.

## Software

Experiments and analysis are carried out with custom built data pipelines. Our custom data pipeline (https://github.com/cajal/pipeline) is developed in Matlab (2016a, 2018b), Python (3.6, 3.8) and R (4.3.3) with the following tools: PsychToolBox 3, ScanImage (2017b), DeepLabCut (2.0.5), CAIMAN (1.0) and Labview (2016) were used for data collection; DataJoint (0.12.9), MySQL (5.7.37) and CAVE (4.12, 4.14, 4.16) were used for storing and managing data; Meshparty (1.16), NEURD (1.0.0) and pcg_skel (0.3, 0.2) were used for morphology analysis; Numpy (1.23.5), pandas (1.5.3), SciPy (1.10.1), statsmodels (0.13.5), scikit-learn (1.2.1), PyTorch (1.12.1), tidyverse (2.0.0), glmmTMB (1.1.10), performance (0.12.2) and emmeans (1.10.3) were used for model training and statistical analysis; Matplotlib (3.7.0), seaborn (0.12.2), HoloViews (1.15.4), Ipyvolume (0.5.2) and Neuroglancer (https://github.com/seung-lab/neuroglancer) were used for graphical visualization; and Jupyter (ipykernel:6.21.2), Docker (23.0.1) and Kubernetes (1.22.11) were used for code development and deployment.

## Reporting summary

Further information on research design is available in the Nature Portfolio Reporting Summary linked to this article.

## Data availability

All MICrONS data are available on BossDB (https://doi.org/10.60533/BOSS-2021-T0SY; please also see https://www.microns-explorer.org/cortical-mm3 for details).

## Code availability

Custom developed code used in the analysis can be found at (https://github.com/cajal/microns-funconn-2025, tag 1.0.0).

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

**Acknowledgements** The authors thank D. Markowitz, the IARPA MICrONS programme manager, who coordinated this work during all three phases of the MICrONS programme; IARPA programme managers J. Vogelstein and D. Markowitz for co-developing the MICrONS Program; and J. Wang, IARPA SETA for her assistance. The work was supported by the MICrONS Program of Intelligence Advanced Research Projects Activity (IARPA) via the Department of Interior/Interior Business Center (DoI/IBC) contract numbers D16PC00003, D16PC00004 and D16PC00005. The US Government is authorized to reproduce and distribute reprints for Governmental purposes notwithstanding any copyright annotation thereon. X.P. acknowledges support from NSF CAREER grant IOS-1552868. Zhuokun Ding, S. Papadopoulos, R.R., K.J., X.P. and A.S.T. acknowledge support from NSF NeuroNex grant 1707400. A.S.T., X.P., K.J. and J.R. are supported by RF1 MH130416. A.S.T. acknowledges support from National Institute of Mental Health and National Institute of Neurological Disorders And Stroke under Award Number U19MH114830 and National Eye Institute award numbers R01 EY026927 and Core Grant for Vision Research T32-EY-002520-37. A.B.K. acknowledges support from a training fellowship from the Gulf Coast Consortia, on the NLM Training Program in Biomedical Informatics and Data Science (T15LM007093). R.R. acknowledges support from Air Force Office of Scientific Research (AFOSR) award number FA9550-21-1-0223. Disclaimer: the views and conclusions contained herein are those of the authors and should not be interpreted as necessarily representing the official policies or endorsements, either expressed or implied, of IARPA, DoI/IBC, or the US Government.

**Author contributions** We adopted the following contribution categories from CRediT (Contributor Roles Taxonomy). Authors within each category are sorted in the same order as in the author list. Conceptualization: Zhuokun Ding, P.G.F., S. Papadopoulos, X.P., J.R. and A.S.T. Methodology: Zhuokun Ding, P.G.F., S. Papadopoulos, E. Y. Wang, B.C., C.P., A.C., D.Y., E.F., K.J., R.R., E. Y. Walker and A.S.T. Software: Zhuokun Ding, P.G.F., S. Papadopoulos, E. Y. Wang, B.C., C.P., D.Y., E.F. and R.R. Validation: Zhuokun Ding, P.G.F., S. Papadopoulos, E. Y. Wang. Formal analysis: Zhuokun Ding, P.G.F., S. Papadopoulos, A.C., A.B.K. and R.R. Investigation: Zhuokun Ding, P.G.F., S. Papadopoulos, E. Y. Wang, S. Patel, D.T., J.F., Zhiwei Ding, L.N., R.F., K.P., T. Muhammad and R.R. Resources: P.G.F., S. Papadopoulos, S. Patel, J.A.B., A.L.B., D.B., J.B., D.J.B., M.A.C., E.C., S.D., L.E., A.H., Z.J., C.J., D.K., N.K., S.K., K. Lee, K. Li, R.L., T. Macrina, G.M., E.M., S.S.M., S.M., B.N., S. Popovych, C.M.S.-M., W.S., M.T., R.T., N.L.T., W.W., J.W., W.Y., S.Y., D.Y., E.F., F.S., R.R., H.S.S., F.C., X.P., J.R. and A.S.T. Data curation: Zhuokun Ding, P.G.F., S. Papadopoulos, B.C., C.P. and Zhiwei Ding. Writing, original draft: Zhuokun Ding, P.G.F., S. Papadopoulos, E. Y. Wang, R.R. and J.R. Writing, review and editing: Zhuokun Ding, P.G.F., S. Papadopoulos, A.C., J.F., S. Patel, C.M.S.-M., K.J., R.R., F.C., N.M.d.C., R.C.R., E. Y. Walker, X.P., J.R. and A.S.T. Visualization: Zhuokun Ding, P.G.F., S. Papadopoulos and R.R. Supervision: P.G.F., X.P., J.R. and A.S.T. Project administration: Zhuokun Ding, P.G.F., S. Papadopoulos, J.R. and A.S.T. Funding acquisition: H.S.S., F.C., N.M.d.C., R.C.R., X.P., J.R. and A.S.T.

**Competing interests** A.S.T., J.R., E.Y. Walker, and D.Y. are co-founders of DataJoint Inc. in which they have financial interests. The other authors declare no competing interests.

**Additional information**
**Correspondence and requests for materials** should be addressed to Jacob Reimer or Andreas S. Tolias.

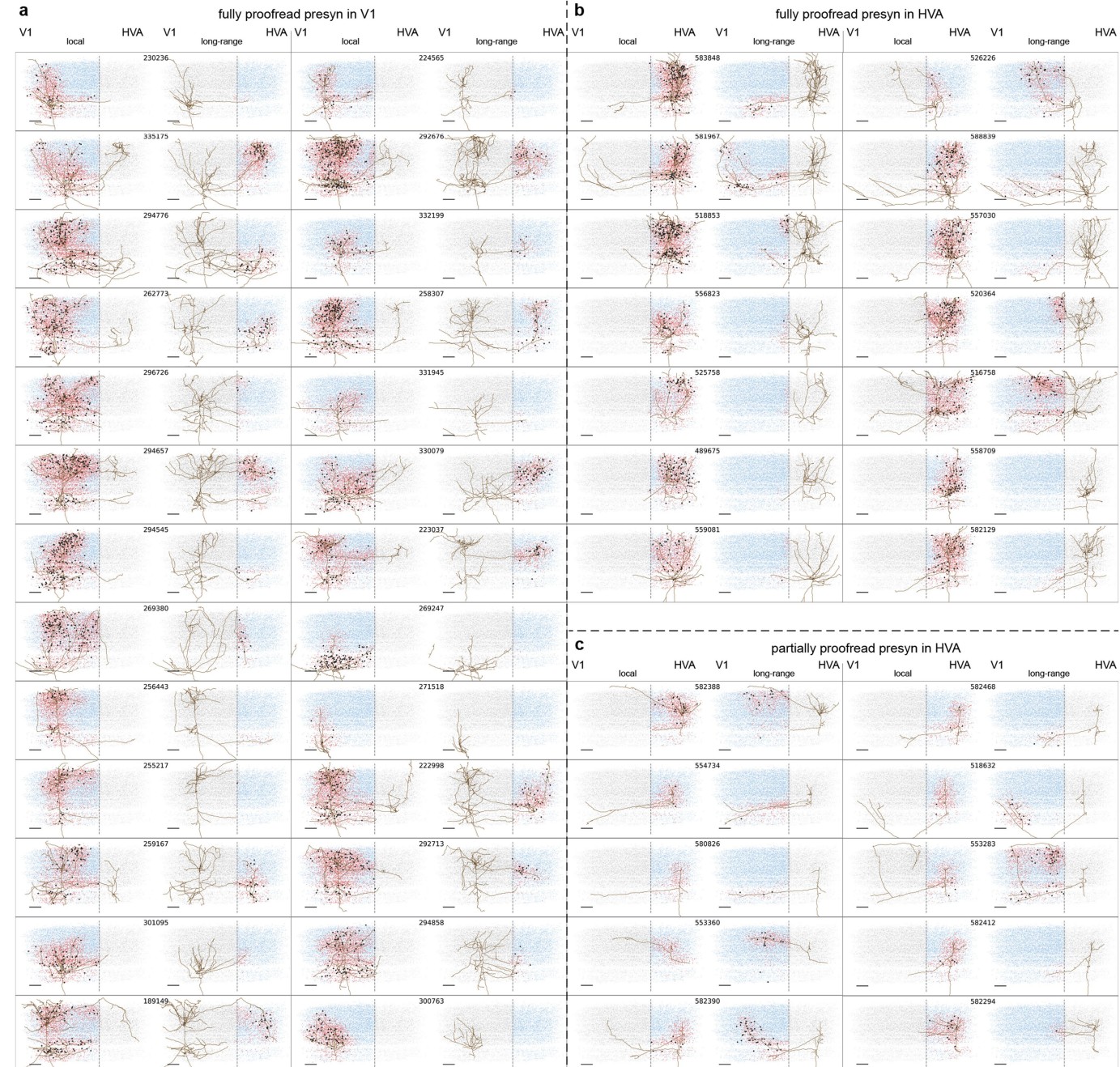

**Extended Data Fig. 1 | Example proofread presynaptic axons in EM cortical space and their connected, ADP, and same region controls.** The axon for every presynaptic (presyn) neuron is shown twice, once as a "local" projection type and again as a "long-range" type (even if the neuron has no local or long-range projections). The six digit ID from Table "nucleus_detection_v0"[14] is displayed above both plots. For each plot, the soma centroids of connected neurons, ADP controls, and same region controls are plotted in black, red, and blue, respectively. Gray dots are soma centroids of all other functionally matched neurons not used as controls for that presyn. The dashed gray line represents the V1-HVA boundary. Scale bar = 100 μm. **a**, Example fully proofread presynaptic axons with somas in V1. "Fully proofread" neurons are those where a proofreader attempted to extend every axonal branch to completion. **b**, Example fully proofread presynaptic axons with somas in HVA **c**, Example partially proofread presynaptic axons with somas in HVA. "Partially proofread" neurons are those where a proofreader only extended axonal branches that were pre-screened for whether they projected inter-areally (specifically to enrich for feedback connections).

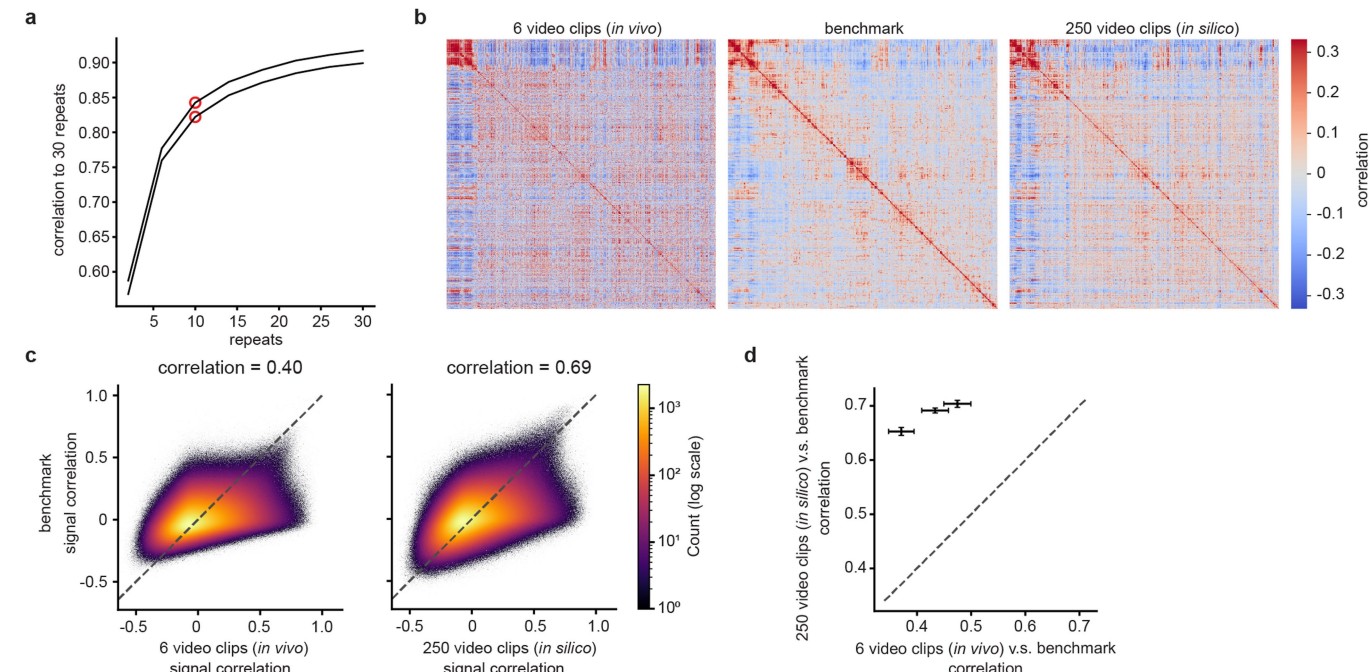

**Extended Data Fig. 2 | The digital twin signal correlations align better with the in vivo benchmark than in vivo signal correlations generated with less data. a**, Correlation of in vivo signal correlations generated with 6 video clips and varying numbers of repeats to in vivo signal correlations generated with 6 clips and 30 repeats, for two animals. 10 repeats (red marker) reasonably approximates the saturation point and is the number used for all other analyses. **b**, Signal correlation matrices of 1000 neurons generated from in vivo responses to 6 video clips (left), in vivo responses to 30 video clips (benchmark, middle) and digital twin responses to 250 video clips (in silico, right). The benchmark matrix is ordered by Ward's hierarchical clustering. The in vivo and in silico signal correlation matrices are ordered in the same order as the benchmark matrix. The fine structure of the in silico matrix is qualitatively more similar

to the benchmark than the in vivo matrix generated with 6 video clips is to the benchmark. **c**, 2D heatmaps of signal correlations from the benchmark (same benchmark as in **b**) vs in vivo responses to 6 video clips (left) and in silico responses to 250 clips (right). The correlation of in silico signal correlations to the benchmark is higher than the correlation of in vivo signal correlations generated with 6 video clips to the benchmark (0.69 vs 0.40). Colorbar: 2D bin counts in log scale. **d**, The correlation of in silico signal correlations to the benchmark vs the correlation of in vivo signal correlations generated with 6 video clips to the benchmark for three animals. Error bars are standard deviations estimated through resampling. All data points are in the upper left corner indicating that in silico signal correlations outperform in vivo signal correlations generated with 6 video clips. (p-value < 0.001 for all three animals).

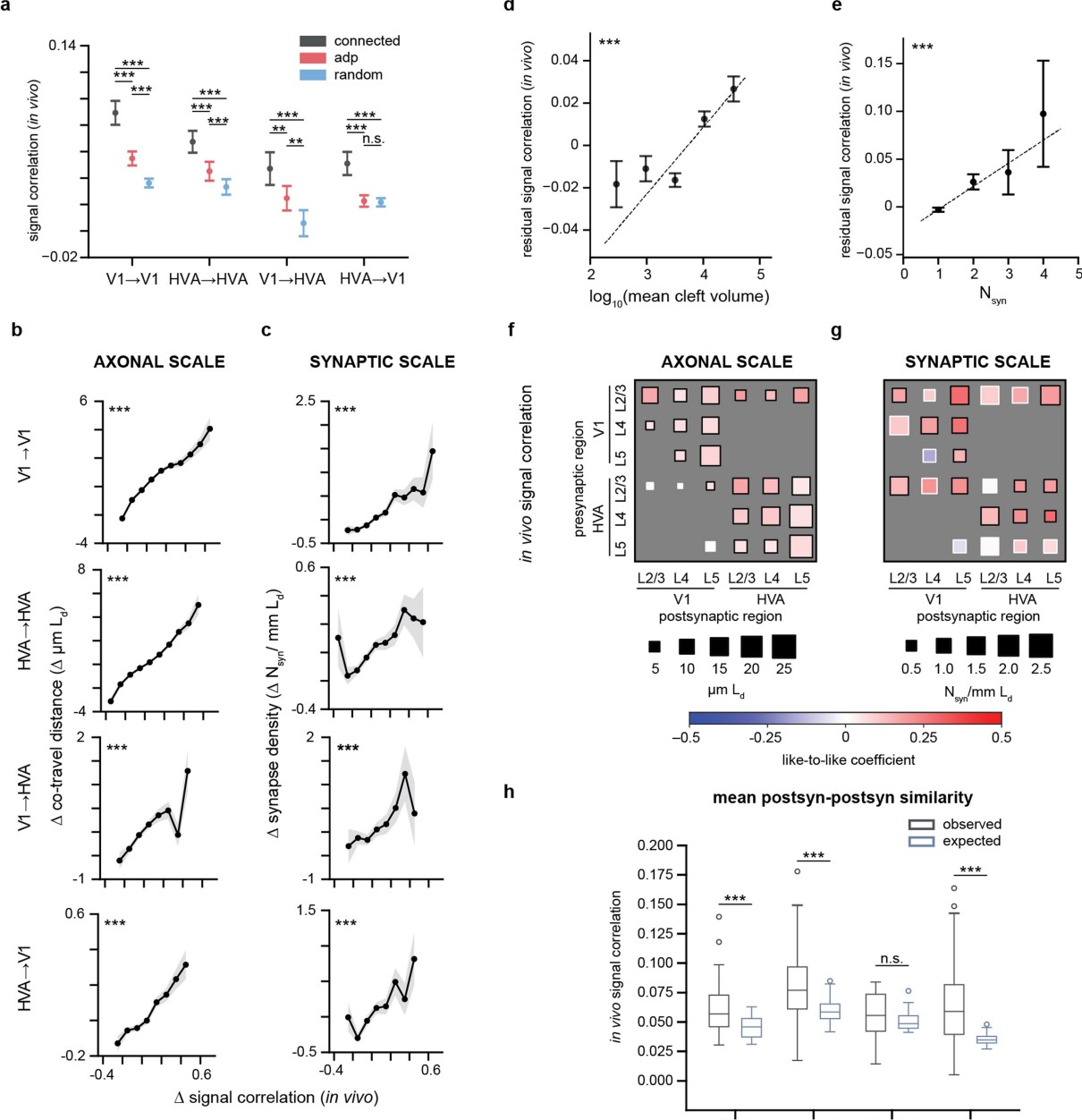

**Extended Data Fig. 3 | Synaptic connectivity increases with empirical signal correlations measured directly in vivo rather than via the digital twin. a**, Mean in vivo signal correlation is different (mean ± sem, paired t-test) for connected pairs, ADP controls, and same area controls for all projection types, as in Fig. 2d. **b**, Axon-dendrite co-travel distance ($\mu m L_d$) increases in a graded fashion with in vivo signal correlation for all projection types, as in Fig. 2e. **c** Synapse density ($N_{syn}/mmL_d$) increases in a graded fashion with signal correlation, for all projection types, as in Fig. 2f. The shaded regions in **b** and **c** are bootstrap-based standard deviation. **d**, Synapse size ($log_{10}$ cleft volume in voxels) is positively correlated with in vivo signal correlation after regressing

out $L_d$ (p-value by linear regression), as in Fig. 2h. **e**, In vivo signal correlations increases with number of synapses after regressing out $L_d$ (p-values by linear regression), as in Fig. 2j. **f**, Area/layer joint membership breakout as in Fig. 4 for in vivo signal correlations at axonal scale. **g**, Area/layer joint membership breakout as in Fig. 4 for in vivo signal correlations at synaptic scale. **h**, Comparison of the observed and expected postsynaptic functional similarity as in Fig. 5 for in vivo signal correlations. (For all panels, * = p-value < 0.05, ** = p-value < 0.01, *** = p-value < 0.001, multiple comparison correction by BH procedure. For statistics and sample sizes, see Supplementary Tables 23–28).

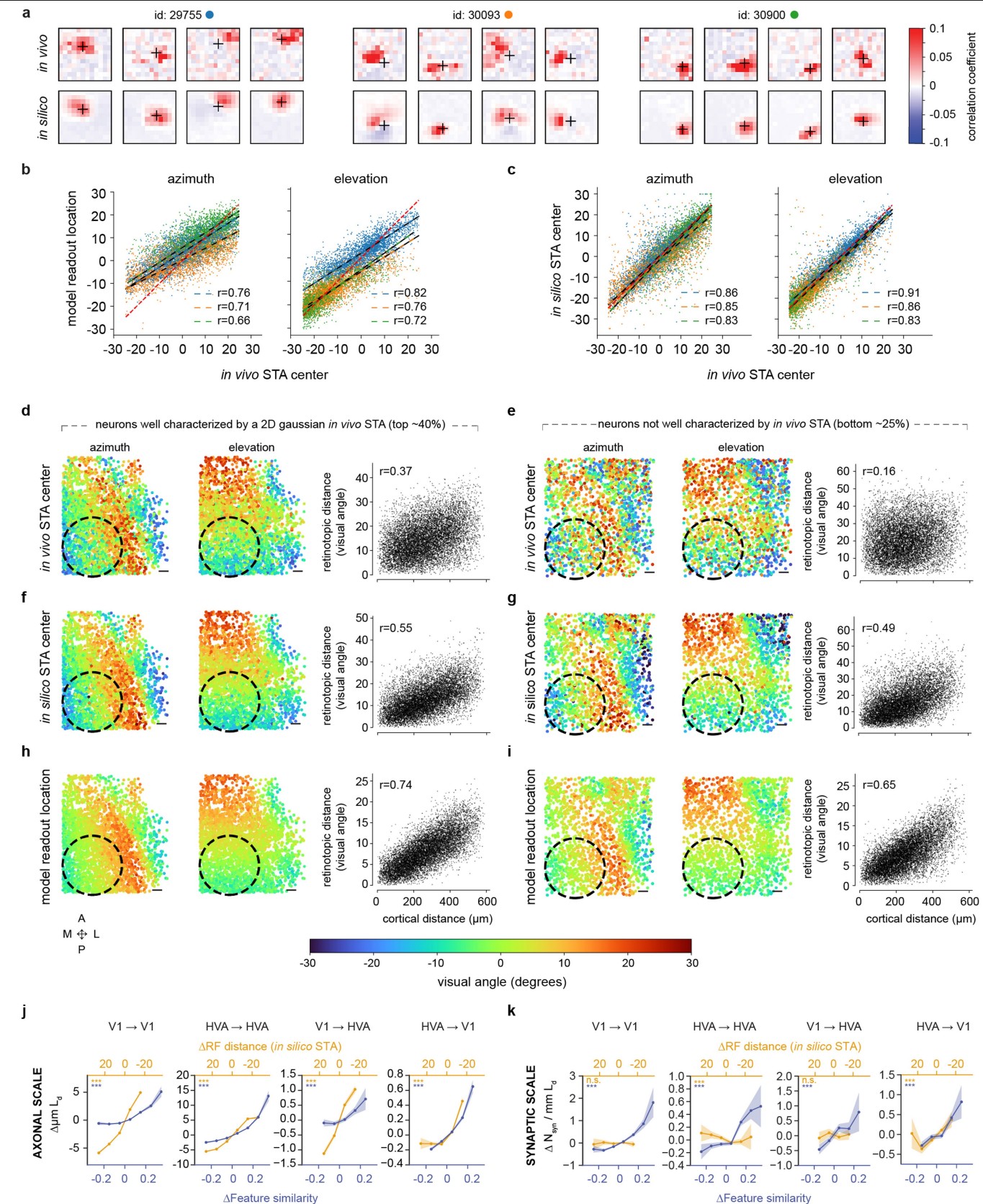

**Extended Data Fig. 4** | See next page for caption.

**Extended Data Fig. 4 | Comparison of in silico and in vivo receptive field centers. a**, Visual comparison of Spike-Triggered Average receptive fields (STAs) generated from in vivo responses to a sparse noise stimulus (top row) vs STAs generated from in silico responses to the same stimulus (bottom row) for three animals (blue, orange, and green). The black cross represents the model readout location. Examples are randomly chosen from the top 44% of neurons remaining after a threshold on in vivo STA quality is applied. **b**, Model readout location vs in vivo STA center for azimuth coordinate (left) and elevation coordinate (right). **c**, in silico STA center vs in vivo STA center for azimuth coordinate (left) and elevation coordinate (right). **d-i**, Retinotopic maps for animal id: 29755. **d**, Retinotopic maps generated from in vivo STA centers with top 44% of neurons after an in vivo STA quality threshold is applied. Left: Azimuth retinotopic map, each dot represents one neuron in the cortical space, the color represents the azimuthal visual angle of its receptive field center.

Middle: Elevation retinotopic map. Right: The coherence of the retinotopic map is visualized as a scatterplot of the pairwise cortical distance vs. the pairwise retinotopic distance for 10,000 randomly selected neuron pairs within the dotted circular region in the retinotopic maps. The coherence is quantified as the Spearman's rank correlation coefficient between the distances. **e**, Retinotopic maps generated from in vivo STA centers with the bottom 25% neurons based on the quality of the in vivo STA. **f, g**, Retinotopic maps generated from in silico STA centers for the same neurons in **d** and **e**. **h, i**, Retinotopic maps generated from the digital twin model readout location for the same neurons in **d** and **e**. Colorbar: degree of visual angle for both azimuth and elevation coordinates. Anatomical axes: A = anterior, P = posterior, M = medial, L = lateral. Scale bar: 100 μm. **j, k**, Analysis in Fig. 3a,b repeated with in silico STA centers instead of model readout location.

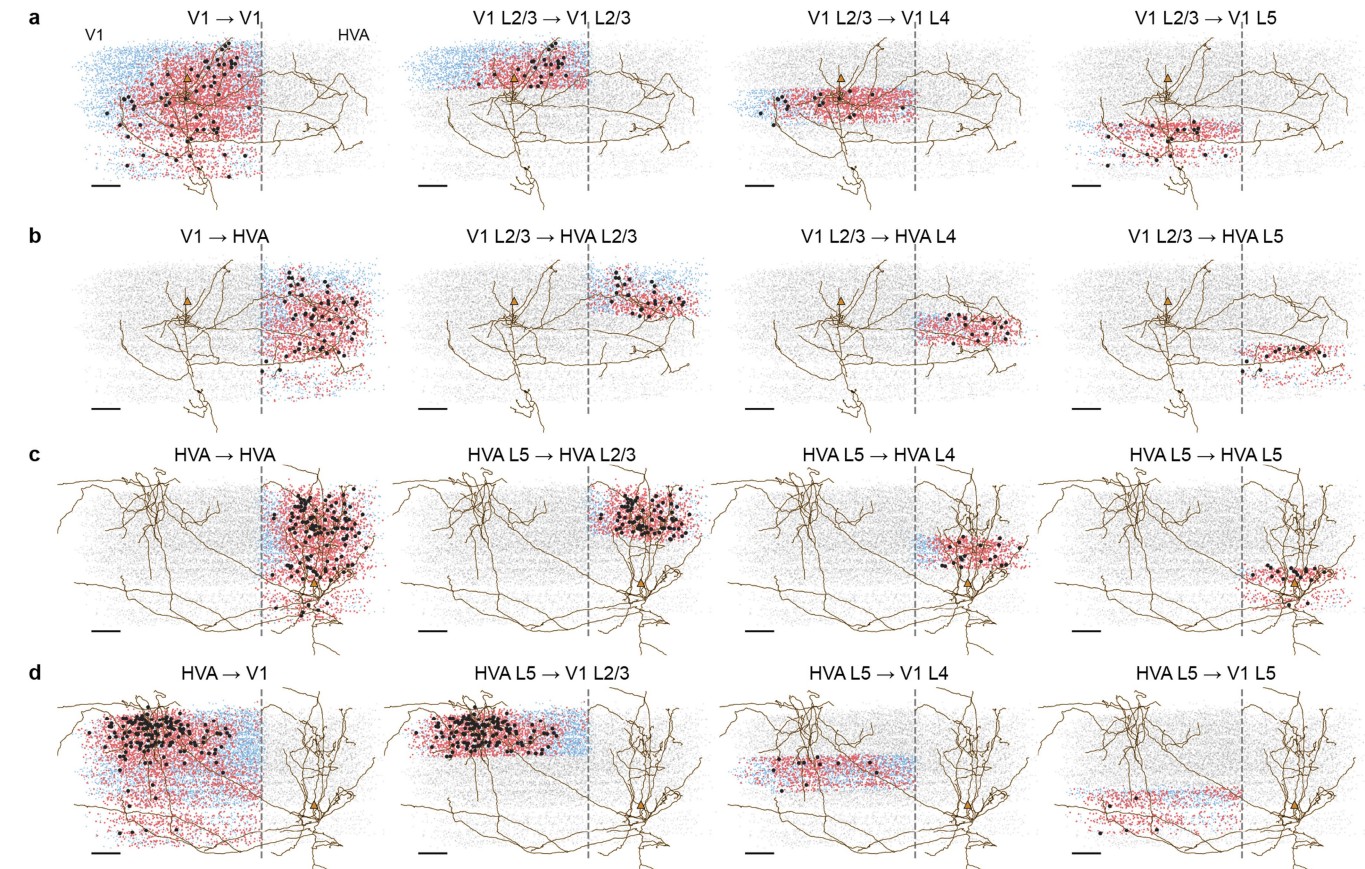

**Extended Data Fig. 5 | Example of connected, ADP, and same area controls for the area/layer analysis in Fig. 4.** For all panels, a single presynaptic neuron skeleton is shown as in Fig. 2c with its postsyns (black dots), ADP controls (red dots), same area controls (blue dots), and all other functionally matched neurons (gray dots). The location of the presynaptic soma is indicated by the orange triangle. The vertical gray dotted line is the V1 and HVA boundary, with V1 on the left, and HVA on the right. The scale bars are 100 µm **a**, The V1 L2/3 neuron reproduced from Fig. 2c with its V1 → V1 postsyns and controls (left), and shown again with only its V1 L2/3 (left middle), V1 L4 (right middle) and V1 L5 (right) postsyns and controls. **b**, The V1 L2/3 neuron reproduced from Fig. 2c with its V1 → HVA postsyns and controls (left), and shown again with only its HVA L2/3 (left middle), HVA L4 (right middle) and HVA L5 (right) postsyns and controls. **c**, The HVA L5 neuron reproduced from Fig. 2c with its HVA → HVA postsyns and controls (left), and shown again with only its HVA L2/3 (left middle), HVA L4 (right middle) and HVA L5 (right) postsyns and controls. **d**, The HVA L5 neuron reproduced from Fig. 2c with its HVA → V1 postsyns and controls (left), and shown again with only its V1 L2/3 (left middle), V1 L4 (right middle) and V1 L5 (right) postsyns and controls.

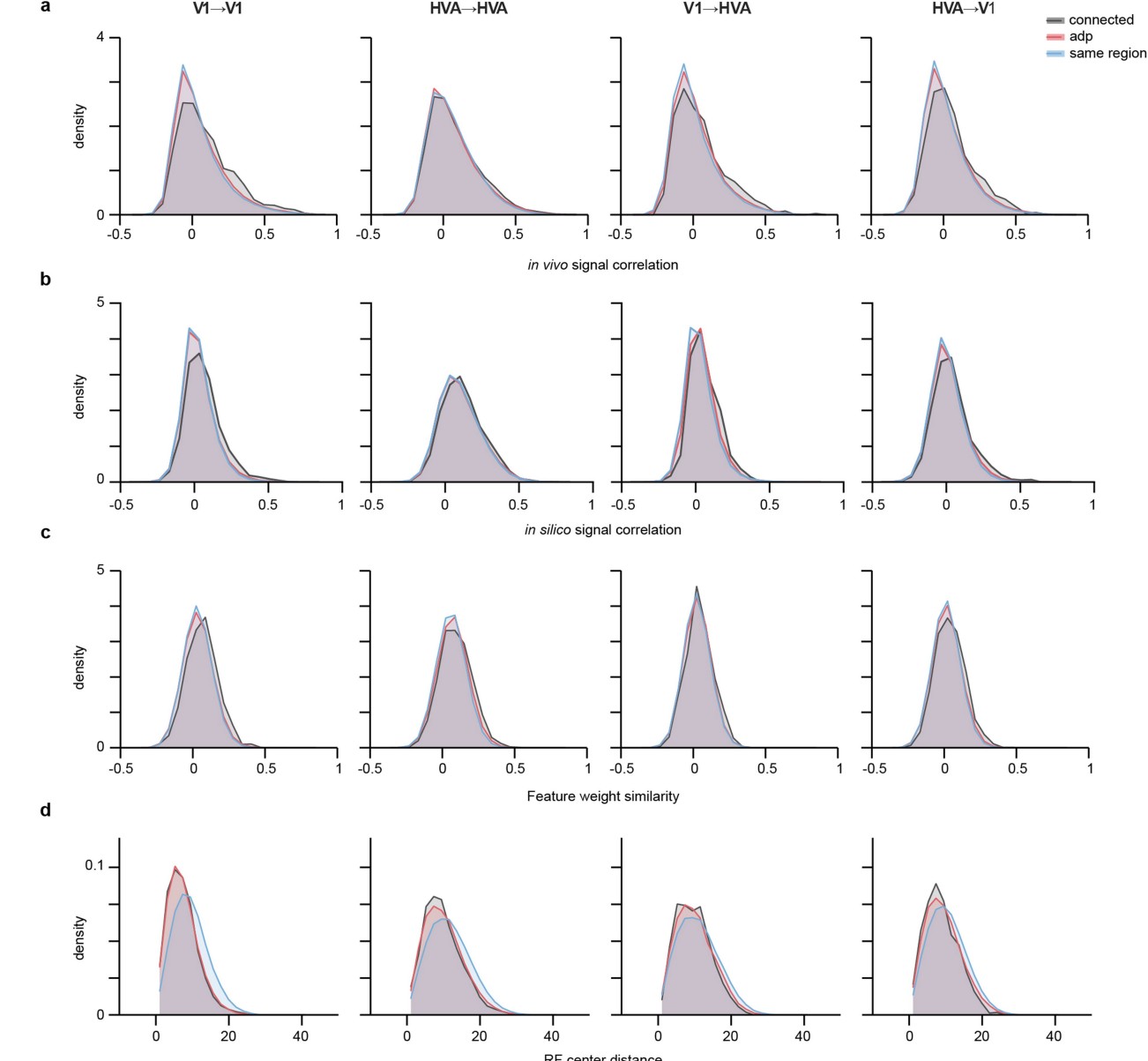

**Extended Data Fig. 6 | Distribution of pairwise functional measurements.**
Density distribution of connected pairs (black), ADP control pairs (red) and
same region control pairs (blue) for in vivo signal correlations (**a**), in silico signal
correlations (**b**), feature weight similarity (**c**), and RF center distance (**d**) for all
projection types.

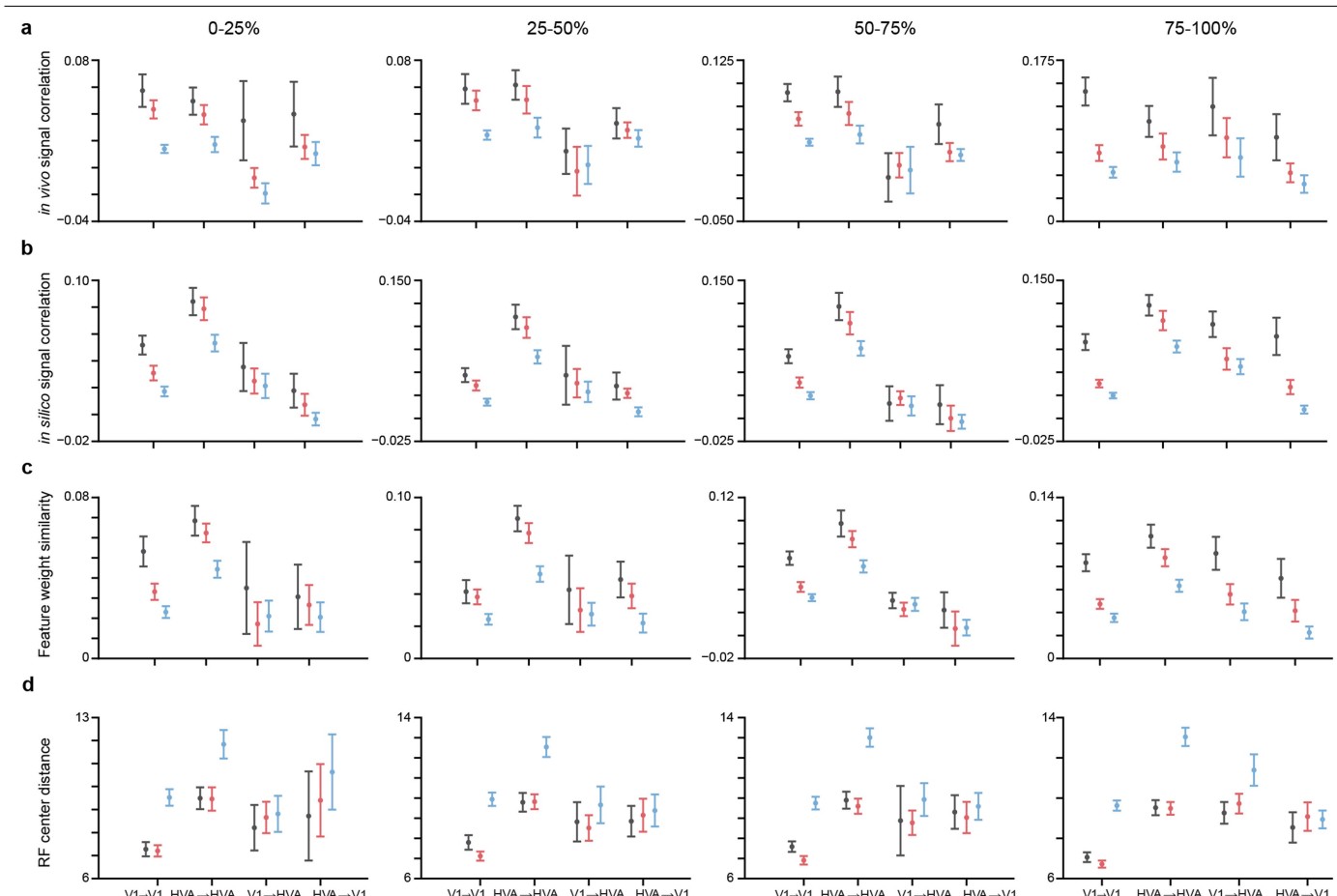

**Extended Data Fig. 7 | Pairwise functional measurements across varying levels of model predictive performance.** Mean of in vivo signal correlations (**a**), in silico signal correlations (**b**), feature weight similarity (**c**), and RF center distance (**d**) for all projection types across 4 quantiles of model predictive performance ($CC_{abs}$). All panels share a base filtering for visual responsiveness ($CC_{max} > 0.4$, 90% of neurons pass this threshold). Presynaptic neurons are filtered to $CC_{abs} > 0.2$ (4 did not pass this threshold).

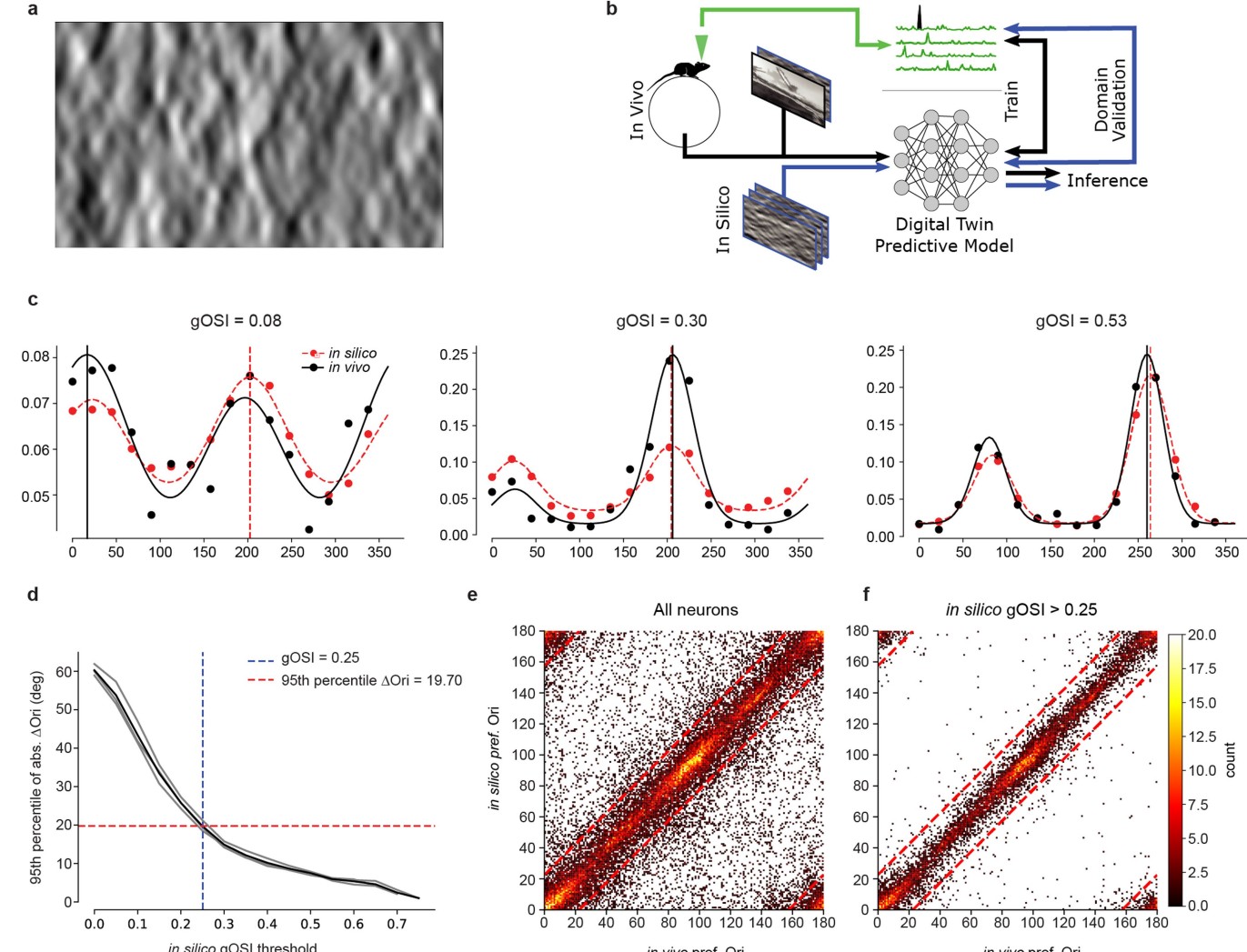

**Extended Data Fig. 8 | In silico orientation tuning is consistent with in vivo orientation tuning. a**, Sample frame from global directional parametric stimulus ("Monet") used to characterize orientation and direction selectivity. Directional motion was orthogonal to orientation, and was tested at 22.5° intervals. **b**, Schematic of domain validation experimental design. In a single scan in a new animal, neuronal responses are collected in response to sufficient stimuli to both train the digital twin model (natural stimuli) and characterize orientation tuning ("Monet") from in vivo responses. Later, in silico orientation tuning is extracted from model responses to parametric stimuli, and compared against in vivo orientation tuning for the same neurons. **c**, Comparison of

in silico (red dotted line) and in vivo (black solid line) mean responses per stimulus direction, fitted tuning curves, and extracted preferred orientation (vertical lines) for three neurons of various gOSI levels. **d**, 95th percentile difference in preferred orientation between in silico and in vivo fitted responses as a function of gOSI threshold. Dotted lines correspond to gOSI > 0.25 threshold applied for all analyses and resulting 95th percentile difference in preferred orientation ≈19.7° across all three animals imaged. Lines correspond to individual animals (gray) or cumulative across all animals (black). **e, f**, Two-dimensional histogram of in silico versus in vivo preferred orientation for all neurons across three animals (**e**) and only neurons with gOSI > 0.25 (**f**).

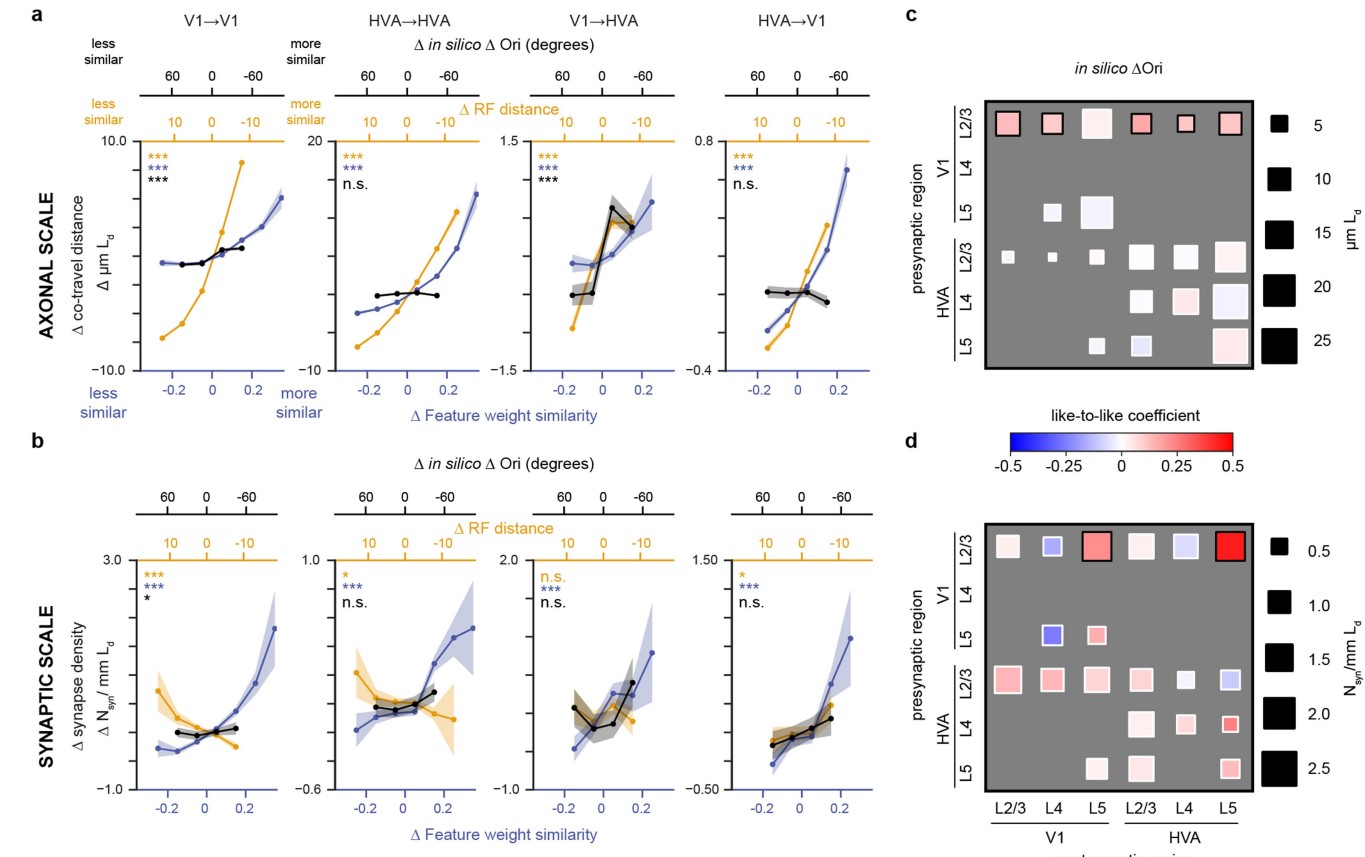

**Extended Data Fig. 9 | Analysis in Fig. 3 repeated with in silico orientation preference. a**, Difference in preferred orientation (Δ Ori) derived from in silico responses to parametric stimuli for tuned (gOSI > 0.25) neurons along with both feature weight similarity and receptive field center distance (reproduced from Fig. 3) at axonal scale. **b**, same as in **a**, at synaptic scale. For analysis with synapse size, see Extended Data Fig. 13) **c**, Area/layer joint membership breakout as in Fig. 4 for in silico Δ ori at axonal scale. **d**, As in **c** but at synaptic scale. All analyses are centered per presyn by accounting for the presyn mean (e.g. Δ feature weight similarity). For statistics and sample sizes, see Supplementary Tables 7–14, 29–34.

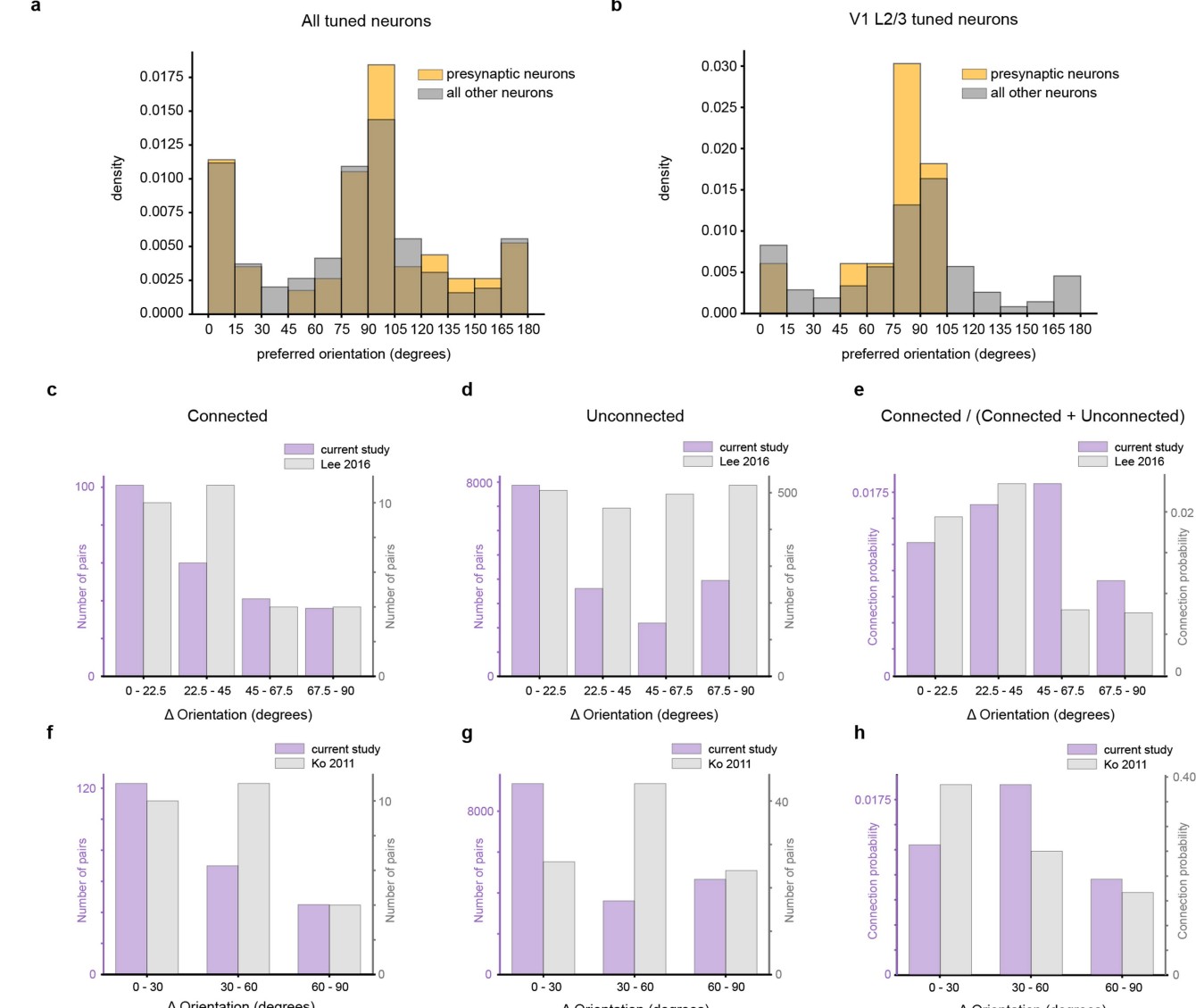

**Extended Data Fig. 10 | Distribution of in silico orientation preference and comparison to previous literature. a**, Distribution of orientation preference of tuned neurons (gOSI > 0.25) derived from in silico responses to parametric stimuli (see Methods). Note the cardinal bias in orientation preference distribution, in which orientation preference for 0 and 90 degree angles is overrepresented. Gold: presynaptic neurons, Gray: all other neurons. **b**, As in **a** but for tuned neurons in V1 L2/3. Difference in preferred orientation (Δ Orientation) for neurons in V1 L2/3 for connected pairs (**c, f**), unconnected pairs (**d, g**), and the ratio of connected / unconnected ("connection probability",

**e, h**) for our study vs Lee et al.[5] (**c-e**) and vs Ko et al.[1] (**f-h**). The connected V1 L2/3 neurons in our study show a strong like-to-like effect, consistent with both Lee et al.[5] and Ko et al.[1] (**c, f**), however unlike Lee et al.[5] and Ko et al.[1], the unconnected neurons in our study also show a strong like-to-like effect (**d, g**) indicating that the like-to-like effect seen in connected pairs results from an orientation preference bias. This bias likely explains why we do not observe significant a like-to-like effect between V1 L2/3 neurons at axonal scale or synaptic scale in Extended Data Fig. 9, (i.e. when pairs are tested against region-matched controls).

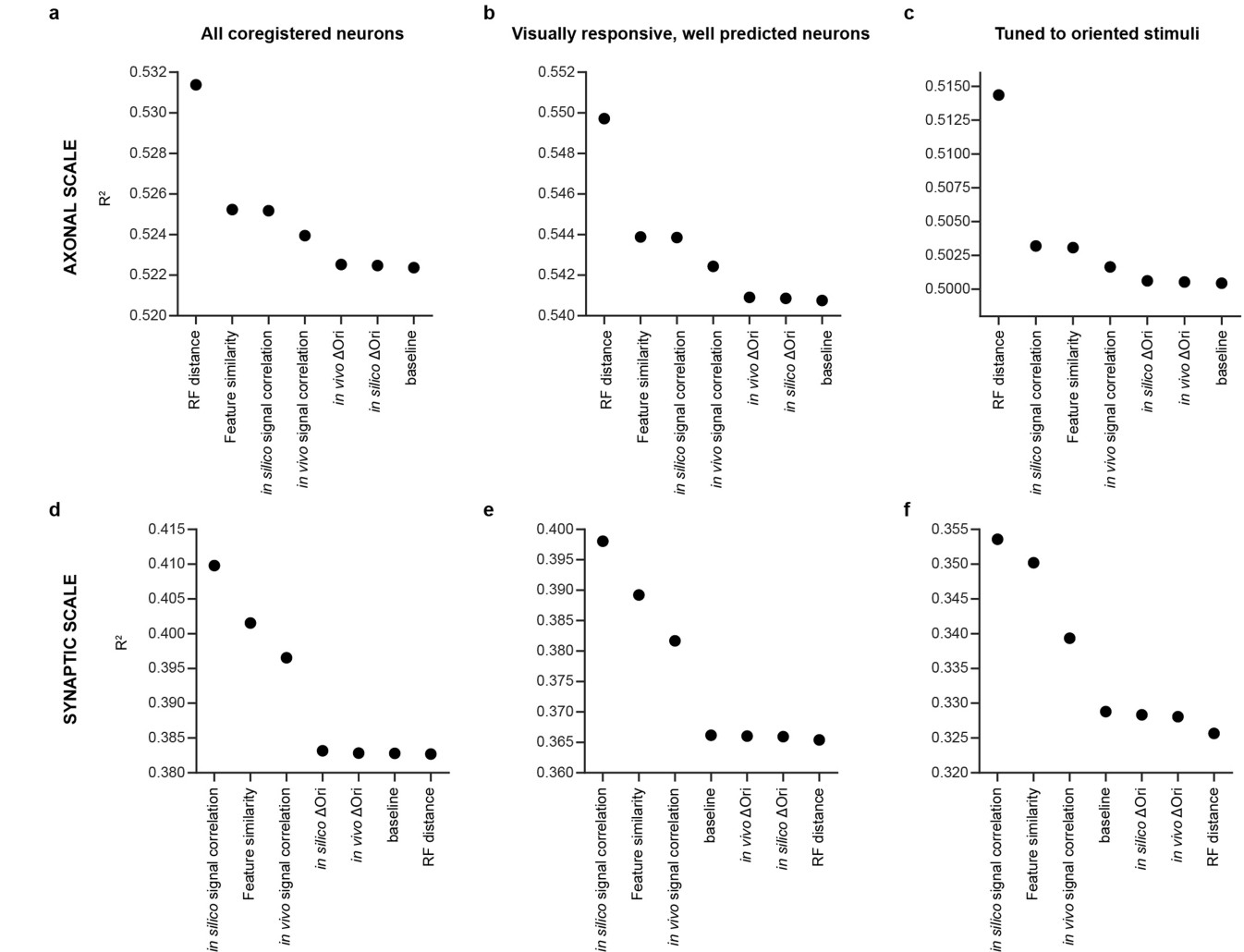

**Extended Data Fig. 11 | Performance of various functional metrics in predicting axon-dendrite co-travel distance ($L_d$, Axonal scale) or synapse density ($N_{syn}/mmL_d$, Synaptic scale).** Model performance of GLMMs (Nakagawa's conditional $R^2$) for predicting axon-dendrite co-travel distance ($L_d$): **a**, **b**, **c** and synapse density ($N_{syn}/mmL_d$): **d**, **e**, **f**, for all coregistered neurons: **a**, **d**, all visually responsive, well predicted neurons: **b**, **e**, and neurons tuned to oriented stimuli: **c**, **f**. The GLMMs are fit to predict axon-dendrite co-travel distance or synapse density independently with each functional metric, the projection type, and the interaction between the two while considering the interaction term of projection type and presynaptic neuron identity as random effects. The baseline models were not fitted with information about functional metrics. They predict axon-dendrite co-travel distance or synapse density with the projection type alone while considering the interaction term of projection type and presynaptic neuron identity as random effects.

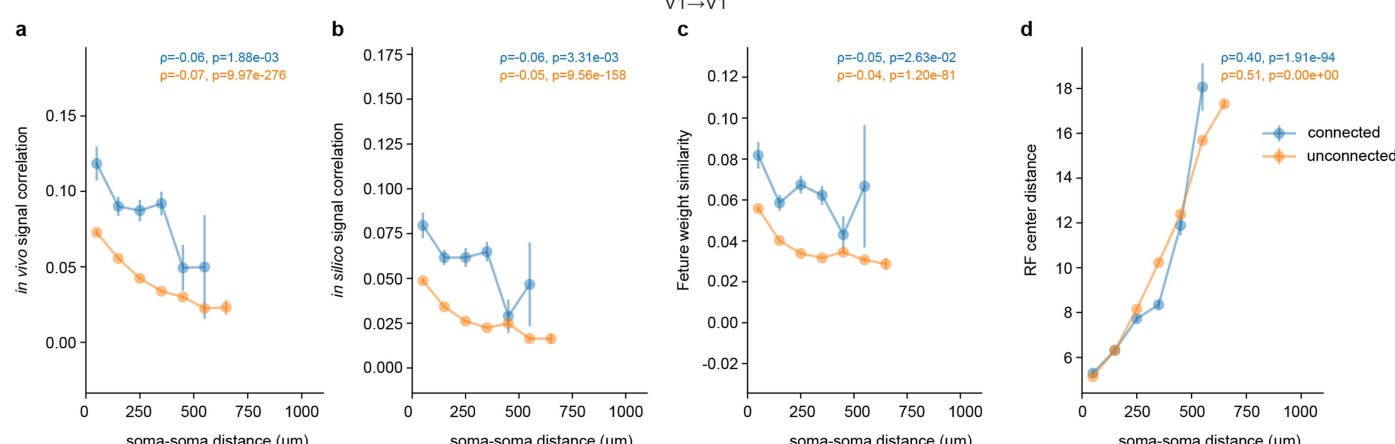

**Extended Data Fig. 12 | Pairwise functional measurements for connected and unconnected pairs vs soma-soma distance for V1→V1 connections.** Mean ± SEM of in vivo signal correlations (**a**), in silico signal correlations (**b**), feature weight similarity (**c**), and RF center distance (**d**) vs soma-soma distance for connected and unconnected pairs for V1→V1 connections.

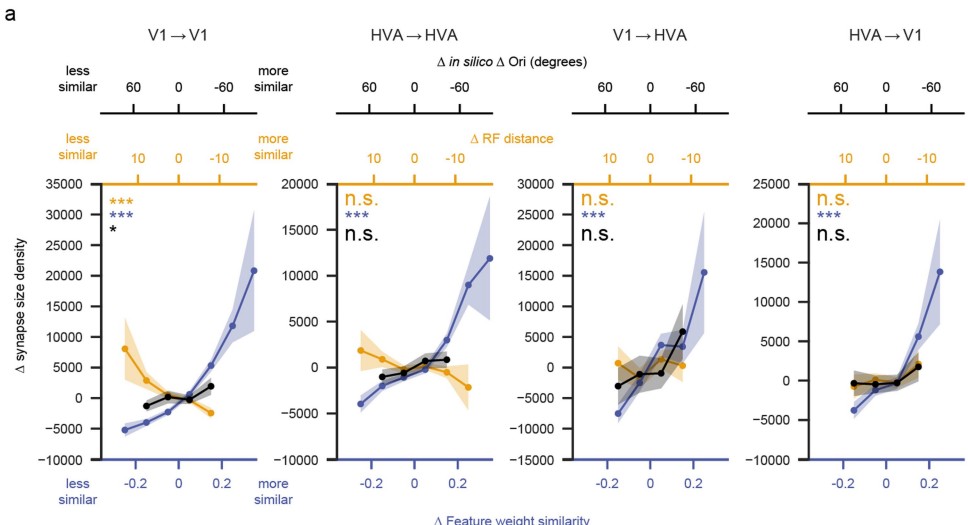

**Extended Data Fig. 13 | In silico Δ Ori, RF location similarity, and feature weight similarity vs synapse size density. a**, Analysis in Extended Data Fig. 9a,b repeated with synapse size density, rather than synapse density.

Synapse size density is computed similarly to synapse density except that the numerator $N_{syn}$ is replaced with the summed synaptic weight (sum of all synapse cleft volumes for a pair of neurons in $4 \times 4 \times 40\,nm^3$ voxels).

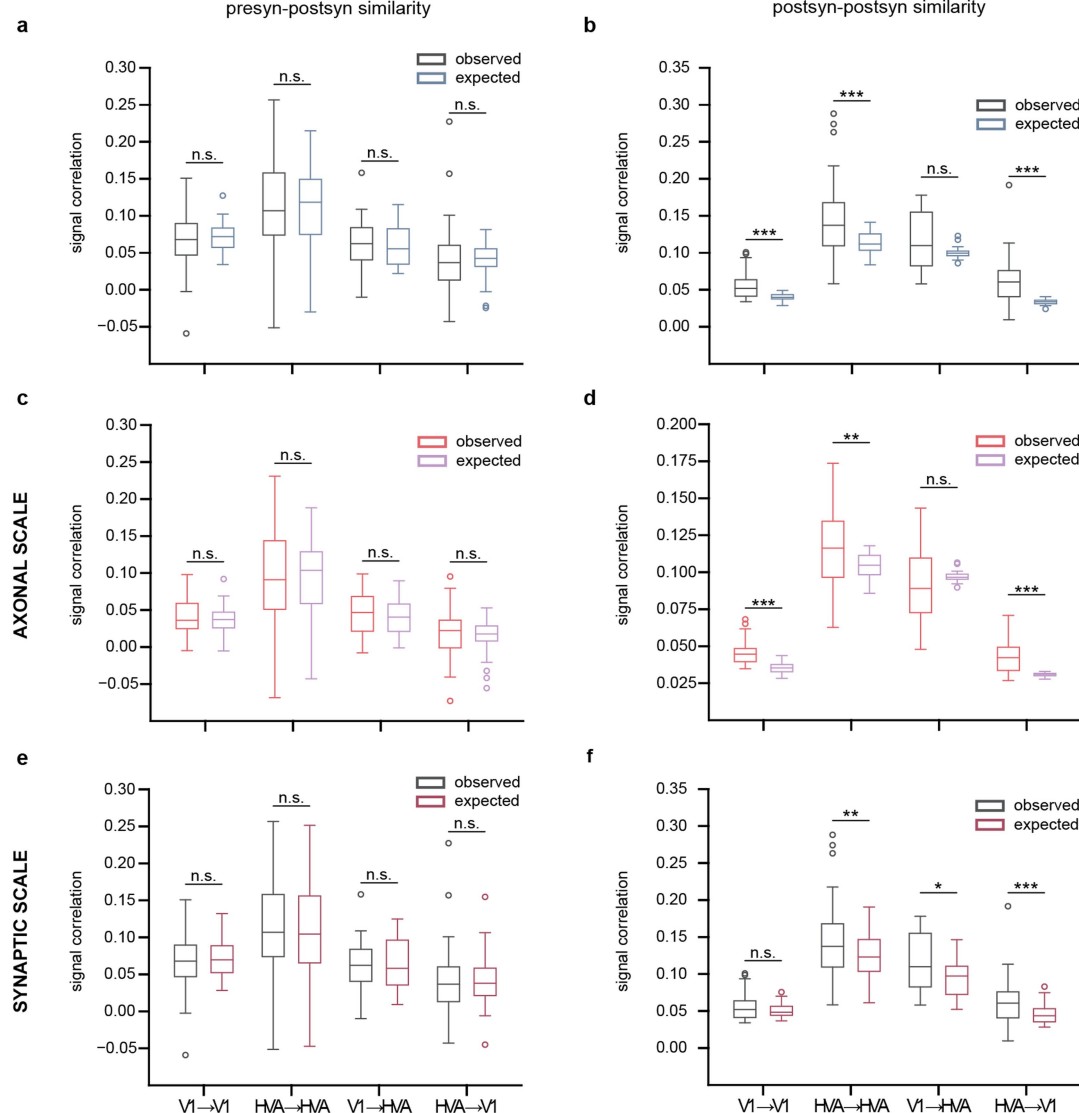

**Extended Data Fig. 14 | Postsynaptic neurons with a common input are more similar to each other than expected by a pairwise like-to-like rule at both axonal and synaptic scale. a**, Mean pre-post signal correlations in the data (dark gray, "observed") and the model (blue, "expected") are not significantly different, indicating that the model reproduces the expected pairwise like-to-like rule **b**, Mean pairwise in silico signal correlation of postsyns, reproduced from Fig. 5c. The observed data shows significantly higher postsyn to postsyn similarity than predicted by the model fit with only a pairwise rule, for three out of four projection types. **c**, As in **a**, but at "Axonal" scale. **d**, As in **b**, but at "Axonal" scale. **e**, As in **c**, but at "Synaptic" scale. **f**, As in **d**, but at "Synaptic" scale. For statistics and sample sizes, see Supplementary Tables 21, 22.

**a**

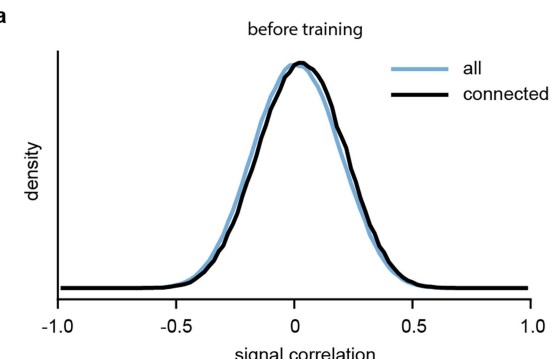

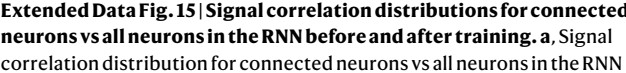

**b**

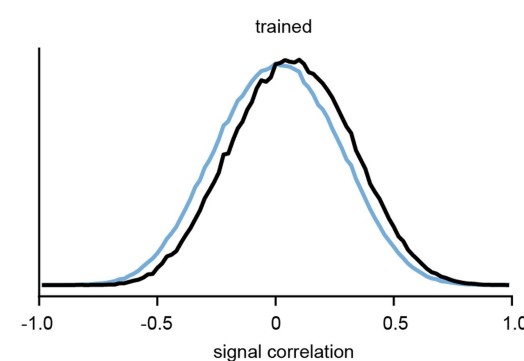

**Extended Data Fig. 15 | Signal correlation distributions for connected neurons vs all neurons in the RNN before and after training. a**, Signal correlation distribution for connected neurons vs all neurons in the RNN before training. A neuron pair was classified as connected if the associated weight was in the top 35th percentile of all weights. **b**, Same as **a** except after training.

**Extended Data Table 1 | Number of presynaptic neurons broken down by projections**

| Postsynaptic brain area / Presynaptic brain area | V1 | RL | AL | LM |
|---|---|---|---|---|
| V1 | 36 | 29 | 19 | 6 |
| RL | 59 | 66 | 39 | 6 |
| AL | 35 | 35 | 35 | 5 |
| LM | 0 | 0 | 0 | 0 |

**Extended Data Table 2 | Number of synapses broken down by projections**

| Postsynaptic brain area<br>Presynaptic brain area | V1 | RL | AL | LM |
|---|---|---|---|---|
| V1 | 3208 | 563 | 183 | 9 |
| RL | 1215 | 1953 | 423 | 7 |
| AL | 384 | 685 | 403 | 5 |
| LM | 0 | 0 | 0 | 0 |

# Reporting Summary

## Statistics

For all statistical analyses, confirm that the following items are present in the figure legend, table legend, main text, or Methods section.

| n/a | Confirmed | |
|---|---|---|
| ☐ | ☒ | The exact sample size (*n*) for each experimental group/condition, given as a discrete number and unit of measurement |
| ☐ | ☒ | A statement on whether measurements were taken from distinct samples or whether the same sample was measured repeatedly |
| ☐ | ☒ | The statistical test(s) used AND whether they are one- or two-sided *Only common tests should be described solely by name; describe more complex techniques in the Methods section.* |
| ☐ | ☒ | A description of all covariates tested |
| ☐ | ☒ | A description of any assumptions or corrections, such as tests of normality and adjustment for multiple comparisons |
| ☐ | ☒ | A full description of the statistical parameters including central tendency (e.g. means) or other basic estimates (e.g. regression coefficient) AND variation (e.g. standard deviation) or associated estimates of uncertainty (e.g. confidence intervals) |
| ☐ | ☒ | For null hypothesis testing, the test statistic (e.g. *F*, *t*, *r*) with confidence intervals, effect sizes, degrees of freedom and *P* value noted *Give P values as exact values whenever suitable.* |
| ☒ | ☐ | For Bayesian analysis, information on the choice of priors and Markov chain Monte Carlo settings |
| ☒ | ☐ | For hierarchical and complex designs, identification of the appropriate level for tests and full reporting of outcomes |
| ☐ | ☒ | Estimates of effect sizes (e.g. Cohen's *d*, Pearson's *r*), indicating how they were calculated |

*Our web collection on statistics for biologists contains articles on many of the points above.*

## Software and code

Policy information about availability of computer code

| Data collection | For image acquisition, we used ScanImage (2017b). Stimuli were presented using PsychToolBox (3). The data collection process was automated with Labview (2016). |
|---|---|
| Data analysis | We used DeepLabCut (2.0.5) for automatic tracking of the pupil. We used CaImAn (1.0) for segmentation / deconvolution of calcium imaging data. Meshparty (1.16), NEURD (1.0.0), pcg_skel (0.3, 0.2) and Neuroglancer (https://github.com/seung-lab/neuroglancer) were used for morphology analysis and visualization. Our custom built analysis pipeline (https://github.com/cajal/pipeline, developed in Matlab (2016a, 2018b), Python (3.6, 3.8), and R (4.3.3)) also used general tools like Numpy (1.23.5), pandas (1.5.3), SciPy (1.10.1), PyTorch (1.12.1), Matplotlib (3.7.0), seaborn (0.12.2), HoloViews (1.15.4), Ipyvolume (0.5.2), tidyverse (2.0.0), Jupyter (ipykernel: 6.21.2), MySQL (5.7.37), Docker (23.0.1), and Kubernetes (1.22.11). We used statsmodels (0.13.5), scikit-learn (1.2.1), glmmTMB (1.1.10), performance (0.12.2), emmeans (1.10.3) for statistical analysis. DataJoint (0.12.9) and CAVE (4.12, 4.14, 4.16) were used for storing and managing data. |

For manuscripts utilizing custom algorithms or software that are central to the research but not yet described in published literature, software must be made available to editors and reviewers. We strongly encourage code deposition in a community repository (e.g. GitHub). See the Nature Portfolio guidelines for submitting code & software for further information.

## Data

Policy information about availability of data

All manuscripts must include a data availability statement. This statement should provide the following information, where applicable:

- Accession codes, unique identifiers, or web links for publicly available datasets
- A description of any restrictions on data availability
- For clinical datasets or third party data, please ensure that the statement adheres to our policy

All MICrONS data are released on BossDB (https://bossdb.org/project/microns-minnie, please also see https://www.microns-explorer.org/cortical-mm3 for details).

## Human research participants

Policy information about studies involving human research participants and Sex and Gender in Research.

| | |
|---|---|
| Reporting on sex and gender | N/A |
| Population characteristics | N/A |
| Recruitment | N/A |
| Ethics oversight | N/A |

Note that full information on the approval of the study protocol must also be provided in the manuscript.

# Field-specific reporting

Please select the one below that is the best fit for your research. If you are not sure, read the appropriate sections before making your selection.

☒ Life sciences ☐ Behavioural & social sciences ☐ Ecological, evolutionary & environmental sciences

For a reference copy of the document with all sections, see nature.com/documents/nr-reporting-summary-flat.pdf

# Life sciences study design

All studies must disclose on these points even when the disclosure is negative.

| | |
|---|---|
| Sample size | No sample-size calculation was performed a priori. Sample sizes (number of connections tested) match or exceed previous studies of similar design. |
| Data exclusions | Of the 14 released MICrONS scans, one scan was excluded a priori from the study due to experimental issues (responses to some stimuli were not collected due to water running out from the objective). Neurons that did not pass the pre-established functional thresholds described in the paper were excluded from the analysis in order to only compare functional properties in neurons that were well characterized. |
| Replication | Due to the cost and time involved in producing the MICrONS volume, a second volume is not yet prepared to allow reproducibility testing. All available proofreading in the existing volume was used in order to increase power, especially where the number of unique presynaptic neurons was the limiting factor. |
| Randomization | No randomization is performed since our study did not include multiple predefined experimental groups for sample allocation. Instead, each sample was controlled with a matched control population ("same region" and "ADP" controls, as described in the study) with matched anatomical properties at the appropriate scale. |
| Blinding | No blinding is performed during data collection since our study did not include predefined experimental groups for sample allocation. Manual annotation of the data is blinded to the functional properties of the neuron. The analysis is performed unblinded, however, the same process is applied to all control and sample groups. |

# Reporting for specific materials, systems and methods

We require information from authors about some types of materials, experimental systems and methods used in many studies. Here, indicate whether each material, system or method listed is relevant to your study. If you are not sure if a list item applies to your research, read the appropriate section before selecting a response.

## Materials & experimental systems

| n/a | Involved in the study |
|---|---|
| ☒ | Antibodies |
| ☒ | Eukaryotic cell lines |
| ☒ | Palaeontology and archaeology |
| ☐ | ☒ Animals and other organisms |
| ☒ | Clinical data |
| ☒ | Dual use research of concern |

## Methods

| n/a | Involved in the study |
|---|---|
| ☒ | ChIP-seq |
| ☒ | Flow cytometry |
| ☒ | MRI-based neuroimaging |

# Animals and other research organisms

Policy information about studies involving animals; ARRIVE guidelines recommended for reporting animal research, and Sex and Gender in Research

| Laboratory animals | For experiments excluding the MICrONS dataset in this manuscript: ten mice, (Mus musculus, 3 female, 7 males) 78-190 days old at first experimental scan. Heterozygous for both Slc17a7-Cre (B6;129S-Slc17a7tm1.1(cre)Hze/J, Jackson Laboratory Strain # 023527) and Ai162 (B6.Cg-Igs7tm162.1(tetO-GCaMP6s,CAG-tTA2)Hze/J, Jackson Laboratory Strain # 031562). The MICrONS dataset was collected from a mouse of the same species and strain, 75 days old. |
|---|---|
| Wild animals | Study did not involve wild animals. |
| Reporting on sex | For new experiments in this manuscript: 3 Female, 7 Males. For MICrONS dataset, 1 Male. Animals were randomly recruited to the study with respect to sex. Analysis disaggregated for sex was not performed, due to low sample size and expected generalization of principles under study across genders. |
| Field-collected samples | Study did not involve samples collected from the field. |
| Ethics oversight | All procedures were approved by the Institutional Animal Care and Use Committee of Baylor College of Medicine. |

Note that full information on the approval of the study protocol must also be provided in the manuscript.

