## [Peer Review File · Nature]

Functional connectomics reveals general wiring rule in mouse visual cortex

Corresponding Author: Professor Andreas Tolias

Version 0:

Reviewer comments:

Referee #1

(Remarks to the Author)

The paper titled “Functional connectomics reveals general wiring rule in mouse visual cortex” by Ding et al. is part of a series of papers derived from the MICrONS Consortium dataset (released on bioRxiv in 2021), which is obtained in an important large-scale effort to correlate anatomical and functional connectivity in the mouse cortex. Despite its limitations (e.g. including only a single animal and cells dropping out due to manual proofreading), the MICrONS Consortium dataset is unprecedented in its scale. The subsequent analyses, of which the manuscript presents one part, are providing the community with insights into which information one can gain by expanding correlative EM and functional imaging to such large scale, and which types of questions can be addressed. The overall dataset is remarkable, and the present manuscript focuses on a specific part of the data analysis. It studies which general wiring rules of mouse visual can be derived from the MICrONS dataset.

On the positive side, it expands the question how functional similarity correlates with anatomical connectivity – a question that had been previously only studied for short-range connections – to long-range anatomical connections.

On the negative side, the interpretability of the data is limited by both methodology and presentation of the data. Most importantly the authors do not show any conclusions backed up by measured data but only by a model which has not yet been peer reviewed. The authors should emphasize in the title of the paper that this is a modeling study.

In my view this paper is only relevant for a high profile publication if the authors show significant correlations in the measured data and not only in the digital twin model. The twin model can be used to strengthen and support data but not to replace data. Further, either before the publication of this manuscript the digital twin-model should be peer reviewed, or it should be part of this manuscript and reviewed here.

Major comments:

(1) All analysis of functional data is performed on the “digital twin”, a Neural network trained to reproduce empirical spiking responses. This choice is motivated by the fact that a “larger stimulus space” can be explored than in the recorded data. However, this approach limits interpretability and is lacking controls.

(1a) Issues with generalization are a known weakness of neural networks. Generalization errors (e.g. due to differences in the stimulus parameters between the test/training stimuli compared to the stimulus space used on the digital twin) should be estimated and taken into account.

(1b) After training, the model predicted responses with >0.35 correlation for the ~37% of selected cells. This means that up to 88% of variance are unexplained even for selected cells, and ~63% of cells were not included. It is therefore unclear, how well the digital twin reflects the physiological responses. At the minimum, the analysis must be repeated on empirically measured signal correlations, including cells on which the digital twin did not perform well, and the data must be presented in parallel. Deviations from the digital twin should be quantified and appropriately discussed.

(1c) The authors mention that the “relationships we describe here, while statistically significant, have an apparently small effect size”. Given the extremely small effect size (difference in signal correlation of 0.02!), the reader would like to see what

the difference in activity between pairs of neurons with these different correlations would look like. Also, the reader wonders if this small effect size is relevant for brain computations.

(2) The abstract presents as a main result that “we did not find that orientation preference predicted connectivity”. Given the multitude of caveats interpreting this piece of data (suboptimal stimulus used, low number of selected cells, low threshold orientation tuning threshold for selection, analysis based on digital twin not empirical responses), this emphasis does not seem appropriate. Indeed, misleading, and incorrect.

422ff: When presenting the data on orientation preference, the visual stimulation paradigm (drifting noise, not drifting gratings) needs to be spelled out.

Further the authors should use criteria for orientation selectivity (>0.4) that was used before in Ko et al. 2011.

(3) Since the main reference (MICrONS Consortium et al. 2021) is only published in preprint up to date, any relevant methodology should be described in the present paper.

(4) The authors use the concept of “feature weights”, which are defined entirely from the artificial neuronal network. The authors show how this metric corresponds to a biologically relevant feature.

(5) The authors should emphasize from the beginning that all results are derived from a model based on recordings from a single animal.

(6) Terminologies and abbreviations need to be defined:

200f: “inappropriate merges” not defined

227f: CCmax and CCabs not yet defined

328ff and 996: “breakout analysis” not defined

337: “center distance” not defined

345: “projection breakout” not defined

(7) Overall, sentences are often hard to understand due to missing information or logical flow:

340: “is selected” unclear what this means

335: Paragraph title difficult to understand

544ff: it is not sufficiently explained what is a feature and why it is less “long-range”. In general (already in the main text), it should be explained, to which feature space the “feature weights” refer to

558ff: argumentation unclear. Why could a genetically determined connectivity not also result in higher signal correlations?

581f: “it may result in an apparently independent feature weight similarity contribution” – not clear why

(8) Statistical issues

Benjamini-Hochberg correction assumes independence between tests – which seems violated due to individual datasets appearing in multiple comparisons. Please explain why the method is applicable or replace it by an appropriate method.

(9) Causality is reversed in arguments

277ff: the authors build their arguments analysis on the assumption that “signal correlation affects connection probability” – but in fact, causality is reverse. The logic should be reversed.

441ff: “Signal correlation and feature weight similarity independently contribute to connectivity” – the causality again is reverse. This should be clarified. The model should be phrased in terms of posthoc likelihood, not as “contribution”, which can be understood as mechanistic contribution.

Minor comments:

197: Does the sentence imply that only axonal, not dendritic processed were proofread? Please clarify in more detail.

Please indicate error rates of automated detection versus proofreading (also for line 640)

Fig 3g: the difference red vs black might be non-significant, but is quantitatively similar to blue vs black

354: RF center distance does not correlate with synapse volume: could this relationship be lost in the digital twin because of a disturbance of the RF map (e.g. filtering effect)

856ff: “Trials with more than 10 consecutive untracked pupil frames were discarded (18-180 trials per scan, 2-39%).” – why?

Referee #2

(Remarks to the Author)

This manuscript begins to deliver on the promise of high-density connectional data combined with functional characterization that can be afforded by 3-D EM reconstructions. While this is a very important manuscript and an impressive effort, one can't help but feel a bit disappointed by what is delivered. The most positive aspect is that principles of connectivity previously described only for local connectivity within and across layers in mouse primary visual cortex are now extended to intra-areal feedforward and feedback connections. These connections, like local connections are more frequent for excitatory neurons with similar functional properties. But if what we get is mostly confirmation of what we already know, then what's all the fuss about? Why do we need to spend more and more money on larger and larger EM volumes?

There are several things that are not included but might have been included to make the manuscript stronger. First, there is no analysis of second-order (or nth order) contingencies of connectivity. For example, if two neurons are connected to each other, are they more or less likely to have other inputs or outputs in common? And it is disappointing that inhibitory neurons

are entirely excluded from the analysis, even if they were not functionally characterized. (Why weren't they?) Do inhibitory neurons follow the same rules of connectivity as excitatory neurons? Are these the same for inhibitory neuron types that target cell bodies (basket cells), axons (axo-axonic), dendritic spines and shafts, or other inhibitory neurons? It is understandable that the authors might want to reserve these descriptions for other publications, but then they should not expect that the limited descriptions here, with their relatively limited impact, will be a good fit for a high-profile journal. The authors do acknowledge all of this in their discussion and state: "Ongoing investments in proofreading, matching, and extension efforts within this volume will have exponential returns for future analyses as they yield a more complete functional connectomic graph. There is much more to discover about this relationship from this dataset, and others like it that are currently in preparation." So perhaps the bottom line is that it is just not quite there yet.

Overall the manuscript is very well written, the logic clear and the citations to the relevant literature are mostly appropriate. There are a few previous publications that probably should have received a bit more prominent mention in the introduction and discussion. Previous studies (Wertz et al.; Rossi et al.) identified and functionally characterized very large numbers of local presynaptic neurons providing direct input to a small number of postsynaptic neurons – hundreds of inputs per postsynaptic cell. These numbers are far greater per postsynaptic neuron than described here, where the numbers of presynaptic cells are only one-tenth the numbers of postsynaptic cells. And these included cells sampled across cortical layers (albeit always with the postsynaptic cell in layer 2/3). The Introduction disposes of this by simply stating "and typically limited to cortical Layers 2 and 3 (L2/3)." And the Discussion makes no mention of these prior studies or how the results relate to the present study. Discussion of this would seem to merit more attention than the long description of methods for estimating receptive fields, which are apparently already described in detail in the Wang et al. Biorxiv paper. For example, Only the Ko et al. data are compared to the EM data with respect to orientation selectivity. But Wertz et al. has data for hundreds of input neurons with far greater statistical power.

Other points:

In the discussion: "This may be due to practical differences, such as the parametric stimulus used to characterize orientation tuning (in silico drifting noise with orientation coherence in our study versus drifting gratings), cell selection criteria ($gOSI > 0.25$ in our study versus $OSI > 0.4$ in Ko et al. 2011)," Why not simply change the criterion to address this? Or is the sample size too small for well-tuned cells? It appears that the OSI distributions in the Wang et al. Biorxiv might be generally low. Perhaps this is not a good method for assessing OSI?

Fig 4 of the Wang et al BioRxiv paper plots the accuracy with which the digital twin model is able to predict the orientation tuning of a neuron's response to gratings based on responses to natural scenes. This might be added as a supplemental figure so that the reader can understand the reliability of the orientation predictions. A reader should not have to refer to another manuscript to obtain this crucial information.

All of the data describing significant positive correlations between connectivity and function are highly processed, leaving one with very little insight into actual observed rates of connectivity. Please provide basic information about the raw data used to calculate fold changes in connection probabilities (e.g. Fig. 2f-g). How many pairs of neurons for each type of comparison were present as both ADP control values and as actual connectivity values? For example state: "For local connectivity within V1, xx presynaptic and xx postsynaptic cells were assessed. Of these cells there were xx actual connections and xx ADP control contacts. XX of the xx actual connections (xx%) were for neurons with similar function (Threshold >0.3 correlation) versus xx that had different functions. This contrasts with xx% of neurons with similar function for ADP contacts. For local connectivity within HVA, xxxx. For feedforward connections from V1 to HVA For feedback connections from HVA to V1" Perhaps these values are somehow embedded in Supplemental Table 4. If so, then please clarify and add relevant detail to the table legend.

Referee #3

(Remarks to the Author)

Being able to relate the activity patterns of neurons within a network with their exact synaptic connectivity promises new insights into the basic operational principles of the brain. Currently, vast research efforts in neuroscience are invested in such projects in different model organisms. Here, Ding, Fahey, and Papadopoulos et al. utilize their previously published, openly available MICrONS dataset and present a first analysis of a subnetwork spanning all layers and several visual cortical areas in the mouse. Their final connectivity matrix consists of 122 excitatory presynaptic neurons that connect with ~2000 excitatory postsynaptic neurons with known functional properties.

The authors analyze the connection probability between neurons, the number of synapses formed, and the synaptic cleft volume for all connections they detect and ask how these features relate to different components of the visual responses. Additionally, they introduce two anatomical control measurements that allow for more in-depth analysis of connection specificity. The authors report that neurons that respond to similar visual features are more likely to be connected, which is supported by previous work. Surprisingly, neurons with similar orientation tuning and neurons whose receptive fields are spatially close do not show a tendency of having a higher connection probability. The authors conclude that previously reported like-to-like connectivity patterns generalize across cortical layers and visual areas.

The combination of methods at this scale is an impressive technical tour-de force effort. The openly available MICrONS dataset promises to become an essential community resource for systems neuroscience. With the current study, the authors are now pioneering analyses tools to test circuits neuroscience questions in their dataset. Therefore, I expect that this study will receive much attention across the neuroscience community and beyond, and it has the potential to become a landmark paper. The paper is well written, and the methods and statistics seem to be robust.

In summary, I am quite enthusiastic about the work, however, I have several questions regarding the analyses, the use of the digital twin model, and potential experimental biases. Also, given the general readership of Nature, I think the paper could

be clearer in some sections.

In detail:

1) It is a strong point of this work that the authors measure the connection probability between neurons, the number of synapses formed, and the synaptic cleft volume of all synapses. The latter is directly related to the postsynaptic density area and therefore the functional strength of the synapse. So, in principle, the authors have a complete picture of the anatomical connectivity between all neurons, which is remarkable and quite unique! However, they only analyze these anatomical features separately, which, in my opinion, is a missed opportunity. To me, the most relevant anatomical readout (the feature that should predict best how strongly one neuron drives another neuron) is the total anatomical synaptic weight between these cells. In the case of this study, this would be the sum of the synaptic cleft volumes of all synapses in a connection. The reason why I think this is important is because by analyzing each anatomical feature separately, the authors could miss if other features 'compensate'. For example, some connections might form only very few, but strong synapses (i.e., synapses with large synaptic clefts). Wouldn't this show up as a weak connection in the analyses? Conversely, a connection with many weak (small) synapses would show up as a strong connection with high connection probability. Intriguingly, the authors report that similar orientation tuning and spatial receptive field features were not associated with higher connection probability. By contrast, Cossell et al. (2015) reported that connection strength measured with slice ephys predicted similar orientation tuning and spatial receptive field overlap. I am wondering if the authors could also uncover similar effects to what Cossell et al. reported if they computed a combined anatomical synaptic weight index. Therefore, I am not convinced that mostly looking at connection probability by itself is the best way to analyze these remarkable data.

2) The analyses rely heavily on the digital twin model that the authors create. If I understand correctly, the digital twin is a way to 'bootstrap' new functional responses for neurons that couldn't all be acquired experimentally, which allows for a more in-depth correlation analysis. With the general readership of Nature in mind, I think more could be done in the paper to explain and justify the model. Just as an example, only in line 260, the authors finally provide a real rationale for the digital twin. This seems to me a relatively unconventional method, and it should probably be explained better in the introduction. Moreover, when talking about the digital twin, the authors frequently reference Wang et al. (2023), which appears to be the unpublished methods-companion paper. Because this model is unpublished, I find it hard to judge the validity of this approach and the authors could do more to convince the readers that this modeling approach produces 'correct' neuronal responses that could also be measured experimentally. The authors claim that this is indeed the case (line 123f), but while the model responses in SFig. 4 c seem to match overall; they are still a bit different from the original in vivo cell responses. As an additional control, for example, would some of the results also hold if only 'real', i.e., experimentally recorded responses were included in some of the analyses?

3) Have the authors assessed during the proofreading stage how completely their neuronal morphologies could be reconstructed? In my experience with such EM datasets, sometimes an axonal or dendritic branch can become untraceable in the tissue (due to fixation issues, cracks, folds, etc.). For the neurons reported in this study, (how often) was this the case and do the authors think it might have affected the results? For example, are long-range projections more likely to be incomplete? Moreover, if I understand correctly, the authors suggest that local axon branches might be proofread less reliably because emphasis was given to long-range projections (line 274f). Maybe the authors could provide some (ideally quantitative) estimate that allows readers to judge how often a local (and long-range) branch might be incomplete?

4) I didn't find any mentioning that synapses were manually proofread, too. Have the authors assessed how reliable their synapse detection algorithm is? Specifically, I think there are two important questions:
(1) what is the overall false-negative and false-positive rate? Some estimates should be reported. The authors provide some estimates for their L2/3 dataset in Turner et al. (2020), but are these numbers also correct for the more heterogeneous Cortical mm³?
(2) More importantly, are there any systematic biases for missing certain types of synapses? Just as an example, are asymmetric synapses on dendritic shafts detected with equal likelihood as asymmetric synapses formed on dendritic spines? Systematic biases might affect certain connections more than others and could bias the results. Along those lines, I noticed in Fig. 4 that the canonical circuit motif of recurrent connections in L4 in V1 appears surprisingly weak in all panels. Could the authors clarify if this is in fact unexpected or if I'm missing something?

5) Previous work could be cited more. While the authors mention previous smaller-scale structure-function studies in mouse visual cortex, they don't mention some of the efforts of relating calcium imaging and EM in the same animal in other model organisms, which are in fact cited in their 2021 MICrONS paper.

6) 277f "To measure how signal correlation affects connection probability compared to either same region or ADP control, we quantified the fold changes in connection probability as a function of signal correlation."
I find this confusing because it suggests to me that connection probability was compared between the connected pre- and postsynaptic neuron and the controls. But don't the controls have a connection probability of zero per definition? So presumably, what was compared were the signal correlations between connected neurons and unconnected neurons? Also, wouldn't it be more logical if the sentence would say: "To measure how connection probability affects signal

correlation...”?

I also find the associated figures (Fig. 2 f, g and Fig. 3 c, f, i) difficult and unintuitive to interpret. I'm not really sure what exactly is compared here. My interpretation is this: to get the red plot for example (Fig. 2 f), the authors pooled all connected neurons and all ADP controls together in one population. Then they asked: for the different bins of signal correlations in this pooled population, what was the probability that cells were connected. Is this correct? In any case, this analysis should be explained in more detail.

7) In their introduction, the authors motivate their structure-function study in visual cortex by citing Hubel and Wiesel: how might center-surround receptive fields from the LGN give rise to edge detectors in V1? However, this question cannot be tested in their dataset because the thalamic afferents are not included. I guess this reference makes sense if the authors just want to highlight the general importance of the approach, but it might also be considered a poorly chosen example because it can't be addressed here. In any case, this shows that the field is still at the beginning of circuit-level structure-function analyses and the authors could further increase the impact of their work by highlighting the importance of, and explicitly inviting future studies in rodents and also in other neuroscience model organisms such as the fruit fly and zebrafish.

Minor points:

First, the authors introduce the digital twin model and explain that they can separate functional tuning into a 'feature component' and a 'spatial component', but then their initial analyses all use 'in silico signal correlation' as a metric, which is not explicitly explained until the discussion. Could the authors explain the differences between these metrics in more detail and explain their rationale for using one or the other?

241 – 253 I feel this is an important paragraph, but it is difficult to understand the logic here. Maybe the authors could try to rephrase or simplify?

469 The abbreviations 'SC' and 'FW' are never formally introduced.

609-615 This sentence is hard to understand, maybe 'even' should be 'evenly'?

858 Could the authors provide the reader with a short rationale for why the model architecture was changed with respect to Wang et al.?

875 Could the authors report whether certain types of neuronal tunings were better modeled by the digital twin than others? In that case, the selection criteria reported here would lead to a dataset in which some response types would be underrepresented, which would be important to know.

Referee #4

(Remarks to the Author)

Ding, Tolia and their many colleagues investigated the relationship between synaptic connectivity and visual function using the large-scale EM MICrONS dataset combined with computational modeling. The “digital twin model” uses a state-of-the-art deep recurrent neural network to innovatively capture single-neuron responses to calculate feature weights and receptive fields, enabling a refined view of functional similarity. Their approach is poised to reveal fundamental rules governing cortical wiring and information processing. With this combination of data and tools the authors set out to either confirm a relationship between functional similarity and connectivity, establishing principles for cortical organization or undermine the link to orientation tuning, challenging existing theories of visual cortex wiring.

Briefly, the authors found neurons with correlated signals tend to connect and that connectivity depends more on feature weight than receptive field similarity; surprisingly, orientation tuning did not predict connectivity. They claim these relationships generalize across layers and areas, though some lack significance. A model found feature and signal correlations independently contribute to connectivity. Overall, this work highlights how in silico modeling and connectomics at scale provide mechanistic insight into population dynamics emerging from heterogeneous single-cell interactions.

While the extraordinary scale and resolution of the data in this study could yield groundbreaking insights into visual cortical organization, the current manuscript does not fully realize this potential. The most compelling finding is that connectivity depends on feature, not orientation, similarity—challenging notions that it relies on any one parameter and suggesting complex rules govern network architecture. Curiously, most relationships reported have very small effect sizes; I found this particularly surprising. One interpretation is simply that the authors, like the rest of the field, have been trying to explain connectivity with the wrong functional correlates. Why should all synapses reflect the simple visual features and not more complex behaviorally-related computations in visual cortex? Interpreting modest effects as averages of diverse, nonlinear interactions across cells could inspire new theoretical frameworks. Finally, perhaps due to the complexity, the major issues limiting this paper's impact are the lack of coherence and clarity in presentation; failure to reconcile uncertainties and using generative models of visual processing to tie findings together. Nevertheless, this work provides a glimpse into the potential of systematically combining functional modulation and EM reconstruction at synaptic resolution. Addressing these issues

could significantly strengthen the study overall and establish whether its scientific contributions are important for a broader audience.

Major comments:

1. The analyses are based primarily on digital twin model to test the relationship between neuronal tuning similarity and synaptic connectivity. The use of digital twin model to factorize neuronal tuning into feature and spatial components indeed provides us a new view of the neuron's functional properties. These model-inferred functional metrics are exciting but also require further validation and assessment of model variations and uncertainties. Feature weight is potentially a more comprehensive characterization of a neuron's tuning properties than metrics that depend on specific parametric stimuli, such as orientation tuning, and it can be generalized to natural and complex stimuli; on the other hand, compared to signal correlation, similarity in feature weights may be a better characterization of tuning similarity between neurons.

However, one may need to be careful when drawing conclusions about the relationship between neuronal tuning similarity and synaptic connectivity with those *in silico* generated metrics. Although *in vivo* and *in silico* functional metrics such as receptive field location and orientation selectivity are well correlated, the correlation between certain *in silico* functional metrics and synaptic connectivity cannot be directly extended to their *in vivo* counterparts, and the same applies to lack of correlation. For the four *in silico* functional metrics used in this study, feature weight must be derived from the digital twin model, but the conclusions about the "like-to-like" rules regarding signal correlation, receptive field location, and orientation selectivity should be validated with *in vivo* metrics too.

2. This study does not find a relationship between synaptic connectivity and orientation selectivity, which contradicts with Ko et al., 2011 and Lee et al., 2016. In the discussion section, the authors explain practical differences that might have led to different conclusions of this study of previous ones: parametric stimulus to characterize orientation tuning, cell selection criteria, and the location and size of the area being studied.

Those claims make sense, and especially the location of the area being studied is a very likely reason, since different parts of the visual area may follow different connectivity rules. Nevertheless, as discussed in the previous point, the conclusion was drawn with *in silico* orientation selectivity, and it would be more decisive to use *in vivo* orientation selectivity to validate this conclusion. Alternatively, the same method of using *in silico* functional metrics to investigate "like-to-like" rule should be tested with the previous functional-EM dataset used in Lee et al., 2016, if possible, to see if the same conclusion can be drawn as this previous study. This can be used to rule out the possibility that any intrinsic flaws in modeling caused the discrepancy with previous studies in the conclusion about connectivity rules and make the practical reasons suggested by the authors more likely.

3. The relationship between functional metrics and synaptic connectivity is quantified by the "fold change of connection probability", which is the connection probability within each bin of a functional metrics normalized against the overall connection probability across all bins. The term "fold change" can be confusing, as it is actually a normalized probability, but "fold change" indicates the "change" relative to a baseline instead. A term such as "normalized connection probability" might be more straightforward and consistent with its true meaning.

4. Failure to consider connectivity relative to potential synapses limits evaluation of how much observed connectivity exceeds chance. Instead of calculating the connection probability relative to ADP control, it would be a more straightforward and critical quantification to calculate a metric like the total synaptic number or area normalized by total potential synaptic length (as used in Lee et al., 2016). I believe this would offer a better metric of how synaptic connectivity is selected on top of axon trajectory – which is presumably more closely related to early development. This analysis could potentially reveal new relationships between functional properties and connectivity.

5. The effect sizes for most relationships found seem small. In other words, separately from the other quantification concerns I was struck by the how little connectivity corresponds to functional metrics. Perhaps my intuitive expectations are incorrect and maybe generative computational models would also predict this. Alternatively, the small effect sizes point to the possibility that the visual metrics used, despite their innovativeness going beyond classical measures, still miss the functional responses. Unfortunately, the MICrONS dataset does not include behavior, a major missed opportunity to go beyond orientation selectivity. In any case, if the authors believe this is a concern, I would suggest that they are explicit about the potential that most synapses don't match features simply because visual features are only one of many relevant computations in visual cortex.

6. The evaluation of goodness of fit in the logistic regression model

The authors used McFadden's pseudo-R-squared as a measurement for the goodness of fit in Fig. 5e. Although this is indeed a way of evaluating how well the regression is, the very small value of pseudo-R-squared makes the effect size unclear for comparing the full model to the reduced models excluding one of the predictive variables; also, it raises concern how well this model is actually able to predict synaptic connectivity, or in other words, how critical those predictive variables are for predicting synaptic connectivity.

7. The cell inclusion criteria based on how well the neurons can be predicted by the digital twin could induce biases.

The methods state that only neurons that had reliable visual responses and were accurately predicted by the digital twin model were used for analyses. "122 out of 152 presynaptic neurons" but only "1975 out of 5502 postsynaptic neurons" passed this criteria. Although clearly this is a necessary step given that the main analyses of this study are based on *in silico* characterization of neurons, it raises concerns that neurons not well predicted by the model might exhibit different

connectivity rules from the ones included in the analyses, and therefore, the cell inclusion criteria based on the model prediction performance may introduce biases. This also emphasizes the necessity of validating the conclusions drawn with in silico metrics with their in vivo counterparts and the full neuron sample.

8. These findings would benefit from putting them in the context of major existing theories to help convey their meaning and importance. The introduction section has long tangents on the history of neuroscience and various connectivity theories that are not directly relevant to the study results. Going back to Hubel-Wiesel would be warranted if their initial ideas are either confirmed or refuted, which is not the case here. The Discussion section could point to concrete examples as to how their findings substantively advance theoretical understanding or inspire new questions; these less abstract discussions would bring together these results and make their importance clearer.

Minor issues:

1. The animal used to collect the MICrONS dataset was head-fixed but awake and behaving during two-photon imaging. It might be better to describe in some details how exactly the animal was set up and was able to behave at the beginning of the results section, since behavioral state can greatly affect the responses of visual areas to stimuli.
2. In rows 196-201, it was described that “Visually responsive and well characterized neurons in retinotopically matched areas in V1 and HVA were chosen for manual morphological proofreading”. Explaining why selecting “retinotopically matched areas” is necessary and how that was performed would be helpful to a broader audience.
3. Could you explain why the postsynaptic neurons greatly outnumber the presynaptic neurons (1975 vs. 122) in this study?
4. In the analyses of this study, there are no statistical tests between ADP and same region control group. Although the major focus is the difference between connected group and the two control groups, the comparison between ADP and same region control will be important for drawing conclusions about the relationship between axon trajectory and certain function metrics. For example, in rows 338-342, “In contrast to signal correlation and feature weight similarity, receptive field location similarity is selected at the axon trajectory level, as evidenced by the leftward shift in receptive field location distance between connected neurons and same region control”. The conclusion that receptive field location similarity is selected at the axon trajectory level should be mainly supported by the significant difference between ADP and same region control (red vs. blue) and insignificant difference between connected neuron pairs and ADP control (black vs. red). Therefore, the conclusion here should be drawn by comparing all three groups instead of only comparing connected neurons and same region control (black vs. blue). Similarly, in rows 378-382, the conclusion that orientation tuning is selected at the axon trajectory level would be better supported by the comparison between ADP and same region control.
5. In Fig. 3h, after breakdown into specific area-to-area groups, the differences between connected neurons and same region control (black vs. blue) became insignificant, compared to Fig. 3g, where the difference between those two groups were significant. Same in Fig. 4g, h when the neuron pairs were further split into both areas and layers. Is this due to correction for multiple comparison, or due to smaller sample sizes for each group after this breakdown?
6. In rows 367-372, it was described that the “cardinal bias” of orientation tuning of the volume resulted in the U-shaped distribution in the difference in preferred orientation between neuron pairs, as shown in Fig. 3g. It would be helpful to a broad audience to briefly explain what exactly this cardinal bias is.
7. In the figure legend of Fig. 5c, it is not explained what the black arrows from FW and RF to SC mean.

Version 1:

Reviewer comments:

Referee #1

(Remarks to the Author)

Ding et al. have made a number of changes to the manuscript in order to address many of the questions that had been raised. Overall, these changes improved the paper and, generally, they have increased the confidence of the reader in their digital twin as it applies to this work, which was one of the major issues in the original manuscript. Despite this, there are remaining issues, some of them introduced by the modifications to the paper, that the authors need to address.

Major issues:

1. In their response to Reviewers, the authors explain how their proofreading strategy resulted in having a limited subset of functionally characterized neurons that were sufficiently proofread to derive a connectivity graph from. However, this does not come across in the manuscript. It still seems surprising to see the low number of neurons used for the analysis, given the large number that were functionally characterized or reconstructed. It would be helpful to provide some insight to the reader within the manuscript about why the final overlapping number decreases suddenly.
2. The authors should provide more explanation of the digital twin in the manuscript, and provide some insight to the reader about the reason that they use specific implementation of the twin, in their analyses. This is particularly important since they refer to two different models in the methods.
3. Figure 1C provides a useful description of the choice of neurons with which connections are compared to. However, it begs the question: are the authors showing all control neurons? In particular, for the same area controls, one would imagine that same area controls extend far beyond the region of the reconstructed axon (especially for large areas such as V1). It might be helpful to show at least one example of how far these extend.

4. An improvement made to the manuscript is that the authors now also compute the signal correlation entirely from in vivo recordings, as opposed to only from the in silico prediction (Supplemental Figure 3). It induces confidence that the results appear similar in both cases. The authors should do a similar analysis for each result in the paper. This should be included, even if the results may not be significant from the in vivo analysis.
5. In the section titled “Functional similarity is enhanced at both the axonal and synaptic scale”, the authors begin by demonstrating that the in silico model provides an effective model of the data, and end by demonstrating that the in vivo results also show a similar trend as the in silico results. Nevertheless, Supplemental Fig 2 shows that though the in silico model shows higher correlation to the benchmark than the in vivo model, the correlation is still only ~60%. The authors should start with the in vivo results and then introduce the in silico results only following the in vivo results.
6. In the section entitled “Factorized in silico functional representation”, the authors introduce Supplemental Fig 4 to demonstrate that in silico estimates of RF center correlate with estimates made using classical RF mapping stimuli. However, this correlation is not quantified in the text, and – in the figure – the authors demonstrate high r values, but do not discuss the fact that – for two of the three mice – the linear trend has a negative offset. It would be helpful for them to discuss this, and to potentially provide some explanation for the offset. Finally, they should also quantify the qualitative difference in Supplemental Fig 4c.
7. In many previous works on like-to-like connectivity, authors have quantified changes in like-to-like connectivity related to physical distance between neuron somas. Given the dense reconstruction in the MICrONS dataset, the authors could also quantify changes with respect to somatic distance, and see how this differs from prior works. This would provide a mid-way control analysis between their ADP controls, and the same area controls (which can extend much further away from the presynaptic neuron’s axon).
8. The authors use statement such as “selectivity ... exists at the scale of axon trajectories, but not at the scale of individual synapse formation”, based on their comparison of the ADP controls and same area controls. It seems that the more correct statement would be that: there is no further selectivity gained at the synapse level, compared to that obtained at the axon trajectory level.
9. There is a limited characterization of the final result in the paper (section titled “Like-to-like connectivity and its functional role in artificial neural networks”). Although the concept is interesting, it’s not clear how general the authors results are, or if they are specific to some particular training algorithm. If the authors include this result (and if it is important enough to add to their abstract), then they need to explore the space of possible RNN models, and provide some insight either as to why their results apply, as a general principle, or to explain why their results apply to a subset of models, trained in a specific way.

Minor Issues:

1. The authors use the term “lesion studies” in their abstract, to refer to simulated perturbations of their RNNs. “Perturbations” would be a term to their manipulations of an artificial network, given that there is no physical lesion involved.
2. There is a potential confusion in the use of the word “HVA” in the manuscript, since this can refer to a number of different areas. For example, HVA HVA connections could mean connections within a single one of the HVAs, or from one HVA to another. Given this potential confusion, the authors should use more precise wording.
3. It would be helpful to know the distribution of presynaptic neurons (and their projections) in each HVA.
4. The authors explain the axis of Figure 1E and F in the text (particularly, why the y scale extends from negative to positive). It would be helpful for the authors to also include (or refer to) this explanation in the figure legend.
5. The authors include many figures where the x and y axis join at a point that is not the origin. This is a potential source of confusion for readers, and it would be helpful to separate the axis, to ensure that readers know that the axes do not extend to zero.
6. The authors use the word “multi-tiered”, which does not have a clear meaning from the text. This is especially difficult since it is used as a title of one of their sections.
7. In page 5, line 258 – 261: The sentence appears incomplete. It’s not clear what hypothesis the authors are testing.
8. The authors should check the ordering of their supplemental / supplementary figures. It appears that they are not addressed in order in the text, which adds some confusion for the reader. Additionally, they should pick one nomenclature (i.e. either Supplemental or Supplementary) since it’s currently difficult to find all references to extended figures.
9. For Supplemental Fig 4, it would be helpful if the authors would center the axes at the origin (and use square plots) – so that any visual difference from the $x = y$ line would be clearly identifiable.
10. It would be helpful for the authors to also show an example of their same region controls for a layer specific analysis (i.e. similar images as in Supplemental Fig 1, but for Fig 4).
11. The schematic diagram in Fig 5 is hard to understand, as it’s not clear how neurons are placed in a functional space. The authors should clarify this schematic.

(Remarks on code availability)

Referee #2

(Remarks to the Author)

This revised manuscript is much improved relative to the original version in several respects. Most importantly, the narrative more clearly points out the novel insights that can be made from the analyses conducted, and new analyses and revised presentation clarify these important distinctions. Notably, the data and analyses not only show that connectivity/function relationships are generally like-to-like across layers within V1 and across both feedforward and feedback connections, but they also compare specificity at synaptic scale to specificity related to similar proximity without synaptic contacts. These new insights, along with demonstration of the power and future promise of this approach make this manuscript suitable for publication in Nature in the context of a larger package of related manuscripts.

It is noteworthy that reviewers were not notified that this manuscript is being considered in the context of a package of manuscripts, a fact that only became clear from reading responses to reviewers. This is important because the synergistic impact of a set of manuscripts can be much greater than the sum of the parts.

The authors have addressed all of the concerns that I raised in my prior review.

(Remarks on code availability)

Referee #3

(Remarks to the Author)

I find the revised manuscript improved in several ways, the writing is clearer, the story easier to follow, and the additional analyses and controls add to the rigor of the study. The authors show, for example, that the results in Fig. 2 also hold when in vivo data, and not just in silico data were used.

The authors have thoroughly addressed all my comments in their rebuttal; however, more should be done to incorporate the additional information they provide into the manuscript. Following up on my question about the completeness of their reconstructions, I have one more concern regarding an anatomical bias that could arise from how presynaptic neurons were selected.

The arborizations of these pyramidal cells are many 100 μ m in diameter, which means most neurons in the dataset are not truly complete because at least some of their dendrites and axons extend beyond the boundaries of the EM volume. Critically, the closer a neuron's soma sits to a volume border, the more of its axons and dendrites must be truncated. Let's assume a presynaptic neuron sits close to a volume border, then its synaptic connections with its neighbors would be missed if they lie outside the imaged EM volume (on the truncated part of its axon and the truncated dendrites of the postsynaptic neurons). Therefore, wouldn't many of the postsynaptic partners of this presynaptic cell be mislabeled as 'ADP controls' or 'same region controls' in the analyses (e.g. Fig. 2d), even though they might be connected outside the volume? This means that there is a general bias that the synaptic connectivity in the dataset is underestimated. If all presynaptic cells would be in the same column, this bias would be similar for all, but Figure 1d suggests that the locations of presynaptic neurons are relatively scattered across the X-Y plane with some cell bodies as close as 50-100 μ m to a volume border, where a much larger proportion of their arborizations is truncated. Therefore, the extent to which synaptic connectivity is underestimated is heterogenous across the dataset and is expected to be much larger for neurons closer to the borders, compared to neurons in the center of the EM volume.

If cells with similar functional responses are spatially clustered, then such a bias could have the effect that the like-to-like connectivity rate is systematically underestimated for cells close to the volume borders: the connection rate with its nearest (and functionally most similar) neighbors would be underestimated, whereas the connection rate with cells that sit further away and in the center of the volume (and are functionally less similar) is not affected as much. (The opposite would be true for cells in the middle of the volume: their like-to-like connectivity rate might be overestimated.)

The authors mention that nearby neurons do have higher signal correlations (Rebuttal line 637), mouse V1 is retinotopically organized (e.g. Supplemental Figure 4) and there is evidence for (weak) spatial clustering of orientation preference (Ringach et al., 2016, Fahey et al., 2019). Because there is indication for spatial clustering in V1, I wonder if the different degrees of completeness of the presynaptic neurons in the study could affect the results.

Minor:

I appreciate that the authors repeated their analyses using synapse size density (total synaptic weight of connections). The authors should add a half sentence to the Results mentioning this analysis and showing Rebuttal Figure 2 in a supplemental figure – maybe just add it to Supplementary Fig. 8? As I explained initially, I believe it is an important analysis because total synaptic weight is the anatomical correlate that should align best with functional similarity, and it is reassuring that the main findings hold.

The authors have thoroughly addressed my questions about the reliability of the synapse detection and a potential bias for missing smaller versus bigger synapses. However, this should also be addressed in the paper itself, such that readers can assess for themselves what error can be expected in the connectivity graph. More specifically:

- The precision of the automated synapse detection in the volume, which the authors describe in their rebuttal, should be reported in the paper, with a reference to the MICrONS paper.
- Along the same lines, the authors should briefly discuss that there is a bias for missing smaller synapses, and whether they expect this to influence the results.
- The percentages of untraceable endings for dendrites and axons, which the authors mention in the rebuttal, should be reported somewhere in the paper.

The authors now state that the digital twin can "accurately predict the response of a neuron to any arbitrary visual stimulus." (Line 104f). This is a very strong statement, and I imagine it is difficult to justify this experimentally because not all arbitrary stimuli can be presented during an experiment.

By just reading the Results, I find it difficult to understand what exactly is meant by 'signal correlation' (line 259). Only the reference to Cossell et al. in line 262 suggests that 'signal correlation' must be the pairwise Pearson correlation coefficient of two neurons in response to a stimulus. For readability, could the authors provide more information here?

(Remarks on code availability)

Version 2:

Reviewer comments:

Referee #3

(Remarks to the Author)

The authors have addressed all of the remaining concerns that I have raised. I believe the manuscript is suitable and ready for publication in Nature.

(Remarks on code availability)

Referee 1 (Remarks to the Author):

The paper titled “Functional connectomics reveals general wiring rule in mouse visual cortex” by Ding et al. is part of a series of papers derived from the MICrONS Consortium dataset (released on bioRxiv in 2021), which is obtained in an important large-scale effort to correlate anatomical and functional connectivity in the mouse cortex. Despite its limitations (e.g. including only a single animal and cells dropping out due to manual proofreading), the MICrONS Consortium dataset is unprecedented in its scale. The subsequent analyses, of which the manuscript presents one part, are providing the community with insights into which information one can gain by expanding correlative EM and functional imaging to such large scale, and which types of questions can be addressed. The overall dataset is remarkable, and the present manuscript focuses on a specific part of the data analysis. It studies which general wiring rules of mouse visual can be derived from the MICrONS dataset.

On the positive side, it expands the question how functional similarity correlates with anatomical connectivity – a question that had been previously only studied for short-range connections – to long-range anatomical connections.

We appreciate the acknowledgement of the scale of the MICrONS volume and the valuable opportunity it provides to relate synaptic connectivity and function across large distances.

On the negative side, the interpretability of the data is limited by both methodology and presentation of the data. Most importantly the authors do not show any conclusions backed up by measured data but only by a model which has not yet been peer reviewed. The authors should emphasize in the title of the paper that this is a modeling study.

In my view this paper is only relevant for a high profile publication if the authors show significant correlations in the measured data and not only in the digital twin model. The twin model can be used to strengthen and support data but not to replace data. Further, either before the publication of this manuscript the digital twin-model should be peer reviewed, or it should be part of this manuscript and reviewed here.

We now replicate the central findings of Figure 2 with empirical data from the *in vivo* experiments performed in the MICrONS mouse (Supplemental Figure 3). Based on substantial new data and new analyses, we also provide additional validation of the *in silico* model, as we describe below. The digital twin model is undergoing peer review at *Nature* in parallel with this manuscript. We are happy to ask the editors to share those reviews and our responses with you.

Specifics

All analysis of functional data is performed on the “digital twin”, a Neural network trained to reproduce empirical spiking responses. This choice is motivated by the fact that a “larger stimulus space” can be explored than in the recorded data. However, this approach limits interpretability and is lacking controls.

Issues with generalization are a known weakness of neural networks. Generalization errors (e.g. due to differences in the stimulus parameters between the test/training stimuli compared to the stimulus space used on the digital twin) should be estimated and taken into account.

We agree completely that the reliability of our results depends on the predictive model’s performance within the domain of interest (for example, predicting signal correlations and orientation tuning), and we acknowledge that even appropriately cross-validated performance in one stimulus domain does not guarantee generalization to others. To address these concerns, we have significantly expanded our validation of the ability of the digital twin to generalize across stimulus domains. We previously showed excellent alignment between *in vivo* and *in*

silico orientation tuning (now Supplemental Figure 7). We now also include the results of experiments to demonstrate model alignment on two additional metrics: (1) signal correlation calculated from *in vivo* responses to natural movies, and (2) receptive field location calculated from *in vivo* responses to sparse noise stimuli (Supplemental Figures 2 and 4, respectively). We describe these two sets of experiments below:

***in silico* vs *in vivo* signal correlations**

To estimate generalization errors in measuring signal correlations with digital twin models, for three additional animals, we collected neural responses to the same rich natural video set as the MICrONS data (to train the *in silico* model) and an additional 60 minutes of 36 natural movie clips, each repeated 10 times (to measure *in vivo* signal correlations). Importantly, these 36 natural movies do not overlap with either the video stimulus set used for digital twin training or the 250 movie clips used for measuring *in silico* signal correlations. We divided the 36 clips into two sets:

- A benchmark set of 30 clips repeated 10 times to estimate the "ground truth" signal correlation matrix.
- A set of 6 clips repeated 10 times (equivalent to the amount of repeated natural clip presentation available in the MICrONS dataset) to estimate the *in vivo* signal correlation matrix measurable with the MICrONS *in vivo* imaging data.

Digital twin models for the three mice were trained using the same training data and model architecture as the MICrONS digital twin. We measured the *in silico* signal correlation matrix using 250 natural movie clips in the digital twin models, replicating our approach in the MICrONS dataset. We also computed empirical signal correlations from the 6 clips \times 10 repeats. We then compared both of these signal correlation estimates with "ground truth" from the 30 clips \times 10 repeats benchmark dataset. Our findings indicate that the *in silico* signal correlation matrix more accurately captures the fine-scale correlation structures in the benchmark signal correlation matrix than empirical correlations from the 6-clip set (Supplemental Figure 2b). Furthermore, the *in silico* signal correlations exhibit significantly higher correlation with the benchmark set compared to the 6-clip *in vivo* data (Supplemental Figure 2c,d). We conclude that the digital twin trained on sufficient rich natural movie stimulus provides a higher-fidelity estimate of signal correlations than empirical measurements from limited experimental data.

***in silico* vs *in vivo* receptive field locations**

To validate the measurement of receptive field centers from the digital twin, we took the same approach as above, collecting very long scans in three additional animals with hybrid stimuli including the video stimulus set required to train the digital twin, and an additional 60 minutes of sparse noise stimuli, which is conventionally used for measuring the spatial location and dimensions of the receptive field via spike-triggered averages. Using this data, we then computed the receptive fields for the same neurons from the *in vivo* data (i.e the measured responses) and *in silico* from the digital twin.

The digital twin-derived spike-triggered average (STA) receptive fields are qualitatively less noisy than those derived from *in vivo* responses while still capturing the same receptive field shape and location (Supplemental Figure 4a). Moreover, the readout location, a parameter learned by the model to represent where the neuron attends in the visual field, corresponds well to the STA centers measured with *in vivo* responses (Supplemental Figure 4b).

Importantly, the *in silico* RF measurements are not just "as good" as those computed using classical techniques, but are significantly better for many neurons. Measuring spatial tuning preference with sparse noise stimuli has a known limitation: it only allows for reliable characterization of spatial tuning for neurons responsive to sparse noise stimuli (a fraction of all visually responsive neurons). Even with 60 minutes of stimulus presentation, we could reliably measure receptive field centers for only about 40% of neurons. To illustrate this limitation,

we compared the retinotopic map measured with well-responding neurons (40% of all imaged neurons) to the map measured with non-responsive neurons (25% of all imaged neurons).
100 The poor estimates of RF location from the spike-triggered averages in these latter set of neurons can be seen in the noisy retinotopic map that they produce.

In contrast, our digital twin models were trained from neural responses to naturalistic movies, which drive neurons much better than simple parametric stimuli. The model's readout location is learned across the high-entropy naturalistic movie domain, enabling it to describe the
105 spatial tuning of neurons responsive to various visual features contained within the naturalistic movie domain. Our validation experiments demonstrate that even for the "good" 40% of neurons, the readout location provides a much smoother description of the retinotopic organization of receptive field centers compared to the map estimated directly from *in vivo* STAs (Supplemental Figure 4c). Furthermore, even for the "bad" 25% of neurons not responsive
110 to sparse noise stimuli, the digital twin model can still identify their retinotopic organization, yielding a smooth map. Importantly, the digital twin model has no knowledge of the cortical location of each neuron, and the readout location of each neuron is fit independently with no inductive bias towards smooth progression. Thus, the smooth retinotopy can only result from an accurate characterization of these neurons' spatial tuning preferences.

115 These results collectively demonstrate that the MICrONS stimulus set includes a diverse range of natural videos with sufficient entropy to enable the digital twin models to learn natural video statistics and extrapolate to previously unseen stimulus domains. The digital twin approach generalizes well across the domain of natural movies and outperforms conventional methods in extracting signal correlations and spatial preferences of visually responsive neurons.
120 This comprehensive validation supports the reliability of our digital twin approach and, we hope, addresses the reviewer's concerns about generalization errors.

After training, the model predicted responses with >0.35 correlation for the 37% of selected cells. This means that up to 88% of variance are unexplained even for selected cells, and 63% of cells were not included. It is therefore unclear, how well the digital twin reflects the physiolog-
125 ical responses. At the minimum, the analysis must be repeated on empirically measured signal correlations, including cells on which the digital twin did not perform well, and the data must be presented in parallel. Deviations from the digital twin should be quantified and appropriately discussed.

130 We agree with the reviewer that clearly communicating the performance of the model is essential to interpreting the impact of the study. The aforementioned 88% can be understood as the fraction of total variance that is left unexplained by the model, which we agree communicates something important about overall progress on predicting single trial, individual neuronal responses in the brain. However, it is important to note that the total variance of measured
135 neural activity not only contains visual-stimuli-driven variances but also contains behavioral modulations, measurement noise, and other single trial variation that can not be explained by visual stimuli. Thus, when interpreting the correlation between model predicted responses and single trial responses, it is crucial to normalize it with an upper bound on the performance that could be achieved by a perfect model with access to the visual stimulus. Using the normalized correlation cc_norm proposed by Schoppe et al. 2016, the digital twin model of the
140 MICrONS dataset achieves 67.8% of the performance achievable by a perfect model.

We agree with the reviewer that by setting a relatively high model performance threshold, we risk biasing our conclusions towards neurons that are well-predicted by the model, and thus our results may not reflect the general wiring rule of all neurons in the brain. We performed a series of new analyses to address this concern. In the revised manuscript, we lowered the
145 model performance threshold (CC_{abs}) to 0.2 and the reliability threshold (CC_{max}) to 0.4, thus including over 80% of all functionally co-registered neurons in the analysis. Moreover, we are able to reproduce the main results with signal correlations measured with *in vivo* responses,

including neurons on which the digital twin model did not perform well (Supplemental Figure 3).

150 To further inspect to what extent model performance might affect our analysis, we split the dataset into four quartiles by the model's performance on predicting the postsynaptic neuron's responses. We found that the like-to-like effects are relatively consistent across the four subsets (although most clear in the neurons that are highly visually-responsive and well-predicted by the model), indicating the connectivity rule we discover generalizes across the neuronal population regardless of how well the digital twin model predicts their responses (Supplemental Figure 11).

The authors mention that the "relationships we describe here, while statistically significant, have an apparently small effect size". Given the extremely small effect size (difference in signal correlation of 0.02!), the reader would like to see what the difference in activity between pairs of neurons with these different correlations would look like. Also, the reader wonders if this small effect size is relevant for brain computations.

We agree that understanding the magnitude and consequence of this effect is an important question to the field. We have included a figure appended to this response showing the stimulus responses for a single presynaptic cell (background shading with brighter grey corresponding to more activity), and the activity of its connected postsynaptic, ADP, and same-region controls. Postsynaptic traces are sorted by signal correlation. Neurons with high and low signal correlations can be discriminated by eye, although smaller differences may not be perceptible in these single-trial responses (Rebuttal Figure 1). While the change in signal correlation mean is small with respect to its standard deviation (Supplemental Fig. 10), comparing connected neurons vs same region controls, the mean signal correlation more than doubled for all projection types except for HVA intra-areal projections, where the mean signal correlation increased by about 60%/ (Fig 2d). This figure has a revised design that we hope emphasizes this more clearly. The increase in signal correlation also corresponds to a large change in axon-dendrite co-travel distance and synapse density for neurons with high signal correlation (Fig. 2e,f), and is of a similar magnitude to previous studies (Figure 3c in Ko et al. 2011).

To better understand the consequences of this effect, we performed additional modeling and analysis with a simple 1000-unit recurrent neural network (RNN, Fig 6a). We found that this RNN exhibited similar connectivity rules after training in a simple visual classification task (MNIST) to those we found in the in vivo data (Fig 6b,c), and that selectively pruning "like-to-like" connections from the network resulted in a larger drop in performance compared to pruning a similar number of randomly selected connections (Fig 6d). Similar to the biological data, the change in signal correlation mean between RNN units after training is once again small with respect to its standard deviation (Supplemental Figure 12). While these findings in the RNN do not necessarily generalize in a straightforward way to cortical circuits, they do demonstrate proof of existence of a network model with a similar connectivity principle that plays an important role in network performance.

The abstract presents as a main result that "we did not find that orientation preference predicted connectivity". Given the multitude of caveats interpreting this piece of data (suboptimal stimulus used, low number of selected cells, low threshold orientation tuning threshold for selection, analysis based on digital twin not empirical responses), this emphasis does not seem appropriate. Indeed, misleading, and incorrect. 422ff: When presenting the data on orientation preference, the visual stimulation paradigm (drifting noise, not drifting gratings) needs to be spelled out. Further the authors should use criteria for orientation selectivity (>0.4) that was used before in Ko et al. 2011.

195 The lack of a strong "like-to-like" effect for orientation was surprising to us, and we thought deserved emphasis — given that functional similarity in the orientation domain has been a central focus of existing literature. However this point appears to be somewhat moot, since

after additional proofreading and soma matching (all performed blind with respect to the potential outcome of the analyses), the previously negative orientation tuning result has moved from the borderline of significance to now being positive for V1→V1 like-to-like. Nevertheless, we still do not observe a significant like-to-like rule for orientation in V1 L2/3 to V1 L2/3. We have performed additional analysis comparing our results directly to Ko et al. 2011 and Lee et al. 2016 in Supplemental Figure 9 to examine this effect and we think we have a clearer understanding of the origin of this deviation from previous studies.

Connected neuron pairs within V1 L2/3 projections in the MICrONS dataset do show a similar like-to-like orientation preference consistent with previous literature (Supplemental Fig. 9). However, unconnected pairs show the same effect. We believe this is because of a local orientation bias in the anterior portion of V1 (Fahey et al., 2019) where the MICrONS volume is located, and this may explain why we do not detect a like-to-like effect with orientation in V1 L2/3 for connected cells, above and beyond the same-region control. The abstract and text have been updated to reflect the new analysis.

Finally, although we do not think they are responsible for the weak effects of orientation tuning we observed in the volume, several of the caveats the reviewer mentions above are still important to address, and we touch on these below.

Suboptimal stimulus: We chose an oriented stimulus (oriented pink noise) that was compatible with our available experimental time (avoiding the combinatorial explosion of drifting grating spatial/temporal frequency matrix), which drives strong visual responses, and which has been previously used in the field to compute orientation tuning (Han et al., 2022). Based on the reviewer's suggestion, we now explicitly spell out the visual stimulation paradigm with the orientation tuning results (near manuscript line 489) so that readers can evaluate its suitability.

Low threshold orientation tuning / gOSI: We apologize for our lack of clarity on the point that global orientation selectivity index (gOSI) is not the same as orientation selectivity index (OSI). The gOSI metric in this paper is based on the $1 - CircVar$ metric in (Mazurek et al., 2014), which is a vector-based method designed to reduce the uncertainty in quantifying orientation selectivity of responses, especially in cases where high throughput, unbiased recording methods return many cells with low orientation selectivity, as is the case with calcium imaging. We computed gOSI and OSI with *in vivo* neural responses to oriented noise stimuli. For neurons selected with our gOSI threshold > 0.25 , the computed OSI ranges from 0.43 to 0.99, with mean of 0.56. For both thresholds, the fraction of cells considered orientation tuned (57.4% of coregistered V1 neurons have gOSI > 0.25 , 62.7% of coregistered V1 neurons have OSI > 0.4) is similar to those reported in other studies (72% in V1 layer 2/3 (Ko et al., 2011), 62.9% in V1 layer 2/3 and 58.0% in V1 layer 4 (Kondo and Ohki, 2016)). We have added text to the results section "Orientation tuning is like-to-like within V1 at both axonal and synaptic scales" near manuscript line 494 and methods section "In silico difference in preferred orientation" near manuscript line 1152 to make the distinction between OSI and gOSI more clear, and now include the equivalent OSI values in the methods.

Low number of selected cells: All neurons passing the orientation selectivity threshold (gOSI > 0.25) were included in the analysis when testing the relationship between orientation preference and connectivity, regardless of their CC_{max} or CC_{abs} scores. This includes 57.4% of coregistered V1 neurons and 48.5% of coregistered HVA neurons.

Analysis based on digital twin not empirical responses: As we hope we have shown convincingly above, *in vivo* empirical measurements are not necessarily more reliable or representative of single cell tuning than the predictions of a suitably-trained digital twin.

Since the main reference ((MICrONS Consortium et al., 2021)) is only published in preprint up to date, any relevant methodology should be described in the present paper.

We agree with the reviewer that all relevant methodology should be available in a peer reviewed format if not in the present paper. However, these other manuscripts are also under review for simultaneous publication. We are happy to ask the editors to make reviews and our responses for any paper that has not been accepted available for you to read.

The authors use the concept of “feature weights”, which are defined entirely from the artificial neuronal network. The authors show how this metric corresponds to a biologically relevant feature.

Feature weights capture the nonlinear tuning of the neuron, and are not *a priori* guaranteed to have any relevance to measured biological features, any more than parametric tuning such as orientation. However, we do find a strong relationship between feature weights and biologically relevant features including synaptic connectivity (Fig. 3a,b), synaptic cleft volume (Fig. 3c), and number of synapses (Fig. 3d). Furthermore, in a companion manuscript (Wang et al., 2024), we show that feature weights predict anatomical cell types.

The authors should emphasize from the beginning that all results are derived from a model based on recordings from a single animal.

We now have incorporated results from additional mice in the validation of the digital twin. However, the central dataset is a single animal and we now emphasize this in the text, including the caption of Figure 1 and the first line of the Methods section.

Terminologies and abbreviations need to be defined:

200f: “inappropriate merges” not defined
227f: CC_{max} and CC_{abs} not yet defined
328ff and 996: “breakout analysis” not defined
337: “center distance” not defined

345: “projection breakout” not defined

We have made the following changes:

Inappropriate merges: We define false merges near manuscript line 175.

CC_{max} and CC_{abs} : These terms are defined in the Methods under the heading “Functional unit inclusion criteria”.

breakout analysis / projection breakout: We have removed this text.

center distance: We have clarified this text near manuscript line 395

Overall, sentences are often hard to understand due to missing information or logical flow:

340: “is selected” unclear what this means

335: Paragraph title difficult to understand

544ff: it is not sufficiently explained what is a feature and why it is less “long-range”. In general (already in the main text), it should be explained, to which feature space the “feature weights” refer to

558ff: argumentation unclear. Why could a genetically determined connectivity not also result in higher signal correlations?

581f: “it may result in an apparently independent feature weight similarity contribution” – not clear why

We have made the requested changes:

340: “Is selected” language has been replaced throughout the manuscript.

335: This and other confusing subsection titles have been rewritten for clarity.

544, 558, 581: This section of the discussion has been removed.

Statistical issues

Benjamini-Hochberg correction assumes independence between tests – which seems violated due to individual datasets appearing in multiple comparisons. Please explain why the method is applicable or replace it by an appropriate method.

295 We appreciate the reviewer's attention to statistical rigor and the details of the analysis. Benjamini-Hochberg correction has been proposed to control for the false discovery rate (FDR) across multiple tests when the tests are independent of each other, i.e., the result of one test does not affect the conclusion of another. It has also been shown that Benjamini-Hochberg correction can successfully control FDR under the assumption of positive dependency (Benjamini and Hochberg, 2000), i.e., the existence of one significant result increases the likelihood of the other tests to be significant.

300 In the MICrONS dataset, the hypotheses we are testing are inherently independent of each other. For example, we do not have apparent a priori evidence to believe feedforward or feed-back projections would exhibit significant like-to-like relationships, whether V1 local connections exhibit significant like-to-like relationships or not. Thus, we believe B-H procedure is applicable to control for FDR in our study. Additionally, we include in the supplemental tables the statistics and uncorrected p-values for all of our analyses to provide a complete perspective of the comparisons.

310 Causality is reversed in arguments 277ff: the authors build their arguments analysis on the assumption that "signal correlation affects connection probability" – but in fact, causality is reverse. The logic should be reversed.

441ff: "Signal correlation and feature weight similarity independently contribute to connectivity" – the causality again is reverse. This should be clarified. The model should be phrased in terms of posthoc likelihood, not as "contribution", which can be understood as mechanistic contribution.

320 We agree that it is important to be cautious with interpretations about the direction of causality, and since the dataset is collected in an adult mouse at a single time point, there is no possibility of testing the direction of effects. However, we disagree that the direction of causality is self-evident, since there are reasonable motivations to expect that causality might occur in either direction. On the one hand, as the reviewer suggests, a synaptic connection between a pair of neurons could lead to connected cells firing more similarly. However, each postsynaptic neuron receives up to thousands of inputs and so a single input from the presynaptic neuron might have a limited contribution to the functional tuning of the postsynaptic neuron. On the other hand, the Hebbian learning rule of "fire together, wire together" has also been well established, and neurons with the opportunity to form synapses (i.e. with axon-dendrite proximities) might be more likely to do so if they are driven by similar stimuli and thus activated together. Statements about explicit causality have been removed throughout the manuscript, and we thank the reviewer for pointing out the ways our assumptions have appeared in the text.

330 **Minor comments**

197: Does the sentence imply that only axonal, not dendritic processed were proofread? Please clarify in more detail. Please indicate error rates of automated detection versus proofreading (also for line 640)

335 Dendrites were only extended in manually proofread neurons, and only rarely as they were largely complete directly from the initial reconstruction. We have now added more detail about the manual and automated proofreading process used for neurons in this study in the Methods under the heading "Morphological Proofreading". For the error rates of the initial reconstruction compared to manual proofreading see Elabbady et al. 2024, and for the validation of automated proofreading please see Celii et al. 2024.

340 Fig 3g: the difference red vs black might be non-significant, but is quantitatively similar to blue vs black

The underlying analysis and findings for this plot have been updated as described in the orientation tuning section above. This exact plot is no longer present in the manuscript, but the closest analog can be found in Supplemental Figure 8a,b.

345 354: RF center distance does not correlate with synapse volume: could this relationship be lost in the digital twin because of a disturbance of the RF map (e.g. filtering effect)

The reviewer asks a great question about the receptive field locations generated from the digital twin. Since the submission we have performed additional validation on the digital twin receptive fields. As described above, the digital twin offers comparable or superior receptive field location mapping compared to *in vivo* data, although this makes it difficult to compare against ground truth data to evaluate for a filtering effect introducing noise into the analysis. The most comparable plot for this finding in the current manuscript is Figure 3d.

856ff: "Trials with more than 10 consecutive untracked pupil frames were discarded (18-180 trials per scan, 2-39%)." – why?

355 The most common cause of pupil frame tracking failure was that the mouse closed its eye, in which case the stimulus as presented would not be reliably available to the visual system and could impact the functional tuning measurement. This is now clarified in the Methods section under the heading "Preprocessing of neural responses and behavioral data."

360 **Referee 2 (Remarks to the Author):**

This manuscript begins to deliver on the promise of high-density connectional data combined with functional characterization that can be afforded by 3-D EM reconstructions. While this is a very important manuscript and an impressive effort, one can't help but feel a bit disappointed by what is delivered. The most positive aspect is that principles of connectivity previously described only for local connectivity within and across layers in mouse primary visual cortex are now extended to intra-areal feedforward and feedback connections. These connections, like local connections are more frequent for excitatory neurons with similar functional properties. But if what we get is mostly confirmation of what we already know, then what's all the fuss about? Why do we need to spend more and more money on larger and larger EM volumes?

370 Nearly ten years ago, in our submission for MICrONS, we proposed a volume of visual cortex situated at the border between V1 and higher visual areas with the explicit goal of studying functional properties of connected feedforward, and feedback projections — the latter being of particular relevance to theories such as predictive coding or probabilistic inference. In choosing the cells for labor-intensive manual proofreading and axon extension, we specifically invested in cohorts of cells with feed-forward and feedback projections. And finally, after extensive analysis against a sophisticated set of controls, we found that — at a first pass — the relationship between function and connectivity for long-range connections is indeed like-to-like, similar to local connections within V1. Whether this is expected or surprising, we would argue that it is unambiguously a novel finding that provides important constraints on our understanding of the role of feedback projections in sensory processing.

380 A central advance of this study is that we are able to answer whether the similarity in neurons connected by long-range projections is due simply to axon trajectories that target areas with similarly-tuned neurons, or because of "Peter's Rule"-violating specificity at the level of individual synapses, above and beyond axon and dendrite geometry. Because of the investment in dense reconstruction and automated cleaning of all postsynaptic neurons in the volume, we are able to unambiguously answer this important question. The answer is both: Long range axons are indeed significantly more likely to intercept or co-travel with dendrites of similarly

tuned cells. *And*, beyond this axon-level specificity, neurons with similar functional properties are also more likely to convert these chance meetings to a synaptic connection. These axonal-scale and synaptic-scale rules relating functional properties and connections are likely to have very different underlying mechanisms and very different implications for learning and computation in neural circuits. They also — as we show for the first time — differ in the importance of similarity in the "what" and "where" of visual responses. Axonal-scale like-to-like depends on receptive field location and feature similarity, while the synaptic-scale like-to-like rule is only true for the visual features that neurons encode. It is not clear that there are any alternative approaches that could reveal this preference for visual features in the synaptic-scale like-to-like rule for long-range projections — over and above axonal-scale selectivity.

There are several things that are not included but might have been included to make the manuscript stronger. First, there is no analysis of second-order (or nth order) contingencies of connectivity. For example, if two neurons are connected to each other, are they more or less likely to have other inputs or outputs in common?

We now include an analysis of second-order functional connectivity principles, focusing on neurons receiving a common input from a single presynaptic neuron (Figure 5). We find that cohorts of neurons downstream of a single presynaptic cell are more functionally similar than would be predicted by a pairwise like-to-like rule. This analysis provides a first glimpse of the relationships between functional properties and higher-order motifs that this volume may reveal, especially as ongoing proofreading restores a greater fraction of the volume to a high-confidence graph.

And it is disappointing that inhibitory neurons are entirely excluded from the analysis, even if they were not functionally characterized. (Why weren't they?) Do inhibitory neurons follow the same rules of connectivity as excitatory neurons? Are these the same for inhibitory neuron types that target cell bodies (basket cells), axons (axo-axonic), dendritic spines and shafts, or other inhibitory neurons? It is understandable that the authors might want to reserve these descriptions for other publications, but then they should not expect that the limited descriptions here, with their relatively limited impact, will be a good fit for a high-profile journal. The authors do acknowledge all of this in their discussion and state: "Ongoing investments in proofreading, matching, and extension efforts within this volume will have exponential returns for future analyses as they yield a more complete functional connectomic graph. There is much more to discover about this relationship from this dataset, and others like it that are currently in preparation." So perhaps the bottom line is that it is just not quite there yet.

Inhibitory neurons were not functionally characterized in this mouse because of the challenge of expressing GCaMP6 in both excitatory and inhibitory cells at the time when the data was collected. In fact, several mice with labeled interneurons were functionally recorded but were not selected based on tissue quality for EM. We certainly agree with the reviewer on the importance of understanding the role of inhibitory neuron connectivity within the circuit. A central companion manuscript (Schneider-Mizell et al., 2024) from the consortium package explores in depth the connectomic cell type and subcellular targeting of inhibitory neurons in the dataset, although without explicit relationship with function.

Overall the manuscript is very well written, the logic clear and the citations to the relevant literature are mostly appropriate. There are a few previous publications that probably should have received a bit more prominent mention in the introduction and discussion. Previous studies (Wertz et al.; Rossi et al.) identified and functionally characterized very large numbers of local presynaptic neurons providing direct input to a small number of postsynaptic neurons – hundreds of inputs per postsynaptic cell. These numbers are far greater per postsynaptic neuron than described here, where the numbers of presynaptic cells are only one-tenth the numbers of postsynaptic cells. And these included cells sampled across cortical layers (albeit always with the postsynaptic cell in layer 2/3). The Introduction disposes of this by simply stating "and typically limited to cortical Layers 2 and 3 (L2/3)." And the Discussion makes no mention of these prior studies or how

the results relate to the present study. Discussion of this would seem to merit more attention
440 than the long description of methods for estimating receptive fields, which are apparently already
described in detail in the Wang et al. Biorxiv paper. For example, Only the Ko et al. data are
compared to the EM data with respect to orientation selectivity. But Wertz et al. has data for
hundreds of input neurons with far greater statistical power.

445 We agree with the reviewer that different methods contribute different strengths to address
this question, and apologize for the oversight. We now include Wertz et al and Rossi et al in
both the Introduction and Discussion.

In the discussion: "This may be due to practical differences, such as the parametric stimulus used
to characterize orientation tuning (in silico drifting noise with orientation coherence in our study
versus drifting gratings), cell selection criteria (gOSI > 0.25 in our study versus OSI > 0.4 in Ko
450 et al. 2011)," Why not simply change the criterion to address this? Or is the sample size too
small for well-tuned cells? It appears that the OSI distributions in the Wang et al. Biorxiv might
be generally low. Perhaps this is not a good method for assessing OSI?

455 We apologize for the confusion arising between global orientation selectivity index (gOSI)
and orientation selectivity index (OSI). Please see our full response to this similar concern
raised by Reviewer 1 above. To briefly recap, the gOSI metric in this paper is based on the
 $1 - CircVar$ metric in (Mazurek et al., 2014), which is a vector-based method designed to
reduce the uncertainty in quantifying orientation selectivity of responses, especially in cases
where high throughput, unbiased recording methods return many cells with low orientation
selectivity, as is the case with calcium imaging. For neurons selected with our gOSI threshold
460 > 0.25, the computed OSI ranges from 0.43 to 0.99, with mean of 0.56. For both thresholds,
the fraction of cells considered orientation tuned (57.4% of coregistered V1 neurons has gOSI
> 0.25, 62.7% of coregistered V1 neurons has OSI > 0.4) is similar to those reported in other
studies (72% in V1 layer 2/3 (Ko et al., 2011), 62.9% in V1 layer 2/3 and 58.0% in V1 layer
4 (Kondo and Ohki, 2016)).

465 Fig 4 of the Wang et al BioRxiv paper plots the accuracy with which the digital twin model
is able to predict the orientation tuning of a neuron's response to gratings based on responses
to natural scenes. This might be added as a supplemental figure so that the reader can under-
stand the reliability of the orientation predictions. A reader should not have to refer to another
manuscript to obtain this crucial information.

470 We have included this validation figure as Supplemental Figure 7. We now also include two
additional supplemental figures based on data from additional validation experiments for *in*
silico versus *in vivo* measurements of signal correlation and receptive field location (Supple-
mental Figures 2 and 4, respectively). Please see our detailed description of these experiments
in our response to Reviewer 1.

475 All of the data describing significant positive correlations between connectivity and function are
highly processed, leaving one with very little insight into actual observed rates of connectivity.
Please provide basic information about the raw data used to calculate fold changes in connec-
tion probabilities (e.g. Fig. 2f-g). How many pairs of neurons for each type of comparison were
present as both ADP control values and as actual connectivity values? For example state: "For
480 local connectivity within V1, xx presynaptic and xx postsynaptic cells were assessed. Of these
cells there were xx actual connections and xx ADP control contacts. XX of the xx actual con-
nections (xx%) were for neurons with similar function(Threshold >0.3 correlation) versus xx that
had different functions. This contrasts with xx% of neurons with similar function for ADP con-
tacts. For local connectivity within HVA, xxx. For feedforward connections from V1 to HVA
485 For feedback connections from HVA to V1" Perhaps these values are somehow embedded in
Supplemental Table 4. If so, then please clarify and add relevant detail to the table legend.

We agree with the reviewer that transparency on the data used for quantifying the correlations between connectivity and function is critical for understanding the power and limitations of the current analysis. We have redesigned our statistical analysis and now report two direct anatomical measurements, i.e., axon/dendrite co-travel distance (L_d) and synapse density (N_{syn}/mmL_d), instead of fold changes in connection probability. We now include the deviations from the mean (ΔL_d and $\Delta N_{syn}/mmL_d$) in the main figures and the corresponding mean values for each projection type in the figure legends. In addition, we have adjusted the supplemental tables to report the number of unique presynaptic neurons, the number of unique postsynaptic neurons, and the number of unique presynaptic/postsynaptic pairs included in the connected and control groups at a level of stratification matching the analysis design as suggested by the reviewer. We have reviewed the tables, table legends, and text to confirm the link to each analysis is made clear and readily available.

500 **Referee 3 (Remarks to the Author):**

Being able to relate the activity patterns of neurons within a network with their exact synaptic connectivity promises new insights into the basic operational principles of the brain. Currently, vast research efforts in neuroscience are invested in such projects in different model organisms. Here, Ding, Fahey, and Papadopoulos et al. utilize their previously published, openly available
505 MICrONS dataset and present a first analysis of a subnetwork spanning all layers and several visual cortical areas in the mouse. Their final connectivity matrix consists of 122 excitatory presynaptic neurons that connect with 2000 excitatory postsynaptic neurons with known functional properties.

The authors analyze the connection probability between neurons, the number of synapses formed,
510 and the synaptic cleft volume for all connections they detect and ask how these features relate to different components of the visual responses. Additionally, they introduce two anatomical control measurements that allow for more in-depth analysis of connection specificity.

The authors report that neurons that respond to similar visual features are more likely to be connected, which is supported by previous work. Surprisingly, neurons with similar orientation tuning
515 and neurons whose receptive fields are spatially close do not show a tendency of having a higher connection probability. The authors conclude that previously reported like-to-like connectivity patterns generalize across cortical layers and visual areas.

The combination of methods at this scale is an impressive technical tour-de force effort. The openly available MICrONS dataset promises to become an essential community resource for
520 systems neuroscience. With the current study, the authors are now pioneering analyses tools to test circuits neuroscience questions in their dataset.

We thank the reviewer for recognizing the enormous value of this data set and the importance of developing analysis tools to test specific questions in this volume.

Therefore, I expect that this study will receive much attention across the neuroscience community and beyond, and it has the potential to become a landmark paper.
525

The paper is well written, and the methods and statistics seem to be robust.

In summary, I am quite enthusiastic about the work, however, I have several questions regarding the analyses, the use of the digital twin model, and potential experimental biases.

Also, given the general readership of Nature, I think the paper could be clearer in some sections.

530 We address these questions and requests for clarifying text below.

Specifics

It is a strong point of this work that the authors measure the connection probability between neurons, the number of synapses formed, and the synaptic cleft volume of all synapses.

The latter is directly related to the postsynaptic density area and therefore the functional strength
535 of the synapse. So, in principle, the authors have a complete picture of the anatomical connectivity between all neurons, which is remarkable and quite unique!

However, they only analyze these anatomical features separately, which, in my opinion, is a missed opportunity. To me, the most relevant anatomical readout (the feature that should predict best how strongly one neuron drives another neuron) is the total anatomical synaptic weight
540 between these cells.

In the case of this study, this would be the sum of the synaptic cleft volumes of all synapses in a connection. The reason why I think this is important is because by analyzing each anatomical feature separately, the authors could miss if other features 'compensate'.

For example, some connections might form only very few, but strong synapses (i.e., synapses
545 with large synaptic clefts). Wouldn't this show up as a weak connection in the analyses?

Conversely, a connection with many weak (small) synapses would show up as a strong connection with high connection probability.

Intriguingly, the authors report that similar orientation tuning and spatial receptive field features were not associated with higher connection probability.

550 By contrast, Cossell et al. (2015) reported that connection strength measured with slice ephys predicted similar orientation tuning and spatial receptive field overlap.

I am wondering if the authors could also uncover similar effects to what Cossell et al. reported if they computed a combined anatomical synaptic weight index.

Therefore, I am not convinced that mostly looking at connection probability by itself is the best
555 way to analyze these remarkable data.

We agree with the reviewer that there is an opportunity in this volume to look at ultrastructural features beyond binarized connectivity matrices. We attempted to compute a combined anatomical synaptic weight index (the total synapse cleft volume) and repeated the analysis for orientation and receptive field center distance. We do not observe a qualitative difference after weighting connectivity by synaptic volume (Rebuttal Figure 2).
560

Partially addressing the reviewer's concern about a binary connectivity analysis, the updated analysis now includes an updated continuous metric for connection probability that includes axon/dendrite co-travel distance, as in (Lee et al., 2016).

Regarding comparisons with previous literatures, after additional proofreading and soma
565 matching, the previously negative orientation tuning result has moved from the borderline of significance to now being positive for V1→V1 like-to-like. Please see above section on orientation tuning validation in response to Reviewer 1, for details. Cossell et al. 2015 reported that connection strength measured with slice ephys predicted spatial receptive field overlap. Spatial receptive field overlap measures both differences in the center distances and shapes of
570 RF, which would have been partially included in the feature similarity in the quantification of our study. A companion paper in the package reproduced results similar to those of Cossell et al. 2015 in V1 L2/3 and showed that connected neurons have more similar most exciting inputs (MEIs) Ding et al. (2023).

The analyses rely heavily on the digital twin model that the authors create.

575 If I understand correctly, the digital twin is a way to 'bootstrap' new functional responses for neurons that couldn't all be acquired experimentally, which allows for a more in-depth correlation analysis.

With the general readership of Nature in mind, I think more could be done in the paper to explain and justify the model. Just as an example, only in line 260, the authors finally provide a
580 real rationale for the digital twin. This seems to me a relatively unconventional method, and it should probably be explained better in the introduction.

We now explain and justify the model in greater detail in the introduction, as requested.

Moreover, when talking about the digital twin, the authors frequently reference Wang et al. (2023), which appears to be the unpublished methods-companion paper.

585 Because this model is unpublished, I find it hard to judge the validity of this approach and the authors could do more to convince the readers that this modeling approach produces 'correct' neuronal responses that could also be measured experimentally.

590 We have added substantial additional validation of the digital twin, as described above in the response to Reviewer 1 regarding model performance validation within domains of signal correlation, receptive field location, and orientation tuning. We are also happy to ask the editors to share the reviews of Wang et al, currently under revision.

The authors claim that this is indeed the case (line 123f), but while the model responses in SFig. 4 c seem to match overall; they are still a bit different from the original *in vivo* cell responses. As an additional control, for example, would some of the results also hold if only 'real', i.e., experimentally recorded responses were included in some of the analyses?

600 We have reproduced the main signal correlation results from Figure 2 with *in vivo* responses as the reviewer suggests (Supplemental Figure 3). As we describe in the response to Reviewer 1 above, noise in *in vivo* measures may make them a worse approximate of longer *in vivo* recordings than the *in silico* estimates for models trained on extensive data. For example, we show that *in silico* signal correlations in an appropriately-trained digital twin outperforms 10 min *in vivo* signal correlations measured from *in vivo* data. (The comparison is with respect to a ground-truth dataset of *in vivo* responses to a larger number of repeated stimuli; Supplemental Figure 2, please also see response to Reviewer 1 near rebuttal line 53 for more detail).

605 Have the authors assessed during the proofreading stage how completely their neuronal morphologies could be reconstructed?

In my experience with such EM datasets, sometimes an axonal or dendritic branch can become untraceable in the tissue (due to fixation issues, cracks, folds, etc.).

610 For the neurons reported in this study, (how often) was this the case and do the authors think it might have affected the results?

The reviewer is correct that this EM dataset contains cracks and folds, and that these artifacts lead to incomplete morphological reconstruction. For more detail on how our collaborators address these artifacts in the context of the automated dense reconstruction, please see the companion paper (Macrina et al., 2021).

615 For the presynaptic neurons in this study, we invested in manual proofreaders to overcome the limitations of the dense reconstruction. We devised a proofreading protocol that balanced completeness (per neuron) with coverage (across neurons). Proofreaders were instructed to extend every branch of the dendritic and axonal arbor to its natural termination, stopping only at volume boundaries or if the false termination was impassable after a 10 minute attempt (MICrONS Consortium et al., 2021). Proofreaders were instructed to leave annotations for every termination indicating its status, whether natural or incomplete. From these annotations we can attempt to estimate the completeness of the reconstruction. For dendrites, the median neuron had a percentage of untraceable ends of 1% (n=148 fully proofread dendritic arbors). For axons, the median neuron had a percentage of untraceable ends (not including those at dataset boundaries) of 43% (n=84 fully proofread axonal arbors).

625 In addition to incomplete morphology because of untraceable ends, there are a subset of neurons (n=64) that are intentionally left incomplete. For these "partially proofread" neurons, we focused on extending axon collaterals that were pre-screened to project inter-areally, specifically to enrich the dataset with feedback projections.

630 In working on the revision of this paper, we realized that the relative completeness of presynaptic neurons was indeed affecting the results. In the local HVA projection group, we noticed

that when pairs were grouped by presynaptic neuron, there was a negative correlation for both mean L_d vs signal correlation and mean N_{syn}/mmL_d vs signal correlation, and this correlation could be explained by whether the presynaptic neuron was fully or partially proofread (Rebuttal Figure 3). Because the partially proofread group is exclusively located in HVA, and because only their inter-area collaterals were extended, their local collaterals are very short. As a result, their local synapses (HVA \rightarrow HVA) are very proximal to the presyn. Because nearby neurons also tend to have higher signal correlations, this gives the partially proofread local connections on average higher signal correlations than the fully extended HVA local connections. In the current version of the manuscript, all of our analyses now take into account these varying baseline levels of L_d , N_{syn}/mmL_d and signal correlation for each presyn. We now look at deviations from the presynaptic means, essentially centering the analysis per presyn before aggregating across all of the presyns in a grand average. To better communicate this caveat to our readers, we have now added a sentence to the discussion near line [Ongoing investments in proofreading].

For example, are long-range projections more likely to be incomplete?

In general, fully proofread long range projections are more likely to encounter an impassable artifact and thus are more likely to be incomplete when compared to fully proofread local projections. However for the subset of partially proofread neurons, it is their local projections that are more incomplete.

Moreover, if I understand correctly, the authors suggest that local axon branches might be proofread less reliably because emphasis was given to long-range projections (line 274f).

Maybe the authors could provide some (ideally quantitative) estimate that allows readers to judge how often a local (and long-range) branch might be incomplete?

Importantly, all of the manually proofread neurons are "cleaned" to the same degree, that is, the false merges are removed from both dendrites and axons. This ensures that there are no incorrect synapses (false positives) associated with the presynaptic neuron. However, for "extension" there is some variance between presynaptic neurons. For partially proofread neurons, local axon branches are proofread to a lesser extent than the inter-areal projections, and thus partially proofread neurons have more missing synapses (false negatives) when compared to fully proofread neurons. We have updated the manuscript to make this distinction more clear, including expanding the proofreading methods in the Methods section under the heading "Morphological Proofreading", adding a table (Supplemental Table 1) describing whether presynaptic neurons are included under the "fully proofread" or "projection focused" protocols, and adding Supplemental Fig 1 to give readers a visual intuition of the relative completeness across fully and partially proofread neurons.

I didn't find any mentioning that synapses were manually proofread, too. Have the authors assessed how reliable their synapse detection algorithm is?

Specifically, I think there are two important questions:

- what is the overall false-negative and false-positive rate? Some estimates should be reported. The authors provide some estimates for their L2/3 dataset in Turner et al. (2020), but are these numbers also correct for the more heterogenous Cortical mm^3 ?

We thank the reviewer for identifying an important point about the reliability of synapse detection in the cortical mm^3 volume. Since the submission, our collaborators have performed extensive validation of the synapse detection algorithm in this volume, and the result is now Supplemental Figure 1 in (MICrONS Consortium et al., 2021). In short, synapses were manually identified in 70 small subvolumes ($n=8,611$ synapses) distributed across the mm^3 volume and compared to the automated synapse detection. The precision of automated detection is 96% and recall is 89% with a partner assignment accuracy of 98%. From these numbers, we can estimate the false negative rate ($1 - \text{recall}$) to be 0.11. With a precision of 96%, we can be confident that the vast majority of identified synapses are correct.

- o More importantly, are there any systematic biases for missing certain types of synapses? Just as an example, are asymmetric synapses on dendritic shafts detected with equal likelihood as asymmetric synapses formed on dendritic spines? Systematic biases might affect certain connections more than others and could bias the results.

685

The reviewer is correct to raise concerns about the possibility of a systematic bias in synapse detection. Smaller synapses are easier to miss and likely to be less reliably detected than larger ones, but the scope of the validation did not differentiate by synapse type (e.g., excitatory vs. inhibitory synapses or whether the synapse involved spines, shafts, somas, etc.).

690

However, with respect to this manuscript, all pre- and post-synaptic partners are restricted to excitatory neurons, limiting the potential bias to these excitatory synapse subtypes.

Along those lines, I noticed in Fig. 4 that the canonical circuit motif of recurrent connections in L4 in V1 appears surprisingly weak in all panels.

Could the authors clarify if this is in fact unexpected or if I'm missing something?

695

In the plots in Figure 4, the diameter of the square represents the number of synapses tested and the face/boundary color represent the strength of "like-to-like" selectivity in where those synapses are formed. Thus, the lesser square diameter in V1 L4 → V1 L4 represents the relative scarcity of this connection type in our proofread dataset, and not a low connectivity rate between cells of this type. We apologize for the confusion. To provide a more informative visualization, we have now used the diameters of the squares to represent the mean axonal dendritic travel distance and mean synapse conversion rate (Figure 4).

700

Previous work could be cited more. While the authors mention previous smaller-scale structure-function studies in mouse visual cortex, they don't mention some of the efforts of relating calcium imaging and EM in the same animal in other model organisms, which are in fact cited in their 2021 MICrONS paper.

705

We apologize for the oversight, and have added these citations.

277f "To measure how signal correlation affects connection probability compared to either same region or ADP control, we quantified the fold changes in connection probability as a function of signal correlation."

710

I find this confusing because it suggests to me that connection probability was compared between the connected pre- and postsynaptic neuron and the controls. But don't the controls have a connection probability of zero per definition? So presumably, what was compared were the signal correlations between connected neurons and unconnected neurons?

715

We agree with the reviewer that the phrasing in L277f was confusing. As we have mentioned elsewhere in this response, we have substantially revised this analysis to (we hope!) improve the clarity of presentation. Rather than counting discrete proximity "events", we now, as in Lee et al. 2016, compute an analog measurement of axon-dendrite co-travel distance. Previously, we did not test the axonal-scale effect directly. We inferred this relationship by comparing "connected vs ADP" to that of "connected vs same region". Now, we directly test the axonal scale by comparing "ADP to same region".

720

We have attempted to describe the analysis more clearly in the revision, as well as providing units (ΔL_d and $\Delta N_{syn}/L_d$) directly on the figure to aid in clarity.

Also, wouldn't it be more logical if the sentence would say: "To measure how connection probability affects signal correlation...?"

725

Please see our reply to Reviewer 1 about the direction of causality between signal correlation and synaptic connectivity near rebuttal line 316. In short, we believe there are good arguments for causality in both directions, and we have tried to soften any strong statements about the direction of causality throughout the manuscript.

I also find the associated figures (Fig. 2 f, g and Fig. 3 c, f, i) difficult and unintuitive to interpret. I'm not really sure what exactly is compared here. My interpretation is this: to get the red plot for example (Fig. 2 f), the authors pooled all connected neurons and all ADP controls together in one population. Then they asked: for the different bins of signal correlations in this pooled population, what was the probability that cells were connected. Is this correct? In any case, this analysis should be explained in more detail.

The reviewer's interpretation is correct. however, in the revision, the analysis has changed substantially. For a summary of the changes please see our response above regarding L227f. We have attempted to clarify the analysis approach and add more detail in the manuscript.

In their introduction, the authors motivate their structure-function study in visual cortex by citing Hubel and Wiesel: how might center-surround receptive fields from the LGN give rise to edge detectors in V1? However, this question cannot be tested in their dataset because the thalamic afferents are not included. I guess this reference makes sense if the authors just want to highlight the general importance of the approach, but it might also be considered a poorly chosen example because it can't be addressed here.

This is an excellent point and we have changed the reference to Hubel & Wiesel's complex cell model. The goal is to highlight the general importance of the approach, but also to identify a relevant question that could potentially be tested with this dataset (please see companion paper for an initial test of this possibility Ding et al. (2023)). It is also worth noting that the simple/complex cell model also has a built-in like-to-like assumption, in that simple cell inputs should have the same orientation as the target complex cell.

In any case, this shows that the field is still at the beginning of circuit-level structure-function analyses and the authors could further increase the impact of their work by highlighting the importance of, and explicitly inviting future studies in rodents and also in other neuroscience model organisms such as the fruit fly and zebrafish.

We have added several sentences in the discussion highlighting the importance of future work in this area.

Minor comments

First, the authors introduce the digital twin model and explain that they can separate functional tuning into a 'feature component' and a 'spatial component', but then their initial analyses all use 'in silico signal correlation' as a metric, which is not explicitly explained until the discussion. Could the authors explain the differences between these metrics in more detail and explain their rationale for using one or the other?

Signal correlation is the correlation between neural responses averaged over repeated presentations of a visual stimulus. *In silico* signal correlations use responses generated from the digital twin, although without averaging because digital twin responses are not variable. The digital twin responses can already be thought of as having learned the trial-averaged *in vivo* responses.

Signal correlations contain information about both feature tuning similarity and spatial tuning similarity. This is because neural responses to a given stimulus depend on whether that stimulus contains the appropriate dynamics and statistics (i.e. features) to drive the neurons to fire and on whether the stimulus is presented to the neurons' receptive field location.

The digital twin enables us to factorize feature and space into the "readout feature weights" and the "readout location" for every neuron and analyze these separately.

241 – 253 I feel this is an important paragraph, but it is difficult to understand the logic here. Maybe the authors could try to rephrase or simplify?

We have completely re-written this section with an aim towards more clarity.

469 The abbreviations 'SC' and 'FW' are never formally introduced.

We now avoid these abbreviations.

609-615 This sentence is hard to understand, maybe 'even' should be 'evenly'?

The sentence has been rephrased near manuscript line 516.

780 858 Could the authors provide the reader with a short rationale for why the model architecture was changed with respect to Wang et al.?

In this case, the difference in architectures was primarily due to the difficulty of aligning submission dates between two continuously evolving projects. The model version and training has now been more closely aligned between both papers in the resubmission. For orientation tuning analyses, we used a model with an older architecture because it was trained more extensively on oriented stimuli (our "Monet" stimulus). In this model, validation experiments showed that *in silico* orientation tuning was slightly better aligned with *in vivo* data.)

875 Could the authors report whether certain types of neuronal tunings were better modeled by the digital twin than others? In that case, the selection criteria reported here would lead to a dataset in which some response types would be underrepresented, which would be important to know.

This is an excellent question. As described above (see "To further inspect to what extent model performance" in response to Reviewer 1 near rebuttal line 150), we have validated our results when using different selection criteria, and the main results are stable (although they are strongest for neurons that are well-tuned and well-predicted by the model), please see Supplementary Fig. 11.

Referee 4 (Remarks to the Author):

Ding, Tolia and their many colleagues investigated the relationship between synaptic connectivity and visual function using the large-scale EM MICrONS dataset combined with computational modeling.

The "digital twin model" uses a state-of-the-art deep recurrent neural network to innovatively capture single-neuron responses to calculate feature weights and receptive fields, enabling a refined view of functional similarity. Their approach is poised to reveal fundamental rules governing cortical wiring and information processing. With this combination of data and tools the authors set out to either confirm a relationship between functional similarity and connectivity, establishing principles for cortical organization or undermine the link to orientation tuning, challenging existing theories of visual cortex wiring.

Briefly, the authors found neurons with correlated signals tend to connect and that connectivity depends more on feature weight than receptive field similarity; surprisingly, orientation tuning did not predict connectivity. They claim these relationships generalize across layers and areas, though some lack significance. A model found feature and signal correlations independently contribute to connectivity. Overall, this work highlights how *in silico* modeling and connectomics at scale provide mechanistic insight into population dynamics emerging from heterogeneous single-cell interactions.

While the extraordinary scale and resolution of the data in this study could yield groundbreaking insights into visual cortical organization, the current manuscript does not fully realize this potential. The most compelling finding is that connectivity depends on feature, not orientation similarity—challenging notions that it relies on any one parameter and suggesting complex rules govern network architecture. Curiously, most relationships reported have very small effect sizes; I found this particularly surprising. One interpretation is simply that the authors, like the rest of the field, have been trying to explain connectivity with the wrong functional correlates.

Why should all synapses reflect the simple visual features and not more complex behaviorally-related computations in visual cortex? Interpreting modest effects as averages of diverse, nonlinear interactions across cells could inspire new theoretical frameworks.

We thank the reviewer for recognizing the novelty of our findings regarding connectivity and feature similarity. We agree that the effect sizes are relatively small, and in the revision we explore two potential explanations, in an attempt to put these small effects in context. The first possibility is that the effect sizes are small because we are focused on pairwise comparisons, and neurons are not pairwise actors. In this case, we would expect functional similarity (or other relationships) to be even stronger in higher order motifs. Our analysis in Figure 5 supports this view, as we find that the functional similarity across a cohort of postsynaptic neurons is higher than predicted from pairwise relationships. We have added the point raised by the reviewer in the discussion near manuscript line 697. A second, related effect, is that small pairwise effects may have large consequences in densely-connected networks. This view is supported by our simulation analysis looking at weak like-to-like connections (of the magnitude we observe *in vivo* in an artificial recurrent neural network). In the RNN, we find that ablating like-to-like connections hurts performance more than random connections, suggesting that they are privileged in shaping network performance.

Finally, perhaps due to the complexity, the major issues limiting this paper's impact are the lack of coherence and clarity in presentation; failure to reconcile uncertainties and using generative models of visual processing to tie findings together. Nevertheless, this work provides a glimpse into the potential of systematically combining functional modulation and EM reconstruction at synaptic resolution.

Addressing these issues could significantly strengthen the study overall and establish whether its scientific contributions are important for a broader audience.

We hope the reviewer finds that coherence and clarity in presentation and these other deficits have been improved in the revised manuscript.

Specifics

The analyses are based primarily on digital twin model to test the relationship between neuronal tuning similarity and synaptic connectivity. The use of digital twin model to factorize neuronal tuning into feature and spatial components indeed provides us a new view of the neuron's functional properties.

We thank the reviewer for recognizing the power of this method, when combined with appropriately rigorous validation experiments.

These model-inferred functional metrics are exciting but also require further validation and assessment of model variations and uncertainties.

Feature weight is potentially a more comprehensive characterization of a neuron's tuning properties than metrics that depend on specific parametric stimuli, such as orientation tuning, and it can be generalized to natural and complex stimuli; on the other hand, compared to signal correlation, similarity in feature weights may be a better characterization of tuning similarity between neurons.

However, one may need to be careful when drawing conclusions about the relationship between neuronal tuning similarity and synaptic connectivity with those *in silico* generated metrics. Although *in vivo* and *in silico* functional metrics such as receptive field location and orientation selectivity are well correlated, the correlation between certain *in silico* functional metrics and synaptic connectivity cannot be directly extended to their *in vivo* counterparts, and the same applies to lack of correlation. For the four *in silico* functional metrics used in this study, feature weight must be derived from the digital twin model, but the conclusions about the "like-to-like"

870 rules regarding signal correlation, receptive field location, and orientation selectivity should be validated with *in vivo* metrics too.

We now include a supplemental figure replicating the main results of Figure 2 with *in vivo* (rather than *in silico*) signal correlations (Supplemental Fig. 3). Furthermore, in a separate series of experiments, we showed that model readout location outperforms classic methods at characterizing the spatial preference of neurons (as discussed in response to Reviewer 1 above) (Supplemental Fig. 4). We now observe a significant like-to-like rule for *in silico* orientation in V1, after adding additional proofreading and 2p-EM coregistration (blinded to subsequent analyses).

This study does not find a relationship between synaptic connectivity and orientation selectivity, which contradicts with Ko et al., 2011 and Lee et al., 2016. In the discussion section, the authors explain practical differences that might have led to different conclusions of this study of previous ones: parametric stimulus to characterize orientation tuning, cell selection criteria, and the location and size of the area being studied.

Those claims make sense, and especially the location of the area being studied is a very likely reason, since different parts of the visual area may follow different connectivity rules.

Please see our earlier responses to Reviewer 1 near rebuttal line 195 for more detail about the current status and interpretation of the orientation tuning results.

Nevertheless, as discussed in the previous point, the conclusion was drawn with *in silico* orientation selectivity, and it would be more decisive to use *in vivo* orientation selectivity to validate this conclusion.

We have attempted this analysis with *in vivo* orientation tuning, and the results are qualitatively similar but are not significant (Rebuttal Figure 4). This is likely due to noise in *in vivo* recording that makes estimates from the limited orientation tuning data we presented to the MICrONS mouse less accurate than orientations computed from the digital twin. However, we now show in separate experiments that the digital twin can extremely accurately capture the same orientation preferences measured from long recordings with oriented stimuli. We provide this validation data in Supplemental Fig. 7.

Alternatively, the same method of using *in silico* functional metrics to investigate “like-to-like” rule should be tested with the previous functional-EM dataset used in Lee et al., 2016, if possible, to see if the same conclusion can be drawn as this previous study.

We would love to have the opportunity to try this analysis, unfortunately the visual stimulus shown in previous studies are not diverse or extensive enough to support training of a digital twin.

This can be used to rule out the possibility that any intrinsic flaws in modeling caused the discrepancy with previous studies in the conclusion about connectivity rules and make the practical reasons suggested by the authors more likely.

We now include much more extensive validation of the *in silico* model with respect to signal correlations, and receptive fields, and orientation (Supplemental Figures 2, 4, and 7, respectively), that we hope will help address these concerns. Please also see the response to Reviewer 1 with respect to validation for more detailed discussion.

The relationship between functional metrics and synaptic connectivity is quantified by the “fold change of connection probability”, which is the connection probability within each bin of a functional metrics normalized against the overall connection probability across all bins. The term “fold change” can be confusing, as it is actually a normalized probability, but “fold change” indicates the “change” relative to a baseline instead. A term such as “normalized connection probability” might be more straightforward and consistent with its true meaning.

We agree that we had trouble communicating about this metric. We now have modified our analyses to report more interpretable metrics (co-travel distances and synapse density). For greater detail, please see below.

920 Failure to consider connectivity relative to potential synapses limits evaluation of how much observed connectivity exceeds chance. Instead of calculating the connection probability relative to ADP control, it would be a more straightforward and critical quantification to calculate a metric like the total synaptic number or area normalized by total potential synaptic length (as used in Lee et al., 2016). I believe this would offer a better metric of how synaptic connectivity is
925 selected on top of axon trajectory – which is presumably more closely related to early development. This analysis could potentially reveal new relationships between functional properties and connectivity.

We agree with the reviewer, and have reframed our results in more interpretable terms, along the lines suggested by the reviewer. We now measure axon-dendrite co-travel distances (i.e.,
930 a pairwise measurement which captures the total extent of postsynaptic dendritic skeleton within five microns from any point on the presynaptic axonal skeleton where synapses have the opportunity to form, aka the total potential synaptic length) and synapse density (i.e., the mean number of synapses per millimeter of co-travel distance, aka the total synapse number normalized by total potential synapse length).

935 The effect sizes for most relationships found seem small. In other words, separately from the other quantification concerns I was struck by the how little connectivity corresponds to functional metrics. Perhaps my intuitive expectations are incorrect and maybe generative computational models would also predict this. Alternatively, the small effect sizes point to the possibility that the visual metrics used, despite their innovativeness going beyond classical measures, still miss
940 the functional responses. Unfortunately, the MICrONS dataset does not include behavior, a major missed opportunity to go beyond orientation selectivity. In any case, if the authors believe this is a concern, I would suggest that they are explicit about the potential that most synapses don't match features simply because visual features are only one of many relevant computations in visual cortex.

945 Although the change in mean is small with regard to the standard deviation, the change in connection probability is fairly large. Furthermore, the simulation study performed in Figure 6 suggests that the effect size observed *in vivo* is relevant for visual processing and learning. Please also see our discussion of the size of the effect in responses to Reviewer 1.

We also agree with the reviewer that future work, including behavioral decision-making during the recording, combined with subsequent reconstruction, has the potential to offer new
950 insights into how these processes modulate sensory cortex activity.

The evaluation of goodness of fit in the logistic regression model. The authors used McFadden's pseudo-R-squared as a measurement for the goodness of fit in Fig. 5e. Although this is indeed a way of evaluating how well the regression is, the very small value of pseudo-R-squared makes
955 the effect size unclear for comparing the full model to the reduced models excluding one of the predictive variables; also, it raises concern how well this model is actually able to predict synaptic connectivity, or in other words, how critical those predictive variables are for predicting synaptic connectivity.

We agree with the reviewer and have improved our statistical modeling strategy to provide a better estimation of the effect size of "like-to-like". We have now adopted generalized linear mixed models (GLMM) to better estimate the effect size of "like-to-like". We utilized GLMMs to account for confounding factors like projection type and presynaptic neuron identity by modeling them as random effects. The effect size of "like-to-like" is now quantified as the coefficients of the functional similarity term and reported in Fig. 4 and supplemental
960

965 tables. To better represent model performance, we now report Nakagawa's R^2 , designed to
measure "variance explained" for GLMMs (Nakagawa and Schielzeth, 2013). Our updated
model explains over 50% of the variance at the axonal scale and over 30% of the variance at
the synaptic scale, indicating our statistical models capture a relatively large portion of the
970 data variance. We also included a baseline model containing no functional similarities as a ref-
erence point. Meanwhile, we acknowledge that the models we utilize in the paper only explain
part of the variance of connectivity since connectivity may be influenced by a lot of other as-
pects, including behavior, as the reviewer suggests. And indeed, we found that the additional
variance explained by functional similarities compared to the baseline model is relatively small
(Supplemental Fig. 6). To better understand the importance of the relatively small portion of
975 connectivity explained by functional similarity, we performed an ablation analysis in an arti-
ficial neural network in which we discovered a similar effect size of "like-to-like." The ablation
analysis showed that although "like-to-like" only explains a small portion of connectivity, the
removal of those connections has a large impact on model performance (Fig. 6).

The cell inclusion criteria based on how well the neurons can be predicted by the digital twin
980 could induce biases. The methods state that only neurons that had reliable visual responses
and were accurately predicted by the digital twin model were used for analyses. "122 out of 152
presynaptic neurons" but only "1975 out of 5502 postsynaptic neurons" passed this criteria. Al-
though clearly this is a necessary step given that the main analyses of this study are based on
in silico characterization of neurons, it raises concerns that neurons not well predicted by the model
985 might exhibit different connectivity rules from the ones included in the analyses, and therefore,
the cell inclusion criteria based on the model prediction performance may introduce biases. This
also emphasizes the necessity of validating the conclusions drawn with in silico metrics with their
in vivo counterparts and the full neuron sample.

This is a good point, which we have also addressed in response to other reviewers. As de-
990 scribed above (see "To further inspect to what extent model performance" in response to
Reviewer 1 near rebuttal line 150), we have validated our results when using different selection
criteria, and the main results are stable (although they are strongest for neurons that are well-
tuned and well-predicted by the model; Supplementary Fig. 11). We now have lowered the
inclusion criteria to include over 80% of all functionally coregistered neurons in the analysis.
995 Finally, we have reproduced the main findings of (Fig. 2) with *in vivo* signal correlation and
the full neuron sample (Supplemental Fig 3).

These findings would benefit from putting them in the context of major existing theories to help
convey their meaning and importance. The introduction section has long tangents on the history
of neuroscience and various connectivity theories that are not directly relevant to the study re-
1000 sults. Going back to Hubel-Wiesel would be warranted if their initial ideas are either confirmed
or refuted, which is not the case here. The Discussion section could point to concrete examples
as to how their findings substantively advance theoretical understanding or inspire new questions;
these less abstract discussions would bring together these results and make their importance
clearer.

1005 We reference Hubel and Wiesel only to emphasize the long history of wondering about the
relationship between function and synaptic anatomy. To make the reference more relevant, we
have changed it to focus on simple and complex cells rather than thalamic inputs. The sug-
gestion regarding the discussion is well-taken, and we have tried to revise it with this guidance
in mind, especially in manuscript line 714.

1010 **Minor comments**

The animal used to collect the MICrONS dataset was head-fixed but awake and behaving during
two-photon imaging. It might be better to describe in some details how exactly the animal was
set up and was able to behave at the beginning of the results section, since behavioral state can
greatly affect the responses of visual areas to stimuli.

1015 We have added a comment near manuscript line 132 explaining that animals were able to walk, groom, and adjust their posture during head fixation, and that we monitored behavioral state with a variety of metrics

In rows 196-201, it was described that “Visually responsive and well characterized neurons in retinotopically matched areas in V1 and HVA were chosen for manual morphological proofreading”. Explaining why selecting “retinotopically matched areas” is necessary and how that was performed would be helpful to a broader audience.

We thank the reviewer for this comment. We have expanded on this to indicate that matching retinotopic areas are expected to enrich long-range projections near manuscript line 168.

1025 Could you explain why the postsynaptic neurons greatly outnumber the presynaptic neurons (1975 vs. 122) in this study?

1030 Recall for dendritic arbors in the raw (unproofread) segmentation microns volume is very high, likely due to the larger diameter of dendritic processes, and more spatially-restricted extent. Merge errors are prevalent in dendrites, but these can be removed with high accuracy by NEURD, the automated proofreading software developed by our team (Celii et al., 2024). In comparison, axons suffer from both merge errors (which can be removed with high accuracy by NEURD) and from split errors, which do not currently have an automated extension option available. Thus, dendrites are much less costly in terms of manual proofreading than axons. New methods are in development for automated axon extension and proofreading that may help balance the cohorts in future work.

1035 In the analyses of this study, there are no statistical tests between ADP and same region control group. Although the major focus is the difference between connected group and the two control groups, the comparison between ADP and same region control will be important for drawing conclusions about the relationship between axon trajectory and certain function metrics. For example, in rows 338-342, “In contrast to signal correlation and feature weight similarity, receptive field location similarity is selected at the axon trajectory level, as evidenced by the leftward shift in receptive field location distance between connected neurons and same region control”. The conclusion that receptive field location similarity is selected at the axon trajectory level should be mainly supported by the significant difference between ADP and same region control (red vs. blue) and insignificant difference between connected neuron pairs and ADP control (black vs. red). Therefore, the conclusion here should be drawn by comparing all three groups instead of only comparing connected neurons and same region control (black vs. blue). Similarly, in rows 378-382, the conclusion that orientation tuning is selected at the axon trajectory level would be better supported by the comparison between ADP and same region control.

1050 We agree completely with the reviewer - the ability to detect selectivity at the level of axon-dendrite proximities is a key advantage of this data, and have now made the axonal-scale analysis (comparing ADP vs same-region controls), a central focus. Likewise, at the synaptic scale, we now compare connected neurons to ADP.

1055 In Fig. 3h, after breakdown into specific area-to-area groups, the differences between connected neurons and same region control (black vs. blue) became insignificant, compared to Fig. 3g, where the difference between those two groups were significant. Same in Fig. 4g, h when the neuron pairs were further split into both areas and layers. Is this due to correction for multiple comparison, or due to smaller sample sizes for each group after this breakdown?

1060 In the revised manuscript, we now avoid comparing connected and unconnected populations without grouping by area, since neuron pairs in different areas have very different mean functional similarities (e.g. HVA intra-area signal correlations are much higher than V1 intra-area

signal correlations). Because of these differences, we report comparisons for each area separately, in order that the comparison between connected neurons, ADP controls, and same region controls are not confounded with inter-area differences. All original and adjusted p-values are now reported in supplemental tables, which are referenced in the text.

1065 In rows 367-372, it was described that the “cardinal bias” of orientation tuning of the volume resulted in the U-shaped distribution in the difference in preferred orientation between neuron pairs, as shown in Fig. 3g. It would be helpful to a broad audience to briefly explain what exactly this cardinal bias is.

1070 We thank the reviewer for their comment. A short description has been added to indicate "in which orientation preference for 0 and 90 degree angles is overrepresented (Supplemental Fig. 9)"

In the figure legend of Fig. 5c, it is not explained what the black arrows from FW and RF to SC mean.

1075 We thank the reviewer for their comment. The previous Figure 5 and related analysis have now been removed from the manuscript, and we have added two new figures: The first is an analysis of higher-order functional connectivity rules (Figure 5 in the revision), and the second is a simulation study investigating the functional implications of the like-to-like effect in neural computation in an RNN (Figure 6 in the revision).

References

- 1080 Y. Benjamini and Y. Hochberg. On the adaptive control of the false discovery rate in multiple testing with independent statistics. *J. Educ. Behav. Stat.*, 25(1):60–83, Mar. 2000.
- B. Celii, S. Papadopoulos, Z. Ding, P. G. Fahey, E. Wang, C. Papadopoulos, A. B. Kunin, S. Patel, J. Alexander Bae, A. L. Bodor, D. Brittain, J. Buchanan, D. J. Bumbarger, M. A. Castro, E. Cobos, S. Dorkenwald, L. Elabbady, A. Halageri, Z. Jia, C. Jordan, D. Kapner, N. Kemnitz, S. Kinn, K. Lee, K. Li, R. Lu, T. Macrina, G. Mahalingam, E. Mitchell, S. S. Mondal, S. Mu, B. Nehoran, S. Popovych, C. M. Schneider-Mizell, W. Silversmith, M. Takeno, R. Torres, N. L. Turner, W. Wong, J. Wu, S.-C. Yu, W. Yin, D. Xenos, L. M. Kitchell, P. K. Rivlin, V. A. Rose, C. A. Bishop, B. Wester, E. Froudarakis, E. Y. Walker, F. Sinz, H. Sebastian Seung, F. Collman, N. M. da Costa, R. Clay Reid, X. Pitkow, A. S. Tolia, and J. Reimer. NEURD: automated proofreading and feature extraction for connectomics. *bioRxiv*, Apr. 2024. URL <https://www.biorxiv.org/content/10.1101/2023.03.14.532674v4>.
- 1085 L. Cossell, M. F. Iacaruso, D. R. Muir, R. Houlton, E. N. Sader, H. Ko, S. B. Hofer, and T. D. Mrsic-Flogel. Functional organization of excitatory synaptic strength in primary visual cortex. *Nature*, 518(7539):399–403, 2015.
- 1095 Z. Ding, D. T. Tran, K. Ponder, E. Cobos, Z. Ding, P. G. Fahey, E. Wang, T. Muhammad, J. Fu, S. A. Cadena, S. Papadopoulos, S. Patel, K. Franke, J. Reimer, F. H. Sinz, A. S. Ecker, X. Pitkow, and A. S. Tolia. Bipartite invariance in mouse primary visual cortex. *bioRxiv*, Mar. 2023. URL <https://www.biorxiv.org/content/10.1101/2023.03.15.532836v2>.
- L. Elabbady, S. Seshamani, S. Mu, G. Mahalingam, C. Schneider-Mizell, A. L. Bodor, J. Alexander Bae, D. Brittain, J. Buchanan, D. J. Bumbarger, M. A. Castro, S. Dorkenwald, A. Halageri, Z. Jia, C. Jordan, D. Kapner, N. Kemnitz, S. Kinn, K. Lee, K. Li, R. Lu, T. Macrina, E. Mitchell, S. S. Mondal, B. Nehoran, S. Popovych, W. Silversmith, M. Takeno, R. Torres, N. L. Turner, W. Wong, J. Wu, W. Yin, S.-C. Yu, The MICrONS Consortium, H. Sebastian Seung, R. Clay Reid, N. M. Da Costa, and F. Collman. Perisomatic features enable efficient and dataset wide Cell-Type classifications across Large-Scale electron microscopy volumes. *bioRxiv*, Jan. 2024. URL <https://www.biorxiv.org/content/10.1101/2022.07.20.499976v2>.
- 1100 P. G. Fahey, T. Muhammad, C. Smith, E. Froudarakis, E. Cobos, J. Fu, E. Y. Walker, D. Yatsenko, F. H. Sinz, J. Reimer, and A. S. Tolia. A global map of orientation tuning in mouse

- visual cortex. *bioRxiv*, Aug. 2019. URL <https://www.biorxiv.org/content/10.1101/745323v1>.
- 1110 X. Han, B. Vermaercke, and V. Bonin. Diversity of spatiotemporal coding reveals specialized visual processing streams in the mouse cortex. *Nat. Commun.*, 13(1):3249, June 2022.
- H. Ko, S. B. Hofer, B. Pichler, K. A. Buchanan, P. J. Sjöström, and T. D. Mrsic-Flogel. Functional specificity of local synaptic connections in neocortical networks. *Nature*, 473(7345):87–91, 1115 2011.
- S. Kondo and K. Ohki. Laminar differences in the orientation selectivity of geniculate afferents in mouse primary visual cortex. *Nat. Neurosci.*, 19(2):316–319, Feb. 2016.
- W.-C. A. Lee, V. Bonin, M. Reed, B. J. Graham, G. Hood, K. Glattfelder, and R. C. Reid. Anatomy and function of an excitatory network in the visual cortex. *Nature*, 532(7599):370–374, 1120 Apr. 2016.
- T. Macrina, K. Lee, R. Lu, N. L. Turner, J. Wu, S. Popovych, W. Silversmith, N. Kemnitz, J. A. Bae, M. A. Castro, et al. Petascale neural circuit reconstruction: automated methods. *bioRxiv*, 2021. URL <https://www.biorxiv.org/content/10.1101/2021.08.04.455162v1>.
- M. Mazurek, M. Kager, and S. D. Van Hooser. Robust quantification of orientation selectivity and 1125 direction selectivity. *Front. Neural Circuits*, 8:92, Aug. 2014.
- MICrONS Consortium, J. Alexander Bae, M. Baptiste, A. L. Bodor, D. Brittain, J. Buchanan, D. J. Bumbarger, M. A. Castro, B. Celii, E. Cobos, F. Collman, N. M. da Costa, S. Dorkenwald, L. Elabbady, P. G. Fahey, T. Fliss, E. Froudakis, J. Gager, C. Gamlin, A. Halageri, J. Hebditch, Z. Jia, C. Jordan, D. Kapner, N. Kemnitz, S. Kinn, S. Koolman, K. Kuehner, 1130 K. Lee, K. Li, R. Lu, T. Macrina, G. Mahalingam, S. McReynolds, E. Miranda, E. Mitchell, S. S. Mondal, M. Moore, S. Mu, T. Muhammad, B. Nehoran, O. Ogedengbe, C. Papadopoulos, S. Papadopoulos, S. Patel, X. Pitkow, S. Popovych, A. Ramos, R. Clay Reid, J. Reimer, C. M. Schneider-Mizell, H. Sebastian Seung, B. Silverman, W. Silversmith, A. Sterling, F. H. Sinz, C. L. Smith, S. Suckow, Z. H. Tan, A. S. Tolia, R. Torres, N. L. Turner, E. Y. Walker, 1135 T. Wang, G. Williams, S. Williams, K. Willie, R. Willie, W. Wong, J. Wu, C. Xu, R. Yang, D. Yatsenko, F. Ye, W. Yin, and S.-C. Yu. Functional connectomics spanning multiple areas of mouse visual cortex. *bioRxiv*, July 2021. URL <https://www.biorxiv.org/content/10.1101/2021.07.28.454025v3>.
- S. Nakagawa and H. Schielzeth. A general and simple method for obtaining R² from generalized 1140 linear mixed-effects models. *Methods Ecol. Evol.*, 4(2):133–142, Feb. 2013.
- C. M. Schneider-Mizell, A. L. Bodor, D. Brittain, J. Buchanan, D. J. Bumbarger, L. Elabbady, C. Gamlin, D. Kapner, S. Kinn, G. Mahalingam, S. Seshamani, S. Suckow, M. Takeno, R. Torres, W. Yin, S. Dorkenwald, J. A. Bae, M. A. Castro, A. Halageri, Z. Jia, C. Jordan, N. Kemnitz, K. Lee, K. Li, R. Lu, T. Macrina, E. Mitchell, S. S. Mondal, S. Mu, B. Nehoran, S. Popovych, W. Silversmith, N. L. Turner, W. Wong, J. Wu, MICrONS Consortium, J. Reimer, A. S. Tolia, H. S. Seung, R. C. Reid, F. Collman, and N. Maçarico da Costa. Cell-type-specific inhibitory circuitry from a connectomic census of mouse visual cortex. *bioRxiv*, Jan. 2024. URL <https://www.biorxiv.org/content/10.1101/2023.01.23.525290v3>.
- O. Schoppe, N. S. Harper, B. D. B. Willmore, A. J. King, and J. W. H. Schnupp. Measuring the 1150 performance of neural models. *Frontiers in Computational Neuroscience*, 10, Feb. 2016. doi: 10.3389/fncom.2016.00010.
- E. Y. Wang, P. G. Fahey, Z. Ding, S. Papadopoulos, K. Ponder, M. A. Weis, A. Chang, T. Muhammad, S. Patel, Z. Ding, D. Tran, J. Fu, C. M. Schneider-Mizell, R. C. Reid, F. Collman, N. M. da Costa, K. Franke, A. S. Ecker, J. Reimer, X. Pitkow, F. H. Sinz, and A. S. 1155 Tolia. Foundation model of neural activity predicts response to new stimulus types and anatomy. *bioRxiv*, Aug. 2024. URL <https://www.biorxiv.org/content/10.1101/2023.03.21.533548v4>.

Rebuttal Figures

connected

adp

same region

Rebuttal Figure 1. *In silico* traces for the connected, ADP, and same region neurons for a single presynaptic neuron. Randomly selected *in silico* traces from the connected, ADP, and same region population of a common presynaptic neuron in V1, sorted by signal correlation (values shown on left side of trace). The presynaptic neuron activity is shown as a heatmap behind the traces with higher gray representing larger activation. The entire trace snippet is 250 seconds in duration.

Rebuttal Figure 2. *In silico* Δ Ori , RF location similarity, and feature weight similarity vs synapse size density. Analysis in Supplemental Fig 8b repeated with synapse size density, rather than synapse density. Synapse size density is computed similarly to synapse density except that the numerator N_{syn} is replaced with the summed synaptic weight (sum of all synapse cleft volumes for a pair of neurons in $4 \times 4 \times 40 \text{ nm}^3$ voxels).

Rebuttal Figure 3. HVA local connections show significant trend that can be explained by proofreading status. Mean L_d vs signal correlation (top row) and mean $N_{syn}/\text{mm}L_d$ vs signal correlation (bottom row) shows a significant negative correlation for HVA → HVA local connections that can be explained by whether the proofread presyn was "fully" or "partially" extended. No significant relationship was seen for other projection types. In the revised version of the paper, this trend is removed by centering the analysis per presyn, that is, for all pairs, subtracting the mean L_d , $N_{syn}/\text{mm}L_d$, and signal correlation of the presyn, and only looking at differences from the mean.

Rebuttal Figure 4. Analysis repeated with *in vivo* orientation tuning. Supplemental Figure 8 reproduced except *in silico* orientation preference is replaced with *in vivo* orientation preference.

Referee 1 (Remarks to the Author):

Ding et al. have made a number of changes to the manuscript in order to address many of the questions that had been raised. Overall, these changes improved the paper and, generally, they have increased the confidence of the reader in their digital twin as it applies to this work, which was one of the major issues in the original manuscript. Despite this, there are remaining issues, some of them introduced by the modifications to the paper, that the authors need to address.

Major Issues

In their response to Reviewers, the authors explain how their proofreading strategy resulted in having a limited subset of functionally characterized neurons that were sufficiently proofread to derive a connectivity graph from. However, this does not come across in the manuscript. It still seems surprising to see the low number of neurons used for the analysis, given the large number that were functionally characterized or reconstructed. It would be helpful to provide some insight to the reader within the manuscript about why the final overlapping number decreases suddenly.

The end of the section "MICrONS functional connectomic dataset" has been rewritten to increase transparency about the number of functionally co-registered neuron pairs after proofreading.

The authors should provide more explanation of the digital twin in the manuscript, and provide some insight to the reader about the reason that they use specific implementation of the twin, in their analyses. This is particularly important since they refer to two different models in the methods.

Thank you for bringing this up. We have modified the paper to now only use one model architecture. The comprehensive technical details and validation studies for this architecture are presented in our companion manuscript "Foundation model of neural activity predicts response to new stimulus types and anatomy" (currently under concurrent review and available as a preprint at <https://doi.org/10.1101/2023.03.21.533548>), and due to the interdependent nature of the papers in this package, we agree that it is crucial for readers to understand the rationale behind our modeling decisions. We have added more context about the model and training paradigm. Specifically, this single architecture was trained on two different datasets - one consisting of only natural movies, and another combining natural movies with parametric stimuli. Adding the additional parametric stimuli to the training data increases the model's accuracy at capturing orientation tuning. These implementation details have been added to the Methods section.

Figure 1C provides a useful description of the choice of neurons with which connections are compared to. However, it begs the question: are the authors showing all control neurons? In particular, for the same area controls, one would imagine that same area controls extend far beyond the region of the reconstructed axon (especially for large areas such as V1). It might be helpful to show at least one example of how far these extend.

Thank you for the opportunity to clarify what is presented in Figure 2c. The apparent limited spatial distribution of same-area control neurons is not because we are selectively showing only a subset of controls (we are showing all of them), but rather because for functional connectomics analysis we can only use neurons as controls if they have both structural data from EM and functional responses from calcium imaging (ie, functionally coregistered neurons). While there are indeed many more neurons throughout the EM volume, the overlap of the EM reconstruction volume and the functional characterization volume naturally constrains the spatial extent of

45 our control population, regardless of how far the axon extends. To illustrate this point, we have created Rebuttal Fig. 1, which shows the full population of all neurons in gray and highlights the subset of functionally coregistered neurons in blue. Additionally, the legend in Figure 2c notes that gray dots represent "all other functionally matched neurons that are not used as controls," which should help clarify this distinction.

50 An improvement made to the manuscript is that the authors now also compute the signal correlation entirely from in vivo recordings, as opposed to only from the in silico prediction (Supplemental Figure 3). It induces confidence that the results appear similar in both cases. The authors should do a similar analysis for each result in the paper. This should be included, even if the results may not be significant from the in vivo analysis.

55 We appreciate this suggestion and agree that comparing *in vivo* and *in silico* analyses strengthens our findings. We have now expanded Supplemental Fig. 3 to include corresponding *in vivo* analyses for all applicable results in the main figures: specifically, we have added panels showing *in vivo* analyses for the findings presented in Fig 4a, b (now in Supplemental Fig. 3f, g) and Fig 5c (now in Supplemental Fig. 3h).

60 In the section titled "Functional similarity is enhanced at both the axonal and synaptic scale", the authors begin by demonstrating that the in silico model provides an effective model of the data, and end by demonstrating that the in vivo results also show a similar trend as the in silico results. Nevertheless, Supplemental Fig 2 shows that though the in silico model shows higher correlation to the benchmark than the in vivo model, the correlation is still only 60%. The authors
65 should start with the in vivo results and then introduce the in silico results only following the in vivo results.

We agree that *in vivo* validation is critical, and we have now included *in vivo* data for all possible *in silico* analyses in the main figures (see our response to the previous point). However, it is important to note that the MICrONS experiments were specifically optimized for training *in silico*
70 models using rich natural video stimuli, rather than for computing tuning directly from parametric stimuli. In particular, we did not include many stimulus repeats of the oriented stimuli that would be needed to average out noise in direct measurements. Our validation experiments (Supplemental Fig. 2) demonstrate that the *in silico* model trained on rich natural stimuli actually outperforms direct *in vivo* measurements from *limited* parametric stimuli when both are compared against a benchmark of *extensively repeated* stimuli (*in silico* correlation = 0.69; *in vivo*
75 correlation = 0.40). Given these performance differences and the dataset's design, we believe the current organization best reflects the strengths of the dataset while maintaining transparency about the correspondence between *in vivo* and *in silico* results.

In the section entitled "Factorized in silico functional representation", the authors introduce
80 Supplemental Fig 4 to demonstrate that in silico estimates of RF center correlate with estimates made using classical RF mapping stimuli. However, this correlation is not quantified in the text, and – in the figure – the authors demonstrate high r values, but do not discuss the fact that – for two of the three mice – the linear trend has a negative offset. It would be helpful for them to discuss this, and to potentially provide some explanation for the offset. Finally, they should also
85 quantify the qualitative difference in Supplemental Fig 4c.

We thank the reviewer for pointing out the negative offset observed in the linear trend of Supplemental Fig. 4 and the quantification of retinotopic map differences, which we did not comment on in the previous draft. We appreciate the opportunity to clarify these points. We have updated Supplemental Fig. 4 to include an $x = y$ line for direct comparison and have modified the panels

90 so that both axes use equal scales. This more clearly illustrates the relationship between the *in silico* estimates and the classical RF mapping estimates.

To further validate the digital twin model's ability to characterize spatial tuning of neurons, we measured *in silico* STA centers from the digital twin model's responses to sparse random noise stimuli, the same stimuli that were shown for *in vivo* STA characterization. The digital twin
95 model not only faithfully recovers the *in vivo* STA centers for neurons with well-characterized *in vivo* STAs but also for neurons with much noisier *in vivo* STAs (Supplemental Fig. 4f, g).

We agree with the reviewer that quantifying the differences between the retinotopic maps is crucial. We quantified the coherence of the retinotopic maps as Spearman's correlation between the cortical distances and the retinotopic distances of randomly selected neuron pairs within a cortical
100 region that does not contain a retinotopic reversal (Supplemental Fig. 4). This quantification confirmed the qualitative observation that the retinotopic maps measured with model readout location are the most coherent, followed by *in silico* STA centers, with *in vivo* STA centers being the most noisy.

As the reviewer noted, the estimated RF centers with the *in vivo* STAs and the readout locations
105 are highly correlated, but the absolute values differ slightly by scaling and shifting.

However, our analyses focus on relative RF center distances and are inherently robust to linear transforms of the RF center locations. We further validated this by replicating our main finding with *in silico* STA centers (Supplemental Fig. 4j, k).

We opted to focus the analysis in the main figure on the readout locations since the model read-
110 out location produces the most coherent retinotopic map and is capable of learning a more realistic representation of RF centers, unconstrained by the simple parametric stimuli and the linear assumption underlying STA estimation.

In the Discussion section, we now discuss the differences between *in vivo* dot mapping STA and *in silico* readout location.

115 In many previous works on like-to-like connectivity, authors have quantified changes in like-to-like connectivity related to physical distance between neuron somas. Given the dense reconstruction in the MICrONS dataset, the authors could also quantify changes with respect to somatic distance, and see how this differs from prior works. This would provide a mid-way control analysis between their ADP controls, and the same area controls (which can extend much further away
120 from the presynaptic neuron's axon).

We agree with the reviewer that the MICrONS dataset provides an excellent opportunity to study the relationship between like-to-like connectivity and somatic distances. We quantified how functional similarity varies across somatic distances for connected and unconnected neuron pairs of all projection types. We considered four measures of functional similarity *in vivo* signal correlations,
125 *in silico* signal correlations, feature weight similarities, and RF center distances. Across projection types and somatic distances, the functional similarities of connected neurons are consistently higher than those of unconnected neuron pairs. Furthermore, focusing on V1 intra-area projections, we observe a weak negative correlation between functional similarity and somatic distances in V1 intra-area projections. This aligns with previous findings (Cossell et al., 2015),
130 which indicates that neurons close to each other are functionally more similar within a visual area. However, this negative correlation between functional similarity and somatic distance was not consistently observed in other projection groups. The interpretation of this relationship becomes considerably more complex when projections cross a retinotopic reversal, as the relative positions within the visual field are no longer straightforwardly related to cortical distances. The
135 sophisticated control design necessary to address these complex relationships approaches the ADP control design already in use in the study. Given the complexities of interpreting these re-

relationships across retinotopic boundaries, we have included the full analysis across all projection groups in this rebuttal (Rebuttal Fig. 2), but have limited the manuscript figure (Supplemental Fig. 13) and discussion to the V1 intra-area projections to provide comparison for continuity with previous studies.

140

The authors use statement such as “selectivity . . . exists at the scale of axon trajectories, but not at the scale of individual synapse formation”, based on their comparison of the ADP controls and same area controls. It seems that the more correct statement would be that: there is no further selectivity gained at the synapse level, compared to that obtained at the axon trajectory level.

145

We have made this change in the text.

There is a limited characterization of the final result in the paper (section titled “Like-to-like connectivity and its functional role in artificial neural networks”). Although the concept is interesting, it’s not clear how general the authors results are, or if they are specific to some particular training algorithm. If the authors include this result (and if it is important enough to add to their abstract), then they need to explore the space of possible RNN models, and provide some insight either as to why their results apply, as a general principle, or to explain why their results apply to a subset of models, trained in a specific way.

150

We recognize that we failed to sufficiently discuss the generality of our results in this section.

155

In particular, we did not sufficiently emphasize that attractor-based recurrent artificial neural networks like Hopfield networks are already known to produce like-to-like connectivity, and like-to-like connectivity is known to play a central role in their computational abilities. For example, in classical Hopfield networks, weights are trained to be exactly proportional to the covariance matrix of activations, so like-to-like connectivity emerges by design. Because the emergence and role of like-to-like connectivity in Hopfield networks and other attractor based models is already known, it was not necessary for us to demonstrate this with examples. However, we failed to make these points clear in the previous version of the manuscript. We have now rewritten the section to clarify these points.

160

The RNN approach was chosen to evaluate the extent to which like-to-like connectivity in artificial networks generalizes to models for which it is not “baked in” by design. We chose to use a vanilla RNN trained with backpropagation on MNIST because it is a simple and standard baseline model with a minimal number of hyperparameters. For several different hyperparameters (number of hidden units, number of time steps, and batch size), we tested sensitivity by varying the hyperparameter across a 4-fold range of values. Under this analysis, our overall findings did not change. We also verified that the results are similar when the model is trained on Fashion-MNIST in place of MNIST. In summary, our results are already known to generalize to other models (e.g., Hopfield nets) and are robust within our RNN model.

165

170

However, the reviewer is correct to point out that we did not explain *why* we think the model produces like-to-like connectivity, and how general we expect this finding to be. Like-to-like connectivity is expected to emerge in recurrent networks that satisfy two properties: 1) Increased connectivity promotes stronger functional covariance, and 2) Similar responses to similar stimuli are promoted by learning. We appreciate the opportunity to clarify these important points and have added text to the section in question to better explain these findings and interpretations.

175

We would finally like to mention that our RNN example helps clarify how a relatively weak like-to-like effect (similar to the magnitude in our data; which is weaker than expected in a Hopfield network) can emerge and play a role in learning and computation. We made sure to clarify this point in the revised manuscript.

180

Minor Issues:

The authors use the term “lesion studies” in their abstract, to refer to simulated perturbations of their RNNs. “Perturbations” would be a term to their manipulations of an artificial network, given that there is no physical lesion involved.

"Lesion studies" and "lesions" have been replaced with "Ablation studies" and "ablations" throughout the text to be consistent with the language in the results and the larger artificial neural network literature.

There is a potential confusion in the use of the word “HVA” in the manuscript, since this can refer to a number of different areas. For example, HVA → HVA connections could mean connections within a single one of the HVAs, or from one HVA to another. Given this potential confusion, the authors should use more precise wording.

A clarifying line has been added to the first occurrence of HVA to HVA in the text, specifying that it includes both connections within an HVA and between two HVAs

It would be helpful to know the distribution of presynaptic neurons (and their projections) in each HVA.

The distribution of presynaptic neurons and their projections is now reported in Supplemental Tab. 2 and 3.

The authors explain the axis of Figure 1E and F in the text (particularly, why the y scale extends from negative to positive). It would be helpful for the authors to also include (or refer to) this explanation in the figure legend.

We have now added a reference to the result section in the figure legend for 2e.

The authors include many figures where the x and y axis join at a point that is not the origin. This is a potential source of confusion for readers, and it would be helpful to separate the axis, to ensure that readers know that the axes do not extend to zero.

We agree this would improve clarity for readers. We will ensure all figures have properly separated axes that do not falsely imply extension to zero in the final version of the manuscript.

The authors use the word “multi-tiered”, which does not have a clear meaning from the text. This is especially difficult since it is used as a title of one of their sections.

This title has now been rewritten as "multi-scale", to clearly match the language used throughout the rest of the text and figures.

In page 5, line 258 – 261: The sentence appears incomplete. It's not clear what hypothesis the authors are testing.

To avoid confusion, this sentence has now been rewritten and expanded to state the definition of signal correlations and to state explicitly what "the hypothesis of like-to-like connectivity" is.

The authors should check the ordering of their supplemental / supplementary figures. It appears that they are not addressed in order in the text, which adds some confusion for the reader. Additionally, they should pick one nomenclature (i.e. either Supplemental or Supplementary) since it's currently difficult to find all references to extended figures.

Figures now appear in the order they are referenced in the text.

For Supplemental Fig 4, it would be helpful if the authors would center the axes at the origin (and use square plots) – so that any visual difference from the $x = y$ line would be clearly identifiable.

225 We have now adjusted the axes to use the same scale. We have also added the $x = y$ line for comparison so that any deviation from the $x = y$ line is apparent.

It would be helpful for the authors to also show an example of their same region controls for a layer specific analysis (i.e. similar images as in Supplemental Fig 1, but for Fig 4).

230 Supplemental Fig 5 is now included which shows the same neurons from Fig. 2c with their same region and ADP controls for some example layer specific analyses.

The schematic diagram in Fig 5 is hard to understand, as it's not clear how neurons are placed in a functional space. The authors should clarify this schematic.

We now clarify in the legend that in the functional space, neurons with similar functional properties will be placed closer to each other.

235 **Referee 2 (Remarks to the Author):**

This revised manuscript is much improved relative to the original version in several respects.

Most importantly, the narrative more clearly points out the novel insights that can be made from the analyses conducted, and new analyses and revised presentation clarify these important distinctions. Notably, the data and analyses not only show that connectivity/function relationships are
240 generally like-to-like across layers within V1 and across both feedforward and feedback connections, but they also compare specificity at synaptic scale to specificity related to similar proximity without synaptic contacts. These new insights, along with demonstration of the power and future promise of this approach make this manuscript suitable for publication in Nature in the context of a larger package of related manuscripts.

245 It is noteworthy that reviewers were not notified that this manuscript is being considered in the context of a package of manuscripts, a fact that only became clear from reading responses to reviewers. This is important because the synergistic impact of a set of manuscripts can be much greater than the sum of the parts.

The authors have addressed all of the concerns that I raised in my prior review.

250 Thank you!

Referee 3 (Remarks to the Author):

I find the revised manuscript improved in several ways, the writing is clearer, the story easier to follow, and the additional analyses and controls add to the rigor of the study. The authors show, for example, that the results in Fig. 2 also hold when in vivo data, and not just in silico data
255 were used. The authors have thoroughly addressed all my comments in their rebuttal; however, more should be done to incorporate the additional information they provide into the manuscript. Following up on my question about the completeness of their reconstructions, I have one more concern regarding an anatomical bias that could arise from how presynaptic neurons were selected.

260 The arborizations of these pyramidal cells are many $100\mu\text{m}$ in diameter, which means most neurons in the dataset are not truly complete because at least some of their dendrites and axons extend beyond the boundaries of the EM volume. Critically, the closer a neuron's soma sits to a volume border, the more of its axons and dendrites must be truncated.

Let's assume a presynaptic neuron sits close to a volume border, then its synaptic connections
265 with its neighbors would be missed if they lie outside the imaged EM volume (on the truncated part of its axon and the truncated dendrites of the postsynaptic neurons). Therefore, wouldn't many of the postsynaptic partners of this presynaptic cell be mislabeled as 'ADP controls' or

'same region controls' in the analyses (e.g. Fig. 2d), even though they might be connected outside the volume?

270 This means that there is a general bias that the synaptic connectivity in the dataset is underestimated. If all presynaptic cells would be in the same column, this bias would be similar for all, but Figure 1d suggests that the locations of presynaptic neurons are relatively scattered across the X-Y plane with some cell bodies as close as 50-100 μ m to a volume border, where a much larger proportion of their arborizations is truncated. Therefore, the extent to which synaptic connectivity is underestimated is heterogenous across the dataset and is expected to be much larger for
275 neurons closer to the borders, compared to neurons in the center of the EM volume.

If cells with similar functional responses are spatially clustered, then such a bias could have the effect that the like-to-like connectivity rate is systematically underestimated for cells close to the volume borders: the connection rate with its nearest (and functionally most similar) neighbors
280 would be underestimated, whereas the connection rate with cells that sit further away and in the center of the volume (and are functionally less similar) is not affected as much. (The opposite would be true for cells in the middle of the volume: their like-to-like connectivity rate might be overestimated.)

The authors mention that nearby neurons do have higher signal correlations (Rebuttal line 637),
285 mouse V1 is retinotopically organized (e.g. Supplemental Figure 4) and there is evidence for (weak) spatial clustering of orientation preference (Ringach et al., 2016, Fahey et al., 2019). Because there is indication for spatial clustering in V1, I wonder if the different degrees of completeness of the presynaptic neurons in the study could affect the results.

The reviewer raises an important point about potential biases from volume truncation effects.
290 We believe this limitation actually makes our findings of like-to-like connectivity more conservative.

Given that functionally similar neurons tend to be spatially clustered (as noted by the reviewer), truncation effects would disproportionately cause us to miss connections between functionally similar neurons near volume borders, leading us to underestimate, not overestimate, the true
295 strength of like-to-like connectivity.

Our analytical approach using axon-dendrite proximities (ADPs) as controls provides inherent robustness against truncation effects. When we identify ADPs, we are by definition examining regions where both axon and dendrite are successfully reconstructed within the volume. The comparison between connected pairs and ADP controls therefore uses only the successfully reconstructed portions of arbors that could have potentially formed synapses. While truncation reduces
300 our total observations, it should not systematically bias the relationship between connectivity and functional similarity within the observed volume.

In the case where multiple ADPs are formed between a presynaptic/postsynaptic pair and some are located outside the volume, this is controlled for by the normalization by ADP length.

305 In the case where we compare ADPs against same area controls, only functional units matched to the EM (and therefore inside the EM boundaries) are included as controls, acting as a partial control for the spatial distribution of possible reconstructed axon trajectories inside the EM volume.

Even when connections are missed due to truncation and potential postsynaptic partners are incorrectly classified as ADP or same-region controls, this misclassification would only make our
310 control groups more functionally similar to the connected population. This conservative bias would make it harder, not easier, to detect differences between connected and unconnected populations. The fact that we still observe robust like-to-like connectivity despite this bias strengthens

our conclusions.

315 Additionally, our statistical approach compares connectivity patterns within individual presynaptic neurons, using each neuron as its own control. This within-neuron analysis helps control for heterogeneity in truncation effects across different spatial locations, as any truncation-related biases would affect both the connected and control populations for a given presynaptic neuron. Therefore, while border effects and truncation certainly impact our ability to recover the complete connectome, these limitations would tend to produce false negatives rather than false positives in detecting like-to-like connectivity. The fact that we observe robust like-to-like connectivity despite these conservative biases suggests this is a genuine organizational principle of cortical connectivity.

We have addressed the incomplete reconstructions in the Discussion section.

325 **Minor Issues:**

I appreciate that the authors repeated their analyses using synapse size density (total synaptic weight of connections). The authors should add a half sentence to the Results mentioning this analysis and showing Rebuttal Fig. 2 in a supplemental figure – maybe just add it to Supplementary Fig. 8? As I explained initially, I believe it is an important analysis because total synaptic weight is the anatomical correlate that should align best with functional similarity, and it is reassuring that the main findings hold.

We have now added Rebuttal Fig. 2 as Supplemental Figure 13 and have referenced it in the Results at the end of section "Orientation tuning is like-to-like within V1 at both axonal and synaptic scales".

335 The authors have thoroughly addressed my questions about the reliability of the synapse detection and a potential bias for missing smaller versus bigger synapses. However, this should also be addressed in the paper itself, such that readers can assess for themselves what error can be expected in the connectivity graph. More specifically:

- o The precision of the automated synapse detection in the volume, which the authors describe in their rebuttal, should be reported in the paper, with a reference to the MICrONS paper.
- o Along the same lines, the authors should briefly discuss that there is a bias for missing smaller synapses, and whether they expect this to influence the results.
- o The percentages of untraceable endings for dendrites and axons, which the authors mention in the rebuttal, should be reported somewhere in the paper.

345 We have now added the precision of automated synapse detection to the discussion section, with a reference to the consortium paper. We now describe the bias for missing smaller synapses in the discussion. The percentage of untraceable endings for dendrites and axons is now included in section "Manual proofreading completion" in the Methods.

The authors now state that the digital twin can "accurately predict the response of a neuron to any arbitrary visual stimulus." (Line 104f). This is a very strong statement, and I imagine it is difficult to justify this experimentally because not all arbitrary stimuli can be presented during an experiment.

This statement has been rewritten as "accurately predict the response of a neuron to naturalistic visual stimulus."

355 By just reading the Results, I find it difficult to understand what exactly is meant by 'signal correlation' (line 259). Only the reference to Cossell et al. in line 262 suggests that 'signal correlation' must be the pairwise Pearson correlation coefficient of two neurons in response to a stimulus. For readability, could the authors provide more information here?

This line has now been expanded to include the definition of signal correlation.

³⁶⁰ **References**

- L. Cossell, M. F. Iacaruso, D. R. Muir, R. Houlton, E. N. Sader, H. Ko, S. B. Hofer, and T. D. Mrsic-Flogel. Functional organization of excitatory synaptic strength in primary visual cortex. *Nature*, 518(7539):399–403, 2015.

Rebuttal Figures

Rebuttal Figure 1. MICrONS volume with soma locations and proofread neuron skeletons Side view (a, b) and top-down view (c, d) of the V1 (a, c) and HVA (b, d) neuron skeletons from Fig 2c. Gray scatter points are the somatic locations of all identified neurons in the volume from CAVE table 'nucleus_neuron_svm'. Blue scatter points are all functionally co-registered neurons. Orange triangles represent soma locations. Scale bars: 100 μm .

Rebuttal Figure 2. Pairwise functional measurements for connected and unconnected pairs vs soma-soma distance. Mean of *in vivo* signal correlations (**a**), *in silico* signal correlations (**b**), feature weight similarity (**c**), and RF center distance (**d**) vs soma-soma distance for connected and unconnected pairs across all projection types.